# Phenotypic evolution of SARS-CoV-2 spike during the COVID-19 pandemic

Wilhelm Furnon [1,3], Vanessa M. Cowton [1,3], Giuditta De Lorenzo [1,3], Richard Orton[1], Vanessa Herder [1,2], Diego Cantoni[1], Georgios Ilia [1], Diogo Correa Mendonca[1,2], Karen Kerr[1,2], Jay Allan[1], Nicole Upfold[1], Gavin R. Meehan[1,2], Siddharth Bakshi[1], Udeet Ranjan Das[1], Sergi Molina Arias [1,2], Marion McElwee[1,2], Sarah Little[1], Nicola Logan[1], Kirsty Kwok[1], Katherine Smollett[1], Brian J. Willett [1], Ana Da Silva Filipe [1], David L. Robertson [1], Joe Grove [1], Arvind H. Patel [1,2,4] ✉ & Massimo Palmarini [1,2,4] ✉

SARS-CoV-2 variants are mainly defined by mutations in their spike. It is therefore critical to understand how the evolutionary trajectories of spike affect virus phenotypes. So far, it has been challenging to comprehensively compare the many spikes that emerged during the pandemic in a single experimental platform. Here we generated a panel of recombinant viruses carrying different spike proteins from 27 variants circulating between 2020 and 2024 in the same genomic background. We then assessed several of their phenotypic traits both in vitro and in vivo. We found distinct phenotypic trajectories of spike among and between variants circulating before and after the emergence of Omicron variants. Spike of post-Omicron variants maintained enhanced tropism for the nasal epithelium and large airways but displayed, over time, several phenotypic traits typical of the pre-Omicron variants. Hence, spike with phenotypic features of both pre- and post-Omicron variants may continue to emerge in the future.

The COVID-19 pandemic has been characterized by the continuous emergence of phenotypically distinct SARS-CoV-2 variants defined largely by mutations within spike. This protein mediates virus entry into cells, and it is a critical determinant of virus infectivity, tissue/host tropism and pathogenesis[1–4]. Spike is also the key target of the immune responses elicited in either infected or vaccinated individuals. Hence, to monitor the course of COVID-19, there is a need to correlate spike variability with virus phenotype.

The role of spike during the virus replication cycle is well understood[1,2]. During virion biogenesis, SARS-CoV-2 spike undergoes proteolytic maturation into S1 and S2 subunits by cleavage at a polybasic furin cleavage site (FCS). The S1 subunit carries the receptor binding domain (RBD) and the N-terminal domain (NTD), and it is a

primary determinant of virus host-range[2,5]. During virus entry into cells, the RBD undergoes a conformational change, enabling it to interact with the receptor angiotensin-converting enzyme 2 (ACE2)[5,6]. ACE2 interaction destabilizes spike to promote membrane fusion, which requires a secondary proteolysis event that can occur at the plasma membrane by transmembrane protease serine 2 (TMPRSS2) or, following endocytosis, in the endosome by cysteine proteases cathepsin L and cathepsin B[2,7].

Two SARS-CoV-2 lineages (A and B) emerged from the initial COVID-19 outbreak in late 2019 in China, with the B lineage dominating the subsequent pandemic[8]. The early pandemic was characterized by the emergence of viral variants with relatively few point mutations within the spike protein[9]. The most significant early spike change was

[1]MRC-University of Glasgow Centre for Virus Research, Glasgow, UK. [2]CVR-CRUSH, MRC-University of Glasgow Centre for Virus Research, Glasgow, UK. [3]These authors contributed equally: Wilhelm Furnon, Vanessa M. Cowton, Giuditta De Lorenzo. [4]These authors jointly supervised this work: Arvind H Patel, Massimo Palmarini. ✉e-mail: arvind.patel@glasgow.ac.uk; massimo.palmarini@glasgow.ac.uk

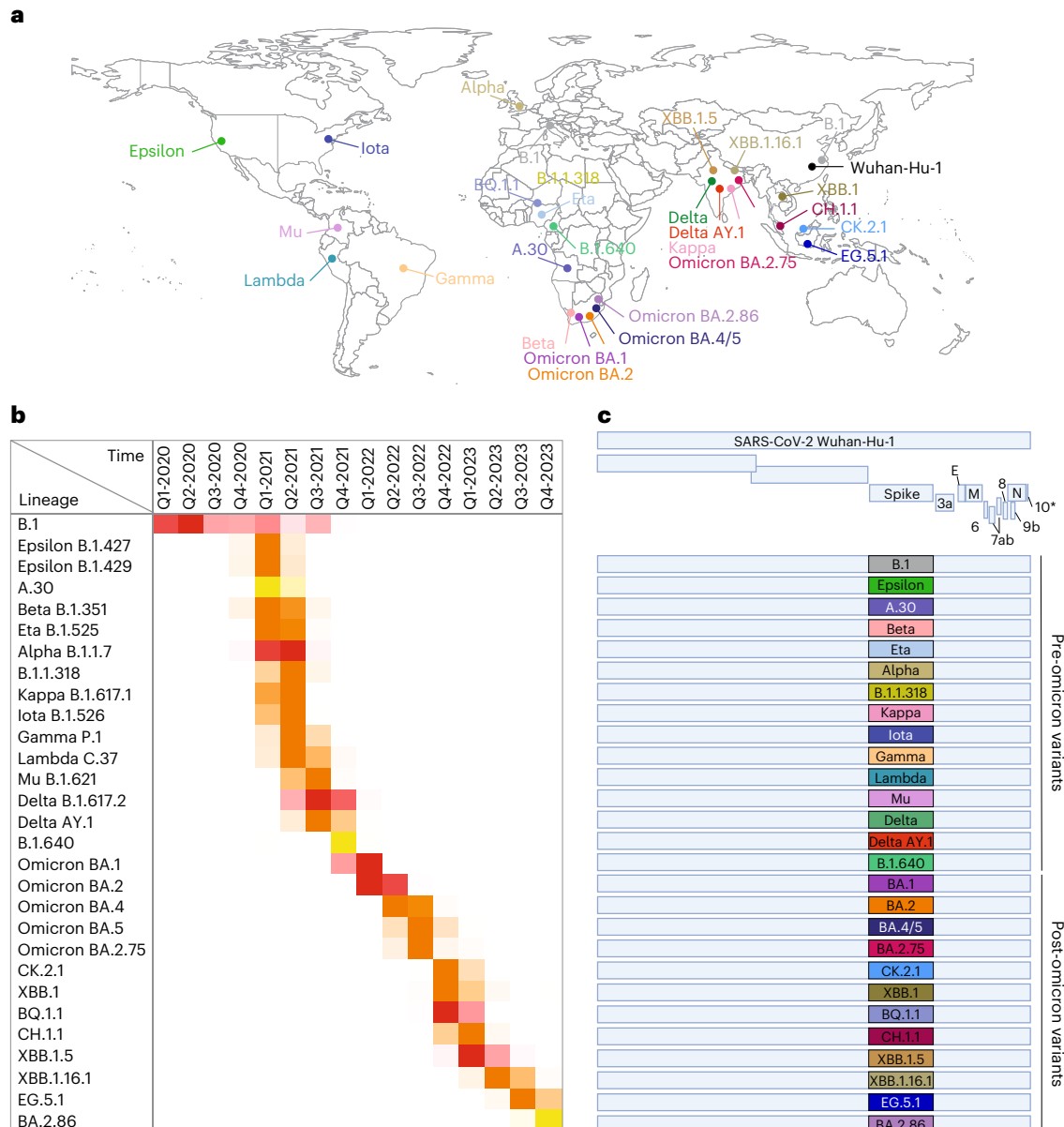

**Fig. 1 | SARS-CoV-2 variants analysed in this study. a**, Map highlighting the country where the first sequence of the variants assessed in this study was reported on GISAID. **b**, Global quarterly sequence count for each variant, scaled independently to highlight the peak occurrence of each SARS-CoV-2 variant. Red scaling is used for variants with >100,000 sequences, while orange and yellow are used for those with more or less than 1,000 sequences, respectively. **c**, Schematic representation of the spike recombinant viruses used in this study. 10* refers to Orf10 as a putative accessory protein, as there is ongoing debate in the literature regarding its existence and expression.

D614G that stabilizes the virion-associated S1/S2 trimer and confers enhanced virus transmissibility[10]. This variant essentially defined the B.1 lineage that remained dominant until autumn 2020 (Fig. 1a,b)[2,10,11]. Between 2020 and 2021, several more heavily mutated and more transmissible variants (henceforth 'pre-Omicron') emerged, with Alpha, Beta and Gamma being declared 'variants of concern' (VOCs)[12–14]. Other variants such as Epsilon, Eta, Iota, Lambda, Kappa, B.1.1.318 and A.30 were designated variants of interest (VOI) (Fig. 1a,b)[15]. Subsequently, the Mu variant emerged in Colombia, while Delta (and its derivative Delta AY.1) spread globally; concurrently, B.1.640.1 emerged in Africa[15].

A major epidemiological shift occurred in late 2021 with the emergence of Omicron BA.1 and then BA.2 in southern Africa[16] (herein 'post-Omicron' including derived sublineages). Omicron carries more than 30 spike amino acid substitutions. As 2022 unfolded, Omicron BA.4, BA.5 and BA.2.75 emerged in southern Africa[17] and

India (Fig. 1a,b)[18]. Other notable variants included BQ.1.1, CK.2.1 and CH.1.1 (sublineages of BA.5 and BA.2.75, respectively). The recombinant variant XBB[19–21] and its derivatives such as XBB.1, XBB.1.5, XBB.1.16.1 and EG.5.1 dominated for most of 2023[22–28]. BA.2.86, was designated a VOI in November 2023 following an increase in global prevalence[15,27].

As vaccine coverage and population immunity increased, spike mutation-associated immune escape was the hallmark of variants that successfully spread worldwide[3,4]. However, other phenotypic changes were also associated with some variants; Delta displayed an increased virulence[29–31], while Omicron preferred a new cell entry pathway and predominantly infected the upper respiratory tract causing a relatively milder disease[29,30,32–34].

Understanding the virus phenotypic implications of spike variability is paramount to predicting the evolutionary trajectories of SARS-CoV-2 and the course of COVID-19. Here, using viruses with an

identical genomic background but carrying spikes of variants that emerged between 2020 and 2024, we systematically examined the evolutionary trajectory of distinct phenotypic traits throughout the pre- and post-Omicron phases of the pandemic.

## Results

### Recombinant viruses

We generated by reverse genetics a panel of 27 viruses containing the spike glycoprotein of 'pre-Omicron' ($n = 15$) or 'post-Omicron' ($n = 12$, including Omicron/BA.1) variants within the same genomic backbone as the ancestral Wuhan-Hu-1 virus (Fig. 1a–c and Extended Data Fig. 1a,b)[35]. The virus containing the B.1 spike D614G mutation (S:B.1), was used as a reference control virus in each experiment.

### Virus replication kinetics in Calu-3

We first assessed the replication of the pre- and post-Omicron viruses in Calu-3, a widely used human lung-derived cell line (Fig. 2a,b)[1,36,37]. The replication kinetics of the pre-Omicron group were in general higher than the post-Omicron viruses (Fig. 2a–c). The pre-Omicron viruses generally replicated similarly to the reference S:B1; only S:Epsilon and S:Eta were slower, while S:Lambda, S:Mu, S:Delta and the related S:AY.1 replicated with faster kinetics.

We observed a broader spectrum of replication kinetics among the post-Omicron group, with seven viruses replicating slower than S:B.1; particularly notable among this subgroup were the earlier variants S:BA.1, S:BA.2, in addition to S:BA.2.75. The remaining viruses replicated either at a rate comparable to S:B1 (S:BA.4/5, S:CH.1.1, S:CK.2.1 and S:XBB.1.16.1) or with slightly higher kinetics (S:EG.5.1 and S:BA.2.86) than S:B.1 (Fig. 2b,c). Thus, spikes of recently emerged variants have regained the ability to impart replication fitness in lung cells.

For comparison, we also assessed selected full-length wild-type (WT) pre-Omicron and post-Omicron variants. Importantly, while all the pre-Omicron WT VOCs replicated at levels similar to or higher than S:B.1, all the WT post-Omicron VOCs tested replicated weakly in Calu-3 cells (Extended Data Fig. 2a,b). Side-by-side comparison between S:EG.5.1, S:BA.2.86 and their WT counterparts confirmed that the recombinant spike viruses displayed higher replication kinetics than the parent variants (Extended Data Fig. 2c,d). Thus, these data suggest that both the pre-Omicron and post-Omicron spikes accumulated adaptive mutations favouring replication in human lung-derived cell lines. However, differences outside spike reduced the ability of post-Omicron viruses to replicate in Calu-3 cells compared with the pre-Omicron variants.

### Fusogenicity in lung cell lines

Expression of SARS-CoV-2 spike protein on the cell surface and subsequent ACE2 interaction and cleavage by TMPRSS2 can mediate cell–cell fusion[38,39]. This phenotype varies between different SARS-CoV-2 VOCs[30,32,40–42]. We therefore systematically compared the kinetics of cell–cell fusion induced by our panel of viruses using an A549-hACE2-TMPRSS2 (AAT) cell-based GFP-split fusion assay[32].

Among the pre-Omicron group, S:Eta, S:Kappa, S:Mu and S:B.1.640 were as fusogenic as S:B.1, while S:Alpha, S:Beta, S:Gamma, S:Epsilon, S:Lambda, S:Iota, S:B.1.1.318 and S:A.30 exhibited reduced fusogenicity (Fig. 2d)[43]. As reported previously, S:Delta was more fusogenic than S:B.1 (refs. 30,40) and this was also true for the other viruses carrying spikes that emerged later in the pre-Omicron phase.

Consistent with previous reports, viruses expressing spike from Omicron BA.1 or its immediate descendants BA.2, BA.2.75 and BA.4/5 were only weakly fusogenic (Fig. 2e)[18,30,32,41,44]. However, the more recent Omicron spike subvariants such as S:BQ.1.1, S:CK.2.1, S:CH.1.1, S:XBB.1, S:XBB.1.5 and S:XBB.1.16.1 displayed increased fusogenicity compared with viruses with earlier Omicron spikes (but not S:B.1). Importantly, S:EG.5.1 and S:BA.2.86 were as fusogenic, if not more, than S:B.1 (Fig. 2e) in contrast to previous reports[27,45]. Hence, recently

emerged post-Omicron spikes have independently regained the ability to induce cell–cell fusion, as well as replication fitness in lung cell lines, at levels similar to the ancestral B.1.

For comparison, we also tested WT VOCs and the reference S:B.1 for their ability to induce cell–cell fusion (Extended Data Fig. 2e,f). As shown previously[30,40], the Delta VOC exhibited significantly increased fusogenicity relative to S:B.1 (Extended Data Fig. 2e,f). The Beta and Gamma VOCs had similar fusogenic activity to S:B.1; Omicron and its sublineages exhibited no (BA.1 and BA.2) or little (BA.4 and BA.5) cell–cell fusion (Extended Data Fig. 2g), while the Alpha VOC showed an intermediate fusogenicity profile. Intriguingly, the fusogenicity of WT Beta and Delta VOCs was greater than their corresponding spike recombinant viruses. These data suggest that viral determinants outside of spike might also contribute (Fig. 2f and Extended Data Fig. 2e,f).

### Spike processing and entry route usage

Spike is subjected to proteolysis by host proteases to enable membrane fusion and subsequent virus entry into the host cell. We next analysed purified virus particles by western blotting. We determined that most recombinants displayed comparative levels of spike processing relative to the S:B1 control (Extended Data Fig. 3).

SARS-CoV-2 can utilize two entry routes, both involving membrane fusion but occurring either at the cell surface or within endosomes, following proteolytic processing of the spike by proteases such as TMPRSS2 or cathepsins, respectively[2,46–48]. Omicron BA.1 and its sublineages, unlike the pre-Omicron variants, have a reduced dependency on TMPRSS2 and a preference for endosomal cathepsins[32,41,42].

To investigate the entry preference of our panel of viruses, we infected AAT cells in the presence of camostat or E64d, inhibitors of TMPRSS2 or Cathepsin L, respectively. Notably, AAT cells express both human ACE2 and TMPRSS2 ectopically and as such are permissive for infection by both entry routes. We observed a broad range of sensitivities to the two protease inhibitors (Fig. 3a,b). Generally, the pre-Omicron viruses showed a comparable or increased sensitivity to camostat relative to S:B.1 virus. The same group showed a comparable or reduced sensitivity to E64d, suggesting that these viruses preferred TMPRSS2-dependent cell surface fusion for cell entry. However, there were exceptions. S:A30 was inhibited poorly by camostat but was sensitive to E64d, indicating a dependence on cathepsin-mediated endosomal fusion (Fig. 3a,b), while S:Iota was relatively insensitive to both camostat and E64d, indicating competence for both entry pathways.

The patterns of drug sensitivity were not as conserved in the post-Omicron group. S:BA.1 was highly sensitive to both inhibitors, suggesting that both proteases are necessary for efficient entry. S:BA.2.75 and S:BA.2, while remaining almost completely inhibited by E64d, retained ~50% or more infectivity following camostat treatment. By comparison, S:BA.4/5 and S:EG.5.1, similar to S:Iota, were not severely inhibited by either treatment, indicating a flexibility for spike processing by either protease (Fig. 3a,b). Other Omicron sublineages showed either an Omicron BA.1-like profile (S:CH.1.1, S:BQ.1.1 and S:XBB.1.16.1), or a restored pre-Omicron-like protease sensitivity pattern (S:CK.2.1, S:XBB.1, S:XBB.1.5 and S:BA.2.86; Fig. 3a,b).

These data suggest that the potential to use the endosomal pathway may have evolved independently before the emergence of Omicron. Importantly, similar to what we showed above for replication and fusogenicity in lung cells, some post-Omicron variants (particularly more recent ones) have regained the ability to use entry routes favoured by the pre-Omicron viruses.

### Determinants of entry pathway

We next infected AAT cells with S:B.1, S:BA.1, S:A.30 S:Epsilon, S:Iota or S:BA.4/5 in the presence of both protease inhibitors. (Fig. 3c). All viruses were strongly inhibited by the combined treatment, ruling out a role of alternative activating proteases in AAT cells. Consistent with Calu-3's poor permissivity for virus entry via the endosomal

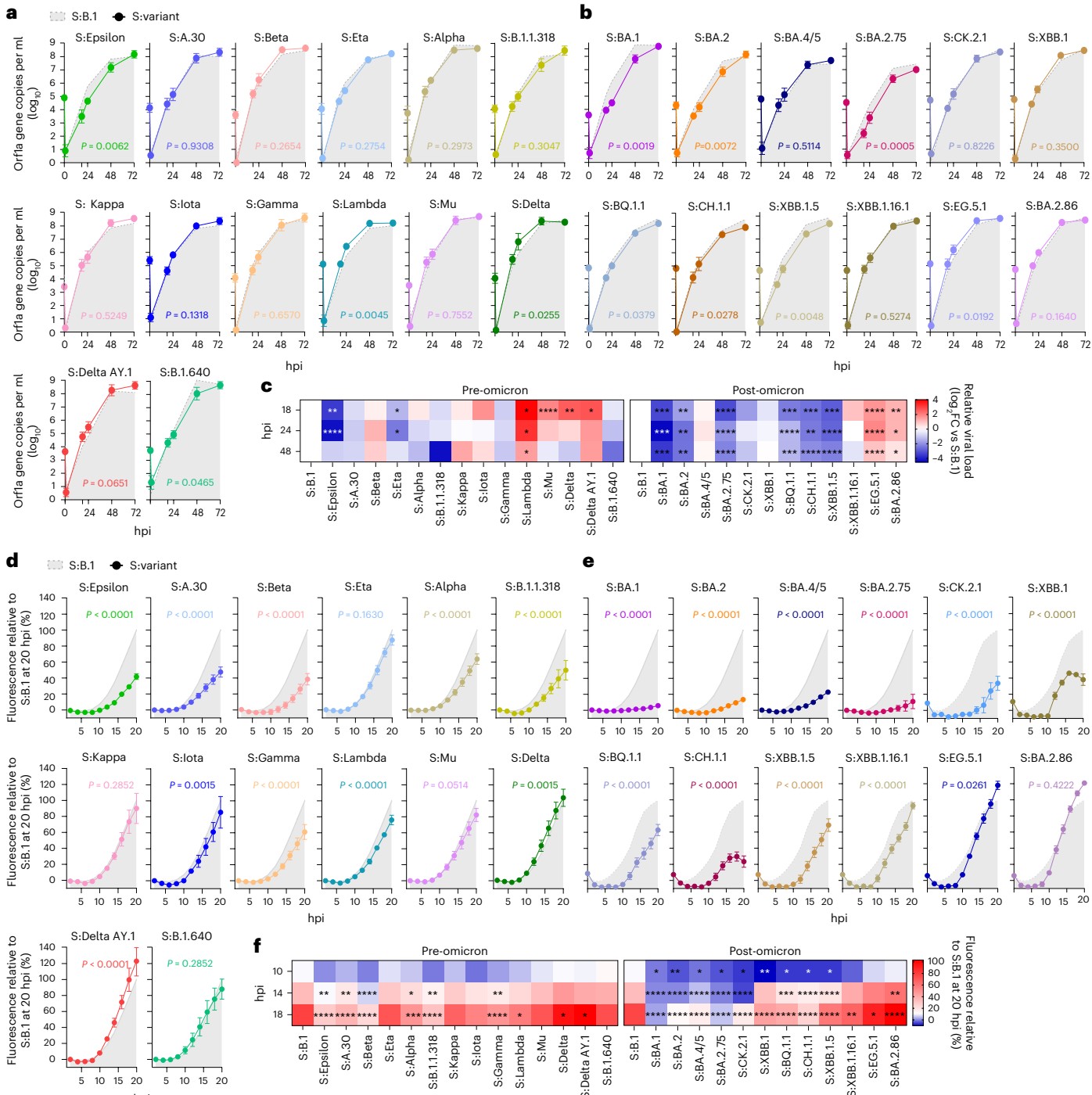

**Fig. 2 | Replication kinetics and fusogenicity of SARS-CoV-2 recombinant viruses in lung cell lines. a**,**b**, Replication kinetics of pre-Omicron (**a**) and post-Omicron (**b**) viruses in Calu-3 cells. Cells were infected with equal viral genome copies of each virus and the amount of virus released in the supernatant was quantified by RT–qPCR at different timepoints post infection (hpi) as indicated. Recombinant viruses were assessed across distinct experiments with S:B.1 used as reference virus in each experiment (indicated as a light grey area). Note that values at time 0 reflect the titres of input virus, while those at 1 hpi are the titres of residual virus in the supernatant after 1 h infection and one wash of the monolayer. **c**, Heat maps of viral titres relative to S:B.1 for all viruses expressed as $\log_2$(fold change) at each indicated timepoint. **d**,**e**, Live virus-based fusion assay in AAT-GFP10/AAT-GFP11 cells. The fusion activity was measured over time and expressed as a percentage of maximum fluorescence over S:B.1; pre-Omicron (**d**) and post-Omicron (**e**) recombinant viruses. For **a**, **b**, **d** and **e**, the light grey

area indicates the data relative to the reference virus S:B.1 in all experiments. Data are represented as mean ± s.e.m. of 3 independent experiments each performed in triplicate. Statistical significance of differences between S:B.1 and the other viruses across timepoints was determined using one-way analysis of variance (ANOVA) with multiple comparisons between area under the curve using the Holm–Šídák post hoc test. For **c**, statistical significance of differences between S:B.1 and the other spike-bearing viruses at each timepoint (heat maps on the right) was instead determined by one-sample $t$-test. *$P \le 0.05$, **$P \le 0.01$, ***$P \le 0.001$, ****$P \le 0.0001$. **f**, Heat maps as percentage of maximum fusion over S:B.1 at indicated timepoints. For **f**, statistical significance of differences between S:B.1 and the other viruses across timepoints was determined using two-way ANOVA with multiple comparisons between area under the curve using the Holm–Šídák post hoc test. The calculated $P$ values are indicated in the figures.

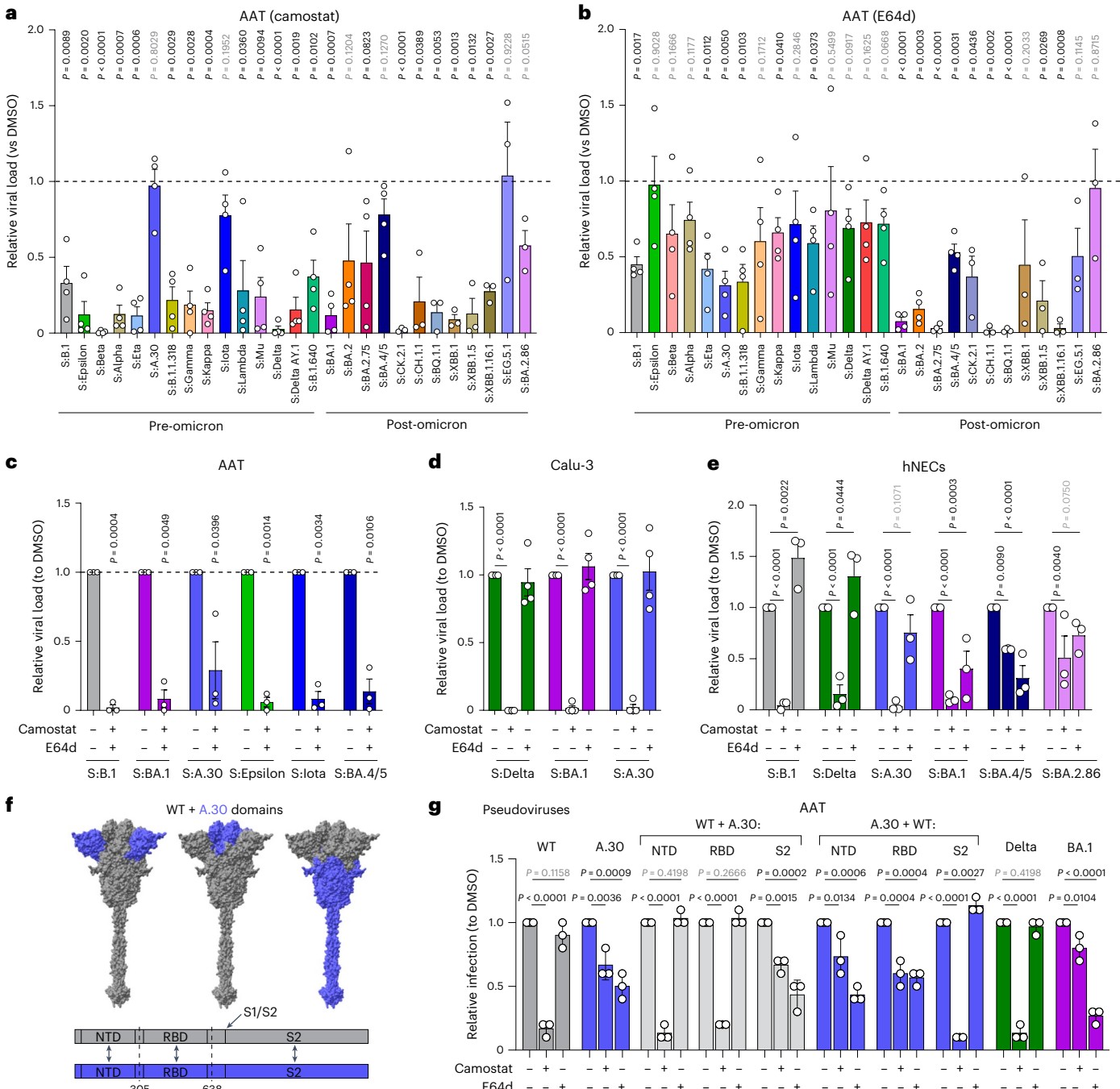

**Fig. 3 | Cell entry pathways of SARS-CoV-2 variants. a–c,** Sensitivity of the panel of SARS-CoV-2 recombinant viruses to 10 μM each of camostat or E64d, either individually (**a**,**b**) or in combination (**c**), in AAT cells. The dotted line represents the relative viral load of DMSO-treated viruses set as 1.0. **d**,**e,** Sensitivity of SARS-CoV-2 viruses to 10 μM each of camostat or E64d in Calu-3 cells (**d**) or hNECs (**e**). **f,** Schematic representation of domain swap constructs between WT (lineage B) and A.30 spike. Full-length spike molecular model[98] is based on PDB:6VXX ref. 5.

**g,** Sensitivity of domain swap pseudoviruses (PV) to 10 μM camostat or E64d in AAT cells. Data are expressed relative to the DMSO-treated control for each virus. In **a–e** and **g,** data are mean ± s.e.m. of at least 3 independent experiments each performed in triplicate (n = 3). Statistical significance of differences between DMSO-treated and inhibitor-treated conditions was determined using one-sample t-test. *P ≤ 0.05, **P ≤ 0.01, ***P ≤ 0.001, ****P ≤ 0.0001.

entry route[32], camostat abrogated infection by S:Delta, S:BA.1 and S:A.30 (Fig. 3d). These results using live virus correlate with the recent observations described for Delta and Omicron[7,32,41,49,50], and further suggest that no variants have completely lost their ability to undergo TMPRSS2-dependent entry. Human nasal epithelium cells (hNECs) express both TMPRSS2 and cathepsin L, and therefore may support both pathways[32]. In hNECs, pre-Omicron recombinants including S:A.30 displayed consistent profiles, being strongly inhibited by camostat

(Fig. 3e) but remaining insensitive to E64d. In contrast and similar to AAT cells, the post-Omicron S:BA.1, S:BA.4/5 and S:BA.2.86 displayed variable sensitivities to both drugs in hNECs (for example, both BA.1 and BA.4/5 exhibited E64d sensitivity).

Collectively, these data suggest that entry pathway preference has complex determinants that are spike and cell-type dependent. More specifically, our data demonstrate cell-type-specific utilization of the endosomal pathway by S:A.30. To identify the viral determinant(s)

involved in S:A.30 entry in AAT, we used lentivirus-based pseudoviruses (PV) incorporating domain-swapped chimaeric spikes possessing the NTD (amino acids (aa) 1–305), RBD (aa 306–637) or S2 (aa 638–Stop) of A.30 in the context of the WT spike, or vice versa (Fig. 3f,g). The PV entry assays were performed in the presence of either camostat or E64d in AAT cells. As expected, the entry of WT (B.1) was strongly inhibited by camostat but not E64d, confirming TMPRSS2-dependent route of virus entry (Fig. 3g). A.30 entry was hindered more by E64d than camostat. Domain swapping A.30 NTD or RBD into WT spike did not affect inhibitor sensitivity, while the entry of PVs bearing WT spike + A.30-S2 phenocopied A.30 (Fig. 3g), suggesting that the S2 portion contains the determinants of A.30 pathway preference. This was confirmed in the reciprocal swap, where A.30 + WT-S2 mirrored the phenotype of the WT pseudovirus.

Entry of pseudoviruses was also assessed in HEK-293T and Calu-3 cells offering primarily endosomal entry or TMPRSS2-dependent entry, respectively (Extended Data Fig. 4a,b). The results highlighted that despite dictating an increased reliance on endosomal route usage for Omicron BA.1 and A.30, mutations in the S2 domain do not necessarily compromise the TMPRSS2-dependent cell entry. Indeed, all PVs were strongly inhibited by E64d in HEK-293T cells but not by camostat (Extended Data Fig. 4a). The reverse was observed in Calu-3, with camostat showing potent inhibition, whereas E64d had negligible effect (Extended Data Fig. 4b), consistent with Fig. 3d. Simultaneous treatment with camostat and E64 d in AATs displayed an additive effect, similar to that seen with live virus (Extended Data Fig. 4c). In accordance with published data, infection with PVs bearing Delta or Omicron BA.1 spike was respectively blocked by camostat or E64d only[32].

## Spike determines nasal cell tropism

Omicron exhibits an increased tropism for the nasal epithelium. We and others previously showed that the SARS-CoV-2 Omicron BA.1 displays a replicative advantage in cells of the upper respiratory tract compared with previously circulating variants[32,51–54]. This change in virus tropism, potentially underscoring enhanced transmissibility, appears to be mediated by the spike protein[51–55]. Here we tested the replication kinetics of a subset of viruses in hNECs by apically infecting hNECs and quantifying the amount of virus released at different timepoints. Strikingly, the data effectively clustered the viruses into two groups (Fig. 4a,b). The replication efficiencies of viruses carrying pre-Omicron spikes were significantly lower at each timepoint compared with post-Omicron viruses. S:A.30 was not associated with any augmented replication in hNECs compared with other pre-Omicron spikes. This correlates with the sensitivity of S:A.30 to protease inhibitors in hNECs (Fig. 3e) and further suggests that it exhibits a pre-Omicron entry phenotype in these cells. As expected, the S:EG.5.1 virus carrying one of the most recently emerged spikes exhibited typical Omicron-like replication kinetics in hNECs, growing at a significantly higher rate than the reference S:B.1 (Fig. 4a,b). Among the post-Omicron group, S:BA.2.86 showed the highest levels of replication in nasal cells (Fig. 4c).

Hence, nasal epithelium tropism is a conserved feature of post-Omicron spike variants. Interestingly, side-by-side comparison of recombinant S:EG.5.1 and S:BA.2.86 with their corresponding WT variants showed that determinants outside of spike also favour replication in both nasal (Fig. 4d) and bronchial epithelium (Extended Data Fig. 5).

We also tested a subset of the panel for their replication in reconstituted human primary bronchial epithelium (hBEC) (Fig. 4e,f). As observed in the nasal epithelium, the post-Omicron recombinants displayed higher replication kinetics than the pre-Omicron ones, supporting data from previous reports[56], but contrasting others[53].

## Virus induced apoptosis and cytopathogenicity of nasal epithelium

We observed that the post-Omicron viruses, when grown in hNECs, detached the apical side of the cultures from the basal layers ('sloughing')

(Fig. 4g,h and Extended Data Fig. 6a) at 4 dpi. Detached cells were spike positive as assessed by RNA in situ hybridization (Fig. 4g). Cultures infected with S:BA.1, compared to the other Omicron-lineage viruses tested (S:BA.2, S:BA.2.75, S:BA.4/5), exhibited the lowest percentage of cell sloughing (Fig. 4h and Extended Data Fig. 6a). To rule out technical artefacts, we measured epithelial slough-off in cells fixed and stained using haematoxylin and eosin (H&E), and obtained essentially the same results (Extended Data Fig. 7a, b). These findings suggest that post-Omicron variants may have a greater capacity to disrupt nasal epithelial cell integrity, potentially enhancing viral transmission through the release of virus particles containing cell debris. In addition, using both cleaved caspase-3 and TUNEL (measuring DNA breaks in the nuclei) assays, we observed elevated levels of apoptosis in cultures infected with viruses bearing spikes of Omicron sublineages compared with the pre-Omicron variants (Fig. 4i,j and Extended Data Fig. 6b–d).

To examine whether reduced apoptosis and slough-off at 4 dpi is a consequence of slower replication of pre-Omicron viruses, we extended the infection duration to 7 days. At this timepoint, we observed comparable levels of epithelial sloughing and cell apoptosis in hNECs infected with either pre-Omicron or post-Omicron spike viruses (Extended Data Fig. 7c–h,j,k). These data suggest that elevated levels of virus replication per se induce apoptosis, as opposed to specific pro-apoptotic pathways activated by the post-Omicron spike proteins.

Next, we compared the levels of type-I interferon (IFN) response in hNECs infected with either the pre- or post-Omicron viruses. Overall, there was a clear correlation in hNECs between the levels of virus replication kinetics and the activation of type-I IFN response (Extended Data Figs. 6g,h, 7 and 8).

## Spike amino acid residues determining nasal epithelial cell tropism

Our data indicate that nasal tropism is the unifying and discriminatory feature of the post-Omicron variants. However, the specific molecular determinants contributing to this phenotype remain uncharacterized. We focused on 13 amino acid residues dispersed between RBD, RBM and the S2 subunit, which are unique to Omicron BA.1 and conserved in the Omicron sublineages[3,16,32,45,57]. We modified the S:BA.1 and S:B.1 viruses by substituting specific amino acids within spike; the residues in S:BA.1 were reverted back to corresponding residues in S:B.1, while the reciprocal changes to BA.1 were introduced in the S:B.1 spike (Fig. 5a,b). Thus, we constructed eight mutant viruses: (1) S:BA.1+WT-13 virus has all the 13 unique Omicron residues reverted to WT B.1; (2) S:BA.1+WTRBD-4 and (3) S:BA.1+WTRBM-4 possess only the B.1 spike mutations in the RBD (excluding the RBM, 4 aa residues) or RBM (4 aa residues), respectively; (4) S:BA.1+WTS2-5 possesses the 5 B.1 mutations in the S2 region. The reciprocal swap mutants are (5) S:B.1+BA.1-13, (6) S:B.1+BA.1RBD-4, (7) S:B.1+BA.1RBM-4 and (8) S:B.1+BA.1S2-5.

S:BA.1+WT-13 displayed a loss of fitness in hNECs, replicating even less efficiently than S:B.1 (Fig. 5c and Extended Data Fig. 9a). S:BA.1+WTRBM-4 and S:BA.1+WTS2-5 showed a similar reduction in fitness as S:BA.1+WT-13. Remarkably, S:BA.1+WTRBD-4 replication in hNECs was completely abolished (Fig. 5c and Extended Data Fig. 9a). All reciprocal mutants in the B.1 spike background displayed slower replication kinetics than S:B.1 in hNECs at the early timepoints post infection (16–24 h post infection (hpi)) (Fig. 5d). To control for any potential global loss of fitness, the mutants were also assessed in Calu-3. In this cell line, most of the viruses displayed similar replication kinetics to their respective parental spike backbone (Fig. 5e,f). However, S:BA.1+WTRBD-4 replicated as robustly as S:B.1 in Calu-3, while S:B1+BA.1-13 and S:B1+BA.1RBD-4 had an intermediate replicative fitness.

Overall, these data suggest a critical role for spike RBD residues D339, F371, P373 and F375 for the enhanced viral replication of the post-Omicron variants in hNECs. Interestingly, another study showed that mutations in the serine residues 371, 373 and 375 in B.1, both alone

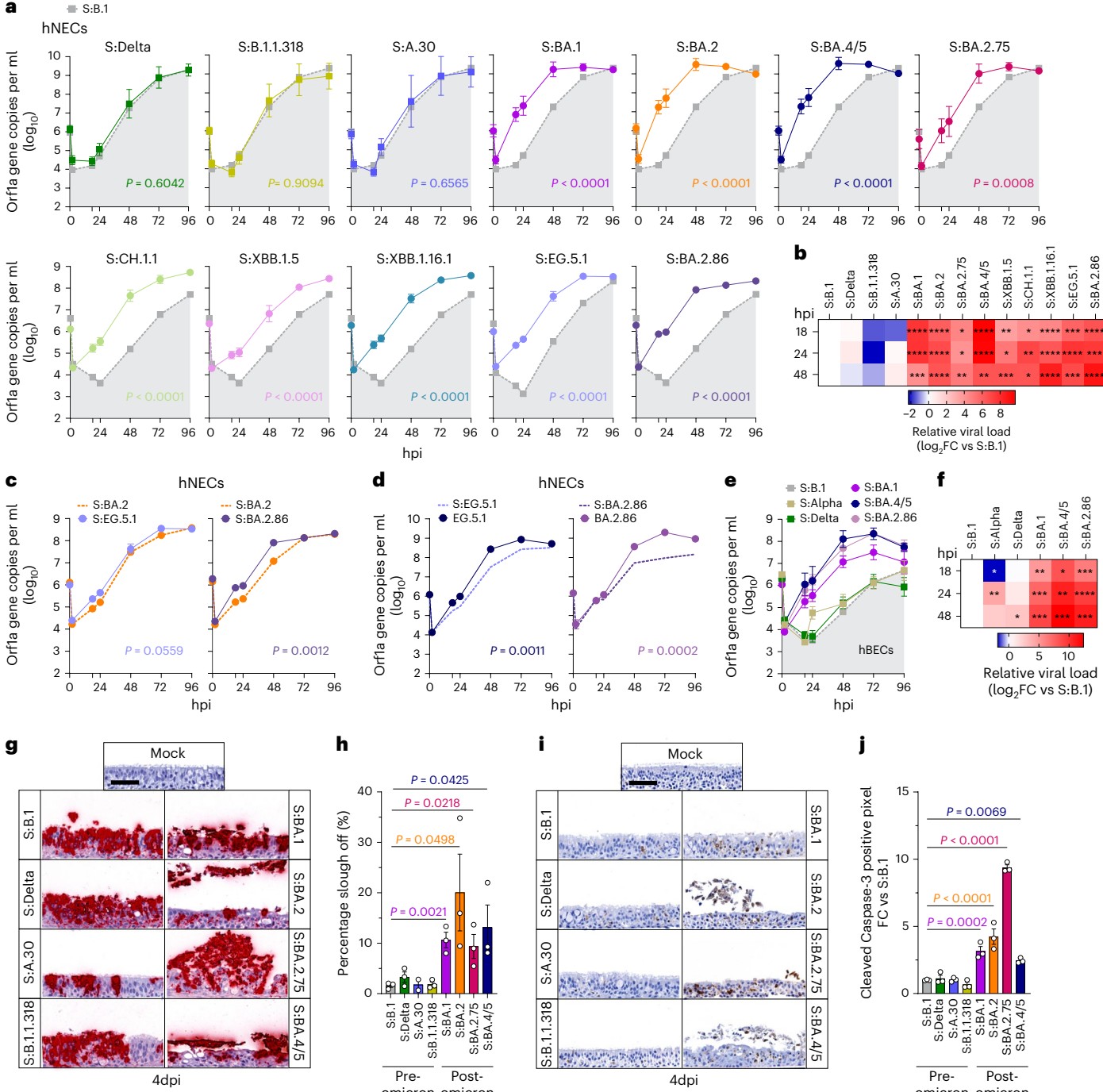

**Fig. 4 | Replication kinetics of SARS-CoV-2 spike recombinants in nasal epithelium. a**, Replication kinetics of SARS-CoV-2 viruses in hNECs. For **a** and **e**, the light grey area indicates the replication curve of S:B.1 used as reference virus in all experiments. **b**, Heat map of relative viral loads to S:B.1 for all SARS-CoV-2 recombinant viruses expressed as log$_2$(fold change) at each indicated timepoint. **c**, Replication kinetics of S:EG.5.1 and S:BA.2.86 compared to the parental S:BA.2 (dashed line). **d**, Replication kinetics of S:EG.5.1 and S:BA.2.86 compared to their corresponding wild-type clinical isolates (dashed line). **e**, Replication kinetics of pre-Omicron and post-Omicron spike recombinant viruses in reconstituted hBEC. For **a**, **c**, **d** and **e**, the copy numbers of viral RNA in the culture supernatant of infected hNECs were quantified by RT–qPCR at the indicated timepoints. Data are mean ± s.e.m.; *n* = 6. Statistical significance of differences between S:B.1 (**a**), S:BA.2 (**c**) or clinical isolates (**d**) and the spike recombinant viruses across timepoints was determined using one-way ANOVA with multiple comparisons between area under the curve of the different recombinant viruses

using the Holm–Šídák post hoc test. The calculated *P* values are indicated in the figures. **f**, Heat map of relative viral loads to S:B.1 for all recombinant viruses expressed as log$_2$(fold change) at each indicated timepoint. For **b** and **f**, statistical significance of differences between S:B.1 and the other spike-bearing viruses at each timepoint was determined using one-sample *t*-test. *$P \le 0.05$, **$P \le 0.01$, ***$P \le 0.001$, ****$P \le 0.0001$. **g**, Photomicrographs showing RNA in situ hybridization of SARS-CoV-2 spike in hNECs infected with the indicated viruses. Sloughing of the top layers of the epithelium is evident at 4 dpi in cultures infected with post-Omicron viruses. **h**, Quantification of cell sloughing (*n* = 3) in hNECs using whole-section software-assisted imaging. **i**, Photomicrographs of cleaved caspase-3 detected in hNECs infected with the indicated viruses at 4 dpi. Apoptotic cells with cleaved caspase-3 are stained in brown. **j**, Quantification of caspase-3 cleavage in hNECs (*n* = 3). For **h** and **j**, statistical significance was determined using one-way ANOVA with multiple comparisons between S:B.1 and other viruses using the Holm–Šídák post hoc test.

or in combination, affect infectivity and fusogenicity of VSV (vesicular stomatitis virus)-based pseudoviruses in the CaCo-2 cell line[58]. Importantly, the same mutations have also been reported to contribute to immune evasion[59–61]. Hence, these Omicron-specific RBD mutations profoundly affect spike biology (Extended Data Fig. 9).

We also assessed this panel of mutants for their cell–cell fusion activity. We found that substitution of all 13 amino acids in either the BA.1 or B.1 spike was sufficient to completely switch the fusion phenotype (Fig. 5g,h). Importantly, in the S:BA.1 context, substitution of only the S2-5 amino acid residues restored the fusion activity of the pre-Omicron variants. These residues include K764, Y796, H954 and K969 distributed across the core fusion machinery of S2, and K679 juxtaposed to the S1/S2 cleavage site (Fig. 5a,b). Therefore, reverting these sites may enhance proteolytic activation and/or fusion efficiency. In the S:B1 context instead, substitution of either just the BA.1 RBD-4 residues or the BA.1 S2-5 residues reduced fusion significantly (Fig. 5h). Notably, introduction of the BA.1 RBM residues S477N, E484A, Q498R and Y505H in the context of the S:B.1 resulted in an increase in fusion activity compared with WT S:B.1.

Finally, we also evaluated the sensitivity of these viruses to protease inhibitors in AAT cells. None of the BA.1 spike mutants significantly altered the sensitivity to camostat or E64d of the parental S:BA.1 (Extended data Fig. 10a). In contrast, we observed more diverse phenotypes among the S:B.1 mutants. Mutants S:B.1+BA.1RBM-4 and S:B.1+BA.1S2-5 displayed a similar phenotype to the parental S:B.1 virus (Extended data Fig. 10b). However, S:B.1+BA.1RBD-4 containing substitutions G339H, S371F, S373P and S375F significantly increased sensitivity to E64d compared with S:B.1, whereas substitution of all 13 residues resulted in a relative decrease in camostat sensitivity. Hence, the overall structural context of these amino acid residues is critical in determining the entry route preference.

## Spike-driven virulence

The emergence of Omicron and its sublineages was characterized by a milder clinical outcome in infected individuals compared with those infected with the early B.1 and other pre-Omicron variants (for example, the highly virulent Delta)[29,34,62]. To investigate the effects of spike evolution on virus virulence, we infected Golden Syrian hamsters with S:B.1 (as reference strain) and with a selection of pre-Omicron and post-Omicron recombinant viruses.

The mock-infected animals gained ~5% of body weight, while those infected with the reference S:B.1 virus, as expected, lost ~10% of their body weight by 6 dpi (Fig. 6a). Weight loss profiles of animals infected with the pre-Omicron variants were similar, while the profile of the post-Omicron-infected group was more variable (Fig. 6a).

Weight loss is an indirect measurement of disease severity, as it can be influenced by a variety of environmental factors. We therefore assessed the virulence of SARS-CoV-2 variants by quantifying the extent of pulmonary pathology in infected animals using software-assisted whole-lung-section imaging and downstream analyses by machine learning ('digital pathology'), as recently described[33]. In particular, we quantified pulmonary macrophages and T cells as indirect evidence of lung inflammation, as well as alveolar epithelial hyperplasia, a direct result of alveolar injury[33,63]. The lungs of animals infected with the pre-Omicron viruses displayed comparable levels of macrophages and T cells infiltrating the lungs, in addition to alveolar epithelial hyperplasia (Fig. 6b–e). Animals infected with S:Delta represented a notable exception, as they presented with significantly fewer macrophages and reduced alveolar hyperplasia (Fig. 6b–e). These data were somewhat surprising, given the enhanced virulence of Delta relative to B.1 shown here and elsewhere[29,30,64].

Post-Omicron viruses showed a more variable phenotype. Overall, S:BA.4/5 and S:EG.5.1 were found to be more virulent than S:BA.1, S:BA.2.75 and S:BA.2.86, but less virulent than S:B.1. While CD3+ cells were comparable across the various infected groups, hamsters infected with S:EG.5.1 and S:BA.4/5 displayed higher levels of lung macrophages and alveolar hyperplasia than hamsters infected with S:BA.1 (Fig. 6b–e). Of note, the viral loads detected in nasopharyngeal swabs collected at 24 hpi was similar between all viruses and the reference S:B.1 (Fig. 6f).

Taken together, these results indicate that the determinants of SARS-CoV-2 virulence are both within and outside of spike. The reduced virulence of S:Delta compared with S:B.1 indicates that the increased virulence of the Delta VOC compared with B.1 is largely conferred by determinants outside of spike. In contrast, the reduced virulence of the Omicron sublineages is determined predominantly by spike.

## Discussion

Our data identify distinct phenotypic trajectories of SARS-CoV-2 spike during the pre- and post-Omicron phases of the pandemic. Pre-Omicron spikes exhibited similar phenotypes with subtle differences underlying host adaptation. A notable exception was the spike of A.30, a lineage A variant distinct from the B lineage responsible for the global pandemic. S:A.30 exhibited preference for the endosomal entry pathway in AAT cells, as opposed to the cell membrane route characteristic of the pre-Omicron variants[2,7,65,66]. While we did not observe this phenotype in primary nasal cells, our data suggest that the capacity to use the endosomal entry route appeared at least once before the emergence of Omicron. Importantly, we show the spike S2 domain to be the key determinant of this altered entry pathway[32].

Omicron emergence represented a major phenotypic 'reset' to the pandemic. This variant evolved to be immunoevasive and displayed enhanced tropism for the nasal epithelium, but was poorly fusogenic[29,32,34,42,54,56,62,67–70]. The evolutionary trajectory of spike of post-Omicron variants explored a wider variety of phenotypes. We showed that the nasal epithelium tropism, conserved in all 13 of the post-Omicron viruses tested, is determined by spike (as previously reported[60,71,72]). A recent study published during the preparation of this paper showed that Omicron tropism is determined by enhanced spike binding to the nasal epithelium, in addition to escape from specific antiviral innate immune effectors (although there are contrasting

**Fig. 5 | Determinants of SARS-CoV-2 nasal tropism. a**, Ribbon diagram of the molecular structure of spike (based on PDB:6VSB, one RBD 'up' conformation[98–100]) illustrating the location of the stated residues, colour-coded as shown in the key. The approximate location of the viral membrane is shown as a dashed line. **b**, Schematic representation of the spike recombinant mutants used in this study. S:BA.1 spike amino acid residues in the RBD, RBM, S1/S2 junction and S2 domain have been reverted to the corresponding S:B.1 residues (top) or inserted in the S:B.1 backbone (bottom) in combination or individually as indicated. **c,d**, Replication kinetics of the SARS-CoV-2 S:BA.1 (**c**) and S:B.1 wt (**d**) spike mutants in hNECs. S:B.1 (indicated by light grey area) and S:BA.1 (dashed purple line) viruses were used as reference viruses. The copy numbers of viral RNA in the culture supernatant of infected hNECs were quantified by RT–qPCR at the indicated timepoints. Data are mean ± s.e.m. of 3 biological replicates. **e,f**, Replication kinetics of the SARS-CoV-2 S:BA.1 (**e**) and S:B.1 wt (**f**) spike mutants in Calu-3 cells. The copy numbers of viral RNA in the culture supernatant of infected Calu-3 cells were quantified by RT–qPCR at the indicated timepoints. **g,h**, Live virus-based fusion assay in AAT-GFP10/AAT-GFP11 cells. The fusion activity of SARS-CoV-2 S:BA.1 (**g**) and S:B.1 wt (**h**) spike mutants was measured over time and expressed as a percentage of maximum fluorescence over S:B.1. The light grey area indicates S:B.1, while the purple dashed line represents S:BA.1; both recombinant viruses were used as reference viruses in all experiments. In **e–h**, data are mean ± s.e.m. of 3 independent experiments each performed in triplicate (n = 3). For **c–h**, statistical significance of differences between S:B.1 and the mutant spike virus panel across timepoints was determined using one-way ANOVA with multiple comparisons between area under the curve of the different viruses and S:B.1 using the Holm–Šídák post hoc test. Mutants were also compared to S:B.1+WT-13 (**c**) or S:B.1+BA.1-13 (**d**). The calculated P values are indicated in the figure.

data with regard to the latter)[54,73]. We identified BA.1 spike RBD residues D339, F371, P373 and F375 (conserved in all post-Omicron variants) that when mutated to the corresponding residues in B.1 (G339, S371, S373 and S375) abrogated nasal tropism, strikingly without affecting

replication in Calu-3 cells. Interestingly, Omicron spike S375F was previously highlighted as a key change associated with increased virus transmissibility[71]. Moreover, a recent study proposes that mutations at these same residues may only be adaptive when they co-occur

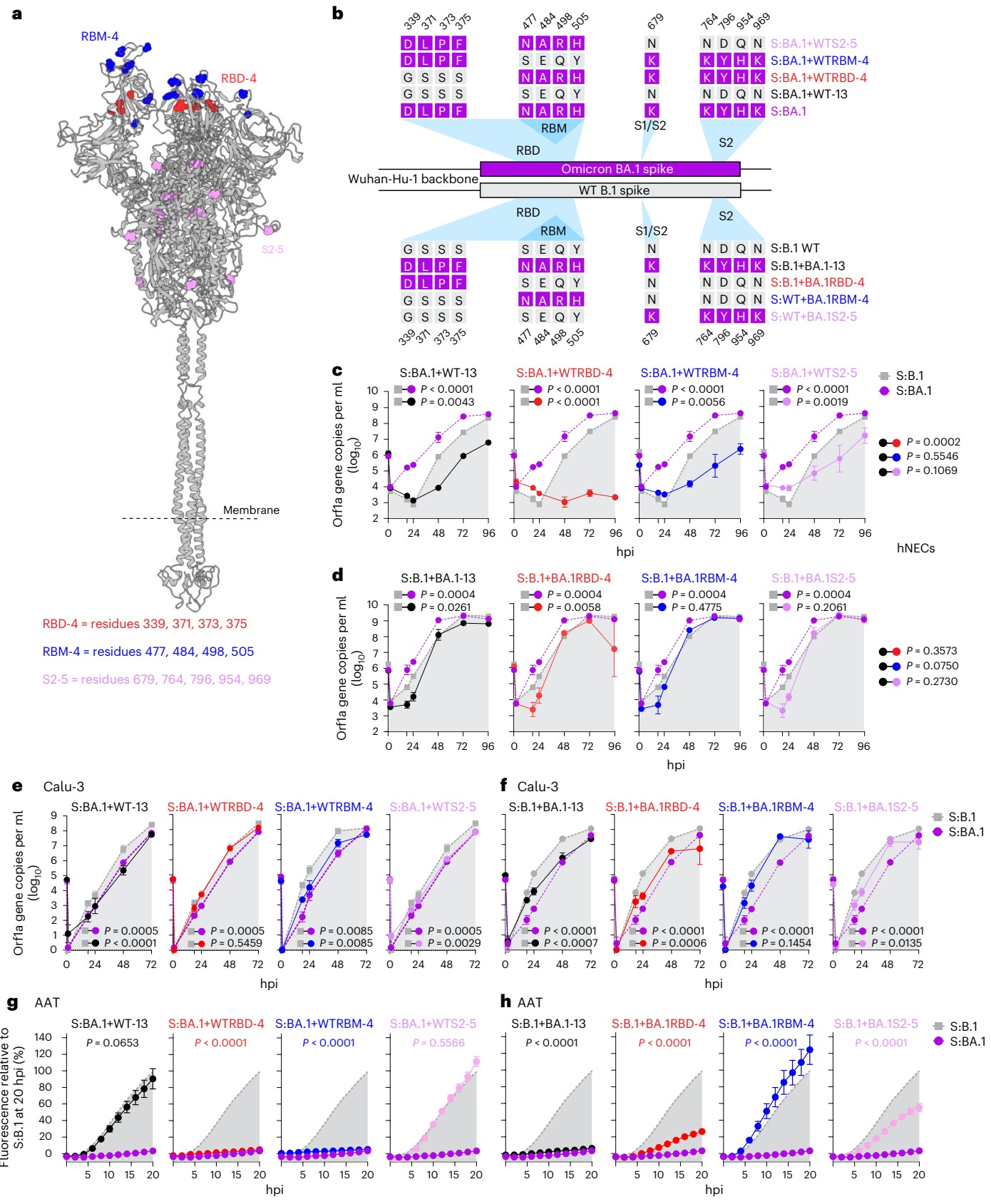

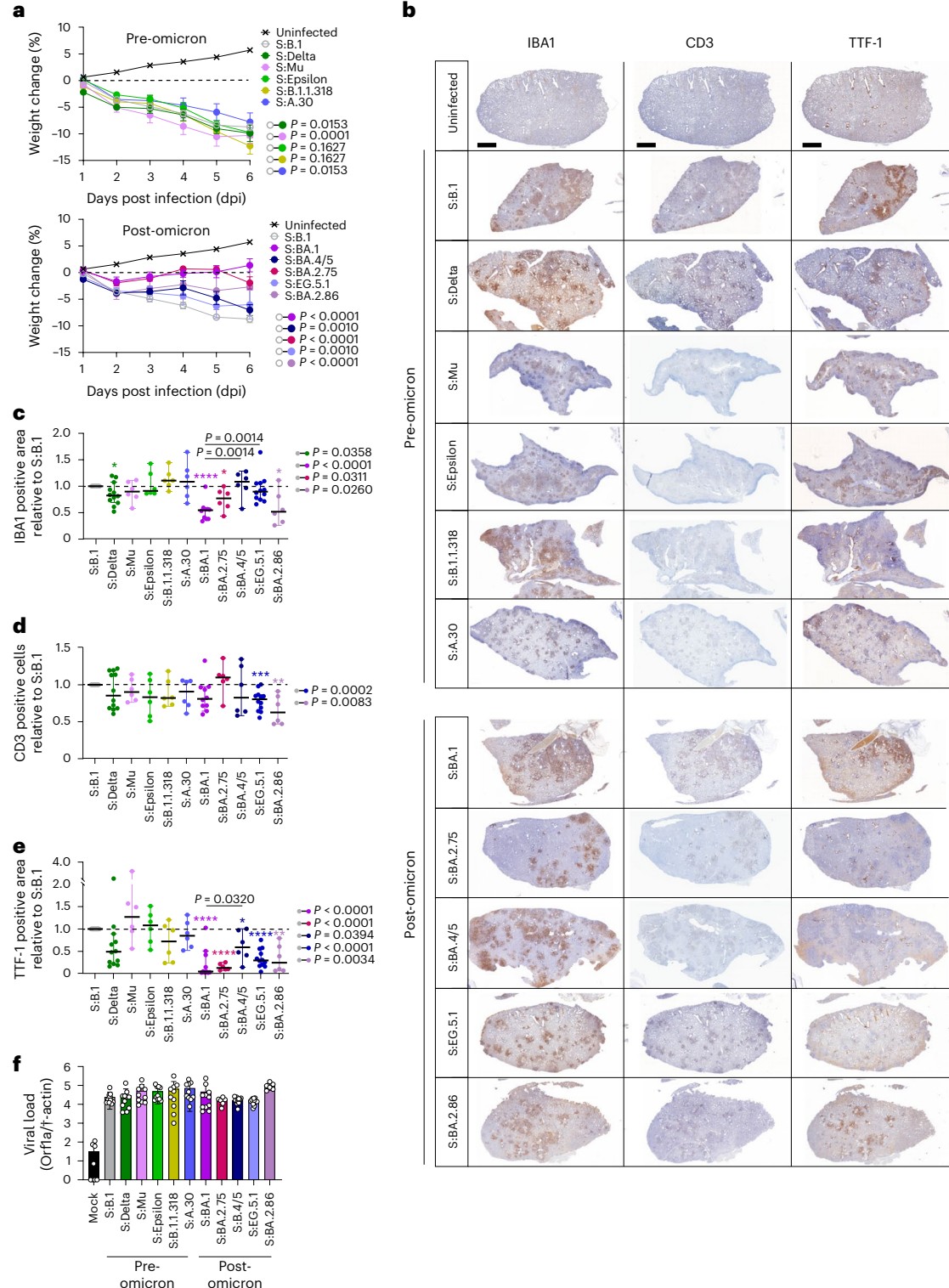

**Fig. 6 | Virulence of SARS-CoV-2 recombinant viruses in experimentally infected hamsters.** Syrian hamsters were intranasally inoculated with the indicated recombinant viruses ($3.75 \times 10^6$ genome copies per animal) or mock infected and killed at 6 dpi. **a**, Daily body weight changes of infected and uninfected hamsters. Data are median ± 95% confidence interval. Each group is $n \geq 6$ animals. Weight comparisons were performed using two-way ANOVA and the Holm–Šídák post hoc test. The calculated $P$ values are indicated in the figures. **b**, Representative photomicrographs of one lung lobe section stained using immunohistochemistry with antibodies for IBA1, CD3 and TTF-1. Scale bar, 2 mm. **c**–**e**, Quantification of each marker for experiments shown in **b** using whole-section software-assisted imaging. Data are median ± 95% CI and

expressed as relative quantification of markers detected in hamsters infected with S:B.1 (set as 1.0 for each marker). **f**, Viral load in nasopharyngeal swabs at 24 hpi in each infected animal. The copy numbers of viral RNA in the resuspended swabs collected from infected hamsters were quantified by RT–qPCR. Data are mean ± s.e.m. of $n \geq 6$ animals. In **c**–**e**, statistical significance of differences between S:B.1 and other spike SARS-CoV-2 was determined using one-sample $t$-test, while statistical significance of differences between S:BA.1 and other post-Omicron recombinant viruses was determined using one-way ANOVA with multiple comparisons between conditions using the Holm–Šídák post hoc test. *$P \leq 0.05$, **$P \leq 0.01$, ***$P \leq 0.001$, ****$P \leq 0.0001$, and exact $P$ values are indicated in figures.

with the other unique substitutions of Omicron spike[57]. Indeed, our in vitro data (Fig. 5) leads us to the same conclusions; reversion of the RBD-4 mutations in isolation completely abolished S:BA.1 replication in hNECs, whereas the same reversions in the context of the S:BA.1+WT-13 virus retained intermediate fitness. We speculate that these complex relationships may reflect the co-evolving networks of molecular interactions that govern spike conformation and its ability to mediate productive entry within a given cellular context.

We also show that spike tropism for the nasal epithelium determines virus-induced apoptosis and sloughing of superficial layers of the epithelium. This process may facilitate spread of post-Omicron variants. Of note, although spike is the key determinant of nasal tropism, our data with more recently emerged Omicron sublineages (EG.5.1 and BA.2.86) indicate that other viral proteins may also contribute to this phenotype. Hence, post-Omicron variants are progressively adapting to the 'niche' of the nasal epithelium.

While tropism for the nasal epithelium was the common feature of post-Omicron spikes, other phenotypes were far more variable. We observed higher replication and fusogenicity in human lung-derived cell lines for viruses carrying spikes from variants that emerged towards the end of the post-Omicron phase. Notably, we demonstrate that reversion of the S2-5 mutations in BA.1 rescues fusogenicity; however, these sites remain unaltered in the fusion-competent Omicron variants (S:BA.2.86 and S:EG.5.1). This suggests that there are multiple genetic pathways to regain fusogenicity. It is important to note that SARS-CoV-2 spike adaptation to human cells seems to mirror the evolution described for proteins outside of spike (for example, Orf6, N, Orf9b) which favoured an enhanced innate immune evasion in both pre-Omicron and post-Omicron variants[74–77].

Another variable phenotype observed with the post-Omicron viruses was the usage of activating protease. Early post-Omicron viruses were mostly sensitive to E64d, indicating reliance on endosomal cathepsins, as previously shown[32,36,41,42]. However, some of the post-Omicron viruses were relatively resistant to E64d (S:BA.4/5, S:CK.2.1, S:XBB.1, S:EG.5.1 and S:BA.2.86). S:CK.2.1 and S:XBB.1 were sensitive to camostat, suggesting that they may undergo TMPRSS2-dependent entry, while S:BA.4/5, S:EG.5.1 and S:BA.2.86 were relatively insensitive to camostat, potentially suggesting a flexible use of entry pathway depending on cellular context.

We also assessed how spike variation affects the intrinsic virulence of SARS-CoV-2 variants in hamsters. In general, post-Omicron recombinants were less virulent than pre-Omicron ones. Among the post-Omicron group, S:BA.4/5, and to a lesser extent S:EG.5.1, were most virulent, complementing previous studies with WT viruses[33,36,44]. Interestingly, the levels of macrophage and T-cell infiltration in hamsters infected with S:BA.4/5 and S:EG.5.1 were comparable to those induced in animals infected with S:B.1 and other pre-Omicron viruses. However, all the post-Omicron viruses induced lower levels of alveolar epithelial hyperplasia than S:B.1 and the other pre-Omicron counterparts. S:BA.4/5 induced significantly higher levels of alveolar epithelial hyperproliferation compared with S:BA.1.

The approach we used to quantify virulence offers different insights. Alveolar epithelial hyperplasia is due to proliferation of type-2 pneumocytes (or other lung progenitor cells) in response to damage or loss of type-1 pneumocytes, while increased levels of macrophages and T cells in the lungs are in response to virus-induced inflammation/immune-activation[72,78–80]. Hence, we can deduce that lung inflammation (and therefore clinical disease) can also be induced by virus infection in the bronchi, while alveolar epithelial hyperplasia can only occur because of damage to type-1 pneumocytes. These data therefore support the notion that post-Omicron variants have a tropism for the upper respiratory tract but not the lung alveoli. In contrast, S:BA.4/5 (and S:EG.5.1) induces more alveolar hyperplasia than S:BA.1, suggesting that these spikes have regained some propensity to replicate in the deep respiratory tract, as shown before with WT variants[33,36,44].

The systematic approach of this study enabled us to disentangle correlations between phenotypes previously proposed by comparing a smaller number of variants. First, our data do not support a model where nasal tropism is directly related to endosomal entry pathway, as previously suggested[32,41,49]. Second, preference of cell entry pathway in lung cell lines is not correlated with virulence[32,41,81]. S:A.30 is as virulent as S:B.1, and yet it displays a different cell entry usage in AAT cells. Finally, spike fusogenicity does not correlate with virulence as originally proposed[30,41,42,81]. S:BA.2.86 and S:EG.5.1 are as fusogenic as S:B.1 but are not as virulent. Conversely, S:BA.4/5 is not fusogenic (similarly to S:BA.1), but it is significantly more virulent than S:BA.1.

A limitation of our study is the difficulty in assessing our recombinant viruses in all the different assays, cells and experimental platforms explored in the extensive literature on SARS-CoV-2 variants. For example, recent studies suggest that membrane-type matrix metalloproteinase (MT-MMP) and a disintegrin and metalloproteinase (ADAM) families can mediate SARS-CoV-2 entry[82,83]. Omicron has been suggested to rely on metalloproteinases for membrane fusion[54] in primary human nasal epithelium. Hence, this class of proteases need to be investigated further. Of note, in this study S:BA.1 replication in primary nasal cells was very sensitive to camostat, while in a previous study, full-length BA.1 was found to be insensitive to both camostat and E64d[54].

In conclusion, the spillover of ancestral spike in the human population was followed by a period of adaptation to the host of the pre-Omicron variants. Omicron 'reset' several phenotypes, exploring broader phenotypic variability with the post-Omicron variants, converging on many of those possessed by pre-Omicron spikes. This adaptation process is ongoing and demonstrates the large extent of phenotypic plasticity possessed by SARS-CoV-2 variants, which may have repercussions on the impact of COVID-19 as it transitions to its endemic phase.

## Methods

### Ethics statement
Procedures involving animals were performed under UK Home Office Licence PP0271643 in accordance with the Animals (Scientific Procedures) Act 1986 and approved by the University of Glasgow Ethics Committee. All animal research adhered to ARRIVE guidelines[84].

### Biosafety measures
All work with SARS-CoV-2, including recombinant and mutant viruses, was carried out at containment level 3 (CL3), after approval by the University of Glasgow GM223 Safety Committee, and after clearance was received from the UK Health and Safety Executive (HSE) under notification GM223/20.1a.

### Cells and viruses
Baby hamster kidney cells clone 21 (BHK-21 ATCC CCL-10) overexpressing human ACE2 receptor and SARS-CoV-2 N protein (BHK-N-ACE2) were used for rescue of reverse genetics (RG)-derived SARS-CoV-2 (ref. 32). Vero E6 cells (generous gift of Michele Bouloy) were used to propagate the RG-derived viruses. Calu-3 cells (ATCC, HTB-55) are human lung adenocarcinoma epithelial cells. A549 cells (ATCC, CCL-185) overexpressing human ACE2 receptor and TMPRSS2 serine protease (AAT) were generated in house as described previously[32]. HEK-293T cells were used to produce HEK-293T-ACE2 cells by stable transduction with pSCRPSY-hACE2 and were maintained in complete Dulbecco's modified Eagle medium (Gibco, 31966021) supplemented with 2 µg ml⁻¹ puromycin (InvivoGen, ant-pr-1). HEK-293T cells were used for the generation of HIV(SARS-CoV-2) pseudoviruses. AAT-GFP1-10 and AAT-GFP11 were described previously and were used to perform membrane fusion assays[32]. All cell lines were maintained at 37 °C and 5% $CO_2$ in DMEM supplemented with 10% fetal bovine serum (FBS, Gibco, 2538812H), except for Calu-3 cells which were maintained in RPMI Medium 1640 (Gibco, 11875-093) supplemented with 20% FBS. Human reconstituted nasal

and bronchial epithelium cultures (MucilAir and SmallAir, abbreviated hNEC and hBEC in this article) were purchased from Epithelix and maintained in MucilAir or SmallAir complete culture medium (Epithelix, EP04MM and EP64SA, respectively) at an air–liquid interface[32,41,85,86]. The Alpha (B.1.1.7), Beta (B.1.351) and Delta (B.1.617.2) VOCs and EG.5.1 isolates were kindly provided by Wendy Barclay (GISAID accession numbers EPI_ISL_723001, EPI_ISL_770441 and EPI_ISL_1731019, respectively)[40,52,87]. The Omicron B.1.1.529.1 (BA.1) VOC was described previously[32]. The Gamma (P.1) variant was obtained by reverse genetics as part of this study[32]. The BA.4 (B.1.1.529.4) and BA.5 (B.1.1.529.5) Omicron sublineage isolates were kindly provided by Greg Towers[74]. The BA.2.86 isolate was kindly provided by Alex Sigal[88].

## Generation of recombinant viruses using reverse genetics

Recombinant viruses reported in this study were generated using transformation-associated recombination (TAR) in yeast (*Saccharomyces cerevisiae*) as previously described[52,74,76,89,90]. Briefly, recombinant complementary (c)DNA genomes under the control of the bacteriophage T7 RNA polymerase or human cytomegalovirus promoter were assembled using a set of overlapping ancestral SARS-CoV-2 lineage B genomic cDNA fragments incorporating sequences encoding the spike protein of different variants or mutated derivatives into *S. cerevisiae-E. coli* shuttle bacterial artificial chromosome vectors following TAR in yeast. The T7 promoter-carrying yeast artificial chromosome vectors isolated from individual yeast clones were then used as templates to in vitro synthesize viral genomic RNA transcripts, which in turn were used for infectious virus rescue through transfection into BHK-N-Ace2 cells. The yeast-derived cytomegalovirus promoter-carrying bacterial artificial chromosome vectors were purified following transformation into *E. coli* and then used directly to transfect into BHK-N-Ace2 or AAT cells for infectious virus rescue. We named recombinant viruses on the basis of spike origin; for example, those encoding Alpha or BA.1 spike were termed 'S:Alpha' or 'S:BA.1', respectively.

## Virus propagation and titration

Working virus stocks were prepared and titrated as previously described. Briefly, between 50–100 µl of the seed virus was inoculated into Vero E6 cells ($5 \times 10^6$ cells in a T-75 flask). After 1 h incubation, the culture medium was replaced with DMEM containing 2% FBS and 1% penicillin–streptomycin. Upon observation of cytopathic effect (CPE), usually between 4 dpi and 6 dpi, the culture supernatant was collected, clarified by centrifugation at $500 \times g$ and stored at −80 °C. The virus stock titre was measured using quantitative PCR with reverse transcription (RT–qPCR) and expressed as Orf1a gene copy number equivalent genome per ml of supernatant.

Either two clones or two independent working stocks from one clone of each rescued virus were passaged in Vero E6 cells to produce P1 stocks for use in experiments described herein. Viral RNA was extracted using a QIAamp viral RNA mini kit (Qiagen, 52906) and viral genome sequenced using Oxford Nanopore as described below[91].

## Viral whole-genome sequencing

Sequencing library preparation was performed using the ARTIC Network nCoV-2019 sequencing protocol (v.3)[91]. RNA extracts were reverse transcribed using LunaScript RT (NEB, E3010), followed by multiplex PCR using ARTIC nCoV-2019 primer (v.3 and v.4) and Q5 HotStart master mix (NEB, M0494) to produce a tilled array of 300-bp amplicons. Amplicons were then subjected to end-repair and A-tailing using the NEBNext Ultra II End-Repair and dA-Tailing module (NEB, E7546), followed by ligation of barcoded adapters (ONT EXP-NBD196 or SQK-NBD114.96) using Blunt/TA ligase (NEB, M0367). After addition of the barcodes, samples were pooled before ligation of the sequencing adapter (ONT, EXP-NBD196 or SQK-NBD114.96) using T4 Quick ligase (NEB, E6056) and sequencing on a GridION nanopore sequencer using FLO-MIN106 or FLO-MIN114 flow cells. Viral genome sequences were created for each

barcoded sample using the ARTIC nCov2019 bioinformatics environment (https://github.com/artic-network/artic-ncov2019).

## Virus replication assays

Multicycle replication assays were carried out in different cell lines by equalizing virus input, measuring Orf1a gene copies by RT–qPCR as described below. This method was also used to assess viral replication kinetics as indicated below. This method, as opposed to classic plaque-forming unit/endpoint dilution analysis is necessary as both we and others have found that the different SARS-CoV-2 variants replicate with different kinetics in different cells lines (including those cells normally used to titrate viruses such as VERO and BHK) and use different mechanisms of cell entry[54,74,75]. Hence, our approach normalizes virus titres independently of variant-specific differences in cell tropism or entry routes which have an impact on plaque assays or 50% tissue culture infectious dose measurements. Calu-3 cells ($1.8 \times 10^5$ per well) were seeded in 24-well plates and infected 5 days later with the equivalent of $10^4$ Orf1a genome copies per well in serum-free RPMI-1640 medium. After 1 h of incubation at 37 °C, cells were washed and replaced with fresh 20% FBS RPMI-1640 medium. Supernatant was collected at the indicated timepoints and viral RNA extracted and quantified as described. Primary hNECs were washed once with serum-free DMEM before infection at the apical side with $10^5$ Orf1a equivalent genomes per well in serum-free medium and incubated at 37 °C/5% $CO_2$ for 2 h. Inoculum was then removed and cells washed once with serum-free DMEM, followed by the first timepoint collection. At the indicated timepoints, virus released from the cells was collected 15 min after incubation of 100 µl of pre-warmed serum-free DMEM to the apical side of the cultures. Supernatant was then added to LBF lysis buffer (Beckman Coulter, A35604) and heat inactivated for 20 min at 56 °C.

## RNA extraction and RT–qPCR

Viral RNA was extracted from culture supernatants using the RNAdvance blood kit (Beckman Coulter, A35604) following manufacturer recommendations. RNA was used as template to detect and quantify viral genomes by duplex RT–qPCR using a Luna Universal Probe one-step RT–qPCR kit (New England Biolabs, E3006E). SARS-CoV-2-specific RNAs were detected by targeting the ORF1a gene using the following set of primers and probes: SARS-CoV-2_Orf1a_Forward 5′-GACATAGAAGTTACTGGCGATAG-3′, SARS-CoV-2_Orf1a_Reverse 5′-TTAATATGACGCGCACTACAG-3′, SARS-CoV-2_Orf1a_Probe ACC-CCGTGACCTTGGTGCTTGT with HEX/ZEN/3IABkFQ modifications. SARS-CoV-2 RNA was used to generate a standard curve, and viral genomes were quantified and expressed as number of Orf1a RNA molecules per ml of supernatant. All runs were performed on an ABI7500 Fast instrument and results analysed with the 7500 Software v.2.3 (Applied Biosystems, Life Technologies).

## Virus particle purification

Calu-3 cells were infected as previously described for the viral growth curves. At 48 hpi, released viral particles were purified from the supernatant using the Dynabeads Intact Virus Enrichment kit (Invitrogen, 10700D) following manufacturer instructions. Briefly, 20 µl of magnetic beads were washed in Binding&Washing buffer (B&W buffer: 10 mM NaCl in 20 mM triethanolamine, pH 6.0) and then incubated with 0.5 ml of cell supernatant for 10 min. After two washes with B&W buffer, samples were processed for immunoblotting as described below.

## Immunoblotting

Cells and purified virus particles on beads were lysed in ice-cold lysis buffer containing 50 mM Tris/HCl pH 7.5, 1 mM EGTA, 1 mM EDTA, 1% (v/v) Triton X-100, 1 mM sodium orthovanadate, 50 mM sodium fluoride, 5 mM sodium pyrophosphate, 0.27 M sucrose, 10 mM sodium 2-glycerophosphate, 1 mM phenylmethylsulphonyl fluoride, 1 mM benzamidine and 1× NuPAGE LDS sample buffer (NP0007, ThermoFisher).

Samples were then analysed by SDS–PAGE and immunoblotting using rabbit anti-SARS-CoV-2 spike S2 domain (ThermoFisher, PA1-41165) and according to the assay, rabbit anti-phospho STAT1 Tyr701 clone 58D6 (Cell Signaling Technologies, 9167), rabbit anti-RSAD2 (Proteintech, 28089-1-AP), mouse anti-IFIT-1 (Origene, TA5009487), mouse anti-alpha-tubulin DM1A (Sigma Aldrich, T6199) and sheep anti-SARS-CoV-2 nucleocapsid protein (3rd bleed)[92]. Secondary antibodies anti-rabbit IgG (H + L) DyLight 800 conjugate (Cell Signalling Technology, 5151S), anti-mouse IgG (H + L) DyLight 680 conjugate (Cell Signalling Technology, 5470S) and donkey anti-sheep IgG451 (H + L) DyLight 800 (ThermoFisher, SA5-10060) were used for immunoblotting before protein visualization using the Odyssey CLx imager (Li-Cor). Image analysis was performed using the Image Studio Lite Software (Li-Cor).

### Fusion assay

A split green fluorescence protein (GFP) cell–cell fusion was used to assess the fusogenic function of various spikes, essentially as described previously[32,93]. Briefly, AAT-GFP1-10 and AAT-GFP11 cells were co-cultured at a ratio of 1:1 and seeded at a total of $2 \times 10^4$ cells per well in black 96-well plates (Greiner, E21063HD) in FluoroBrite DMEM medium (ThermoFisher, A18967-01) supplemented with 10% FBS. The following day, cells were infected with the indicated viruses using the equivalent of $10^6$ Orf1a genome copies per well in FluoroBrite DMEM and 2% FBS, or transfected with 0.1 µg DNA per well of spike plasmid using Lipofectamine LTX (ThermoFisher, 15338-100). The GFP signal was acquired for 20 h using a CLARIOStar Plus (BMG LABTECH) equipped with an atmospheric control unit to maintain 37 °C and 5% $CO_2$. Data were analysed using MARS software and plotted with Graph-Pad Prism 9 software.

### Protease inhibitor studies

AAT cells were seeded in 48-well plates at a cell density of $10^5$ cells per well. Cells were pretreated for 1 h with 10 µM of either camostat mesylate (SML0057, Sigma Aldrich) or E64d (E8640, Sigma Aldrich) or both inhibitors before infection with indicated viruses ($4 \times 10^5$ Orf1a genome copies per well) or pseudoviruses. Supernatant was collected at 48 hpi, and viral RNA extracted and quantified as described above.

### Pseudovirus-based assays

Pseudoviruses were generated as previously described[32,94]. Briefly, the day before transfection, HEK-293T CD81 knockout cells were seeded in 6-well plates at a seeding density of $1 \times 10^6$ cells per well. The following day, 200 ng of plasmid expressing the SARS-CoV-2 spike with 18 amino-acid C-terminus deletions was mixed with 1.3 µg of p8.91 lentiviral packaging plasmid, 1.3 µg of pCSFLW luciferase reporter plasmid and Opti-MEM. Fugene 6 (Promega, E2691) was mixed with the plasmids at a 3:1 ratio following manufacturer protocol and incubated for 15 min before adding transfection complexes onto the cells. The following day, transfection media were replaced with fresh media. At 48 and 72 h post transfection, media were collected and filtered through a 0.45-µm filter before use. Producer cell lysates were generated to confirm expression of spike by western blot analysis.

To assess PV entry, HEK-293T, AAT or Calu-3 cells were seeded at a density of $3 \times 10^4$ cells per well of a white F-bottom 96-well plate, followed by addition of undiluted PVs at equal volumes. For drug inhibitor assays, cells were seeded at the same density in the presence of either camostat or E64d to a final concentration of 10 µM and incubated for 1 h before adding 50 µl of undiluted PV. Plates for all PV infection assays were incubated for 48 h before lysis using Glo-bright luciferase reagent (Promega).

### Domain swap constructs

Nucleotide sequences encoding codon-optimized C-terminally deleted spike proteins of B.1 and A.30 viruses were synthesized (BioBasics) and cloned into the pCDNA3.1 expression plasmid. Domain swaps were then performed between the two spikes by restriction enzyme digestion and subcloning. The A.30 mutations found within each domain swap are as follows: NTD: D80Y, ΔY144, ΔI210, D215G, ΔRSY246-248, L249M and W258L; RBD: R346K, T478R and E484K; S2: H655Y, P681H and Q957H.

### Animal experiments

Golden Syrian hamsters (HsdHan, AURA) were bred and maintained by Envigo. Experimental animals were housed on site at the Veterinary Research Facility (VRF) at the University of Glasgow. All experiments were carried out under Containment Level 3 (CL3) conditions. Animals were housed in individually ventilated cages on a 12-h light/dark cycle and provided with food and water ad libitum. Both male and female hamsters between 8–12 weeks old were used throughout the study.

Animals were randomized to treatment groups, but blinding was not possible. All experiments were performed in a Class I microbiological safety cabinet (MSC) in a CL3 suite. Hamsters were anaesthetized with oxygen (1.5 l min⁻¹) containing 5% isoflurane and intranasally dosed with 50 µl of virus diluted in DMEM ($3.75 \times 10^6$ genome copies) or media only for mock-infected controls. Animal were weighed daily and scored for signs of clinical disease. Successful virus replication in infected animals was assessed by RT–qPCR of oral swabs at 24 hpi. Throat swabs were collected by restraining hamsters and inserting a swab (MWE, MW813) into the mouth and rotating it five times on each tonsil. The swabs were then placed in 2 ml DMEM (ThermoFisher, 31966021) containing 2% FBS (ThermoFisher, 10270106) and a 100 IU ml⁻¹ penicillin and 100 µg ml⁻¹ streptomycin mixture (Merck, 11074440001) for 1 min. Animals were culled on day 6 post infection via a rising concentration of $CO_2$ at the peak of clinical disease, and the extent of lung pathology in tissues collected at post-mortem examination was quantified[33].

### Quantification of SARS-CoV-2 RNA in hamster throat swabs

Total RNA was isolated from throat swab samples homogenized in Trizol LS using the DirectZol Magbead RNA kit (Zymo Research, R2103) following manufacturer instructions. SARS-CoV-2 genomic RNA was quantified by RT–qPCR, as described above; however, reactions were multiplexed to also detect hamster β-actin as a reference gene using the following primer and probe sequences: hACTB-F: CTCCCAGCAC-CATGAAGATC; hACTB-R: GCTGGAAGGTGGACAGTG; hACTB-Probe: TGTGGATCGGTGGCTCCATCCTG with Cy5/BHQ modifications (Cy5). Viral RNA levels were first normalized to hamster β-actin and genomic copies were calculated by interpolating the adjusted ORF1ab Ct values from a standard curve. All runs were performed on the ABI7500 Fast instrument and results analysed with the 7500 Software v.2.3 (Applied Biosystems, Life Technologies).

### Histology, immunohistochemistry and in situ hybridization

Sections (3-µm thick) of formalin-fixed (8%) and paraffin-wax-embedded air–liquid-interface (ALI) cultures or hamster lung tissue were cut and mounted on glass slides. Slides were either stained with H&E or used for immunohistochemistry or in situ hybridization. For immunohistochemistry, the following antibodies were used: Caspase-3 (R&D Systems, diluted 1:500), TTF-1 (Leica), CD3 (Agilent Dako, A0452) and IBA1 (Fujifilm, Alpha labs). As a negative control, the primary antibody was replaced by isotype serum. For visualization, either the EnVision+/HRP, mouse, HRP kit (Agilent DAKO) or EnVision+/HRP, rabbit, HRP kit (Agilent DAKO) were used in an automated stainer (Autostainer Link 48, Agilent). 3,3′-Diaminobenzidine (DAB) was used as a chromogen. Apoptosis was detected using the HRP-DAB TUNEL-assay kit (ab206386, Abcam) according to manufacturer instructions, and mammalian mammary gland tissue served as positive control.

For in situ hybridization, RNA was detected using RNAScope according to manufacturer instructions (Advanced Cell Diagnostics, RNAScope) with simmering in target solution and proteinase K treatment. The following probes were used: SARS-CoV-2 spike (848561),

DapB (310043), ubiquitin (310041) as well as human IFIT-1 (415551) as previously described[33].

## Digital pathology

All slides were scanned with an Aperio VERSA 8 Brightfield, Fluorescence and FISH Digital Pathology Scanner (Leica Biosystems) at ×200 brightfield magnification and images analysed essentially as already described[33]. Areas to be analysed (either sections from ALI cultures, cells detached ('sloughed off') from the cultures, or sections from hamster lungs) were manually outlined using QuPath[95]. The algorithm to detect the percentage of positive stained area in pixel (IFIT-1 and spike RNA) was tuned individually for each set of immunohistochemistry or in situ hybridization experiments.

Quantification of epithelial lung hyperplasia was carried out as already described[33]. Briefly, the whole lung area was outlined. Subsequently, the HALO algorithm 'AI' was trained on several slides to detect clusters of TTF-1+ nuclei within the lung corresponding to type-2 pneumocytes hyperplasia while ignoring individual TTF-1+ cells positively stained in the normal alveoli or in the bronchial epithelium. The percentage of the area of hyperplastic TTF-1+ type-2 pneumocytes per total lung area was then calculated. For measuring caspase-3+ and TUNEL+ cells, as well as spike+, IFIT-1+, CD3+ and IBA1+ cells, QuPath was used to detect positive signals with the pixel classifier algorithm in the ALI cultures and the lungs. Slides with artefacts were excluded from the analysis. All photomicrographs were captured with an Aperio ImageScope (Leica Biosystems).

## Statistical analysis

Statistical analysis was performed using GraphPad Prism v.9 and v.10.2.2. Unless stated otherwise, the results are expressed as mean ± s.e.m. Data distribution was assumed to be normal unless stated differently, but this was not formally tested. Details of statistical tests performed are indicated in the figure legends. No statistical methods were used to predetermine sample sizes but our sample sizes are similar to those reported in previous publications[32,33,42,74,75]. Data collection and analysis were not performed blind to the conditions of the experiments. No animals or data points were excluded for the analyses.

## Illustrations and molecular modelling

Rendering of spike molecular structures was performed with ChimeraX[96].

## Reporting summary

Further information on research design is available in the Nature Portfolio Reporting Summary linked to this article.

## Data availability

Raw data underpinning the figures associated with this paper are available in the Enlighten repository at https://doi.org/10.5525/gla.researchdata.1698 (ref. 97). Representative whole-organ histopathology sections corresponding to some of the images shown in Fig. 6b can be visualized at variable magnifications using the 'CVR Virtual Microscope' (https://covid-atlas.cvr.gla.ac.uk). Whole-organ histopathology sections that can be visualized at different magnifications and representatives of the images shown in Fig. 7b are shown in the CVR Virtual Microscope (https://covid-atlas.cvr.gla.ac.uk). All data generated and analysed during this study are included in this paper and its supplementary files. Source data are provided with this paper.

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

## Acknowledgements

This study was funded by the MRC to the G2P2 consortium (MR/Y004205; to M.P. and A.H.P.), and the Wellcome Trust to the G2P-global consortium (226141/Z/22/Z; to M.P.). Additional support by the MRC is also acknowledged (MC_UU_00034/9; MC_UU_00034/5; MC-UU-00034/6; MC_UU00034/7 and MC_UU_00034/8; to M.P., J.G., D.L.R., A.H.P., B.J.W.). The funders had no role in study design, data collection and analysis, decision to publish or preparation of the manuscript. We thank colleagues in the G2P consortia for insightful discussions; D. Wright for the development and curation of the 'CVR Virtual Microscope' (https://covid-atlas.cvr.gla.ac.uk/); C. Boyd, S. McCall and N. Munro at the Biological Services of the University of Glasgow for assistance and guidance on animal experiments; L. Oxford, L. M. Stevenson, F. Bell, J. Lee and J. Duncan of the Veterinary Pathology Unit for excellent technical assistance.

## Author contributions

W.F., V.M.C., G.D.L., A.H.P. and M.P. conceptualized the project. W.F., V.M.C., G.D.L., R.O., V.H., D.C., D.C.M, K. Kerr, K.S., A.D.S.F., J.G., A.H.P. and M.P. developed the methodology. W.F., V.M.C., G.D.L., R.O., V.H., D.C., G.I., D.C.M., K. Kerr, J.A., N.U., G.R.M., S.B., U.R.D., S.M.A., M.M., S.L., N.L., K. Kwok, K.S., B.J.W. and J.G. conducted investigations. W.F., R.O., D.C. and J.G. performed visualization. W.F., V.M.C., G.D.L., D.C., J.G., A.H.P. and M.P. wrote the original draft. W.F., V.M.C., G.D.L., V.H., D.C., D.C.M., N.U., D.L.R., J.G., A.H.P. and M.P. reviewed and edited the paper. J.G., A.H.P. and M.P. acquired funding. J.G., A.H.P. and M.P. coordinated and supervised the project.

## Competing interests

The authors declare no competing interests.

## Additional information

**Extended data** is available for this paper at https://doi.org/10.1038/s41564-024-01878-5.

**Correspondence and requests for materials** should be addressed to Arvind H. Patel or Massimo Palmarini.

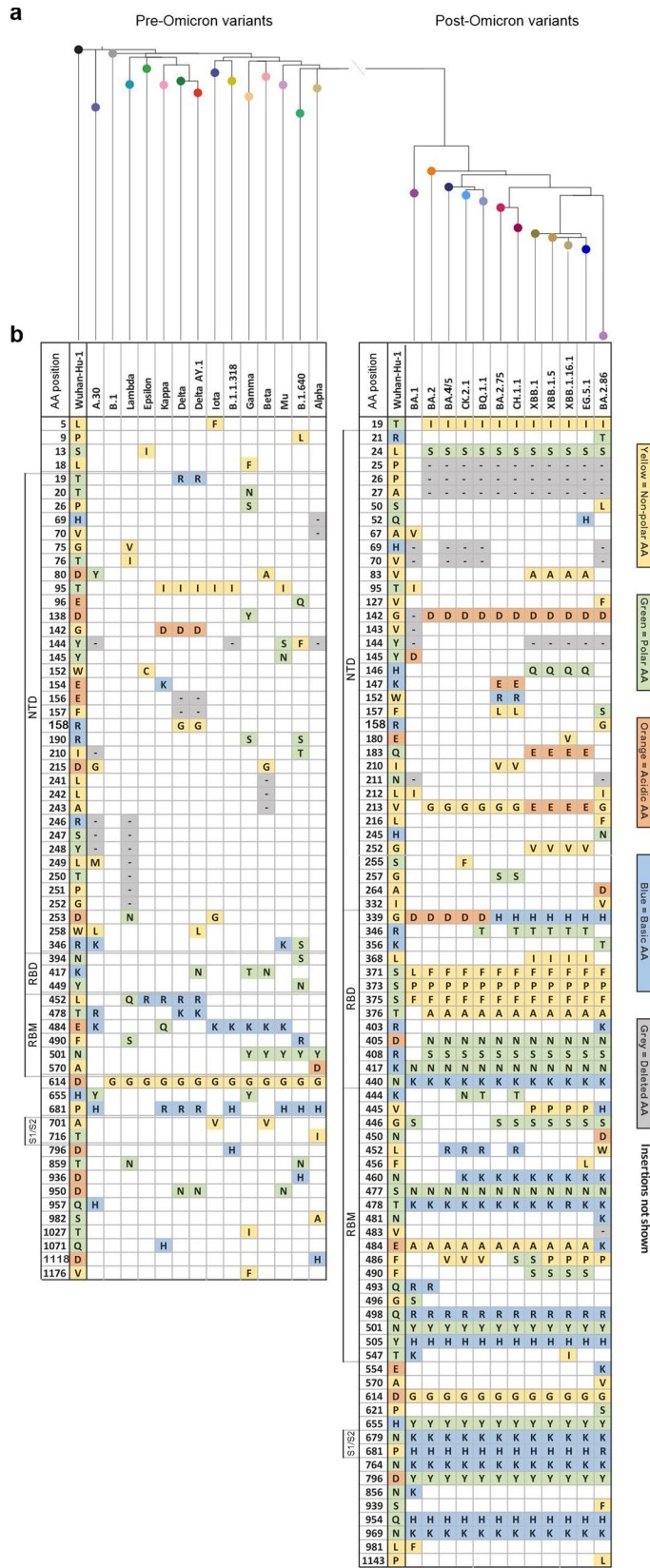

**Extended Data Fig. 1 | Phylogenetic relationship and spike amino acid mutations profiles of different SARS-CoV-2 variants. a**, Phylogenetic tree of variants nucleotide spike sequences, rooted on the Wuhan-Hu-1 reference sequence. **b**, Heatmap showing amino acid mutations and deletions in each spike variant sequence with respect to Wuhan-Hu-1 (yellow = non-polar amino acids,

green = polar, orange = acidic, blue = basic, grey = deletion). Variants are ordered based on their nucleotide spike phylogenetic tree shown at the top. The design of the B.1.640 spike was based on the initial genome sequences where the 136-144 deletion was not present in >50% and therefore not included; B.1.640 has since been split into two sub-lineages (.1 and .2).

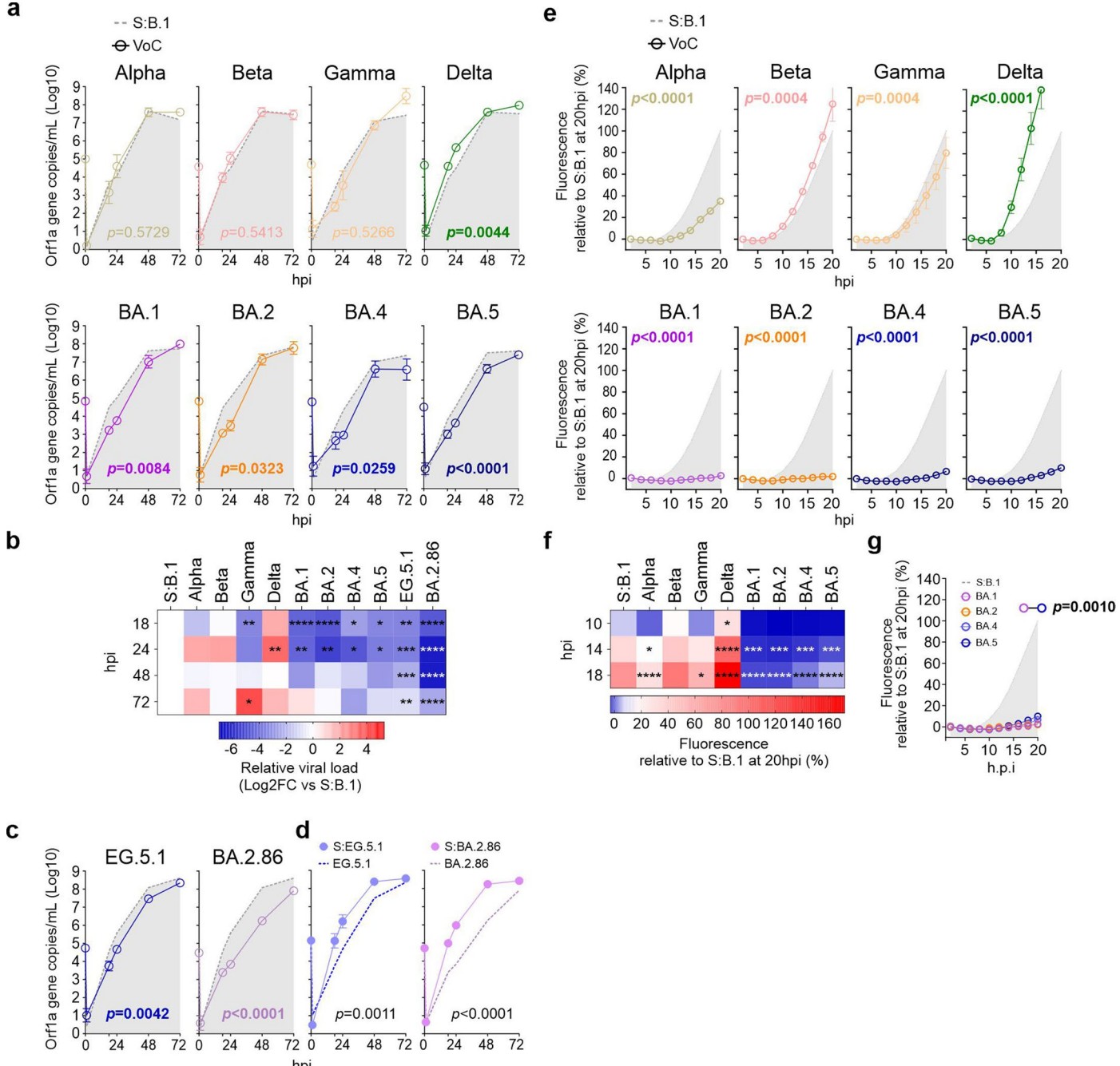

**Extended Data Fig. 2 | Replication kinetics and fusogenicity of wild type SARS-CoV-2 variants. a**, Replication kinetics of indicated VOCs. Calu-3 cells were infected with the pre-Omicron (top) or post-Omicron (bottom) SARS-CoV-2 VOCs. Copy numbers of viral RNA in the culture supernatant were quantified by qRT-PCR at the indicated timepoints. For (**a**, **c**, **e** and **g**) the light grey area highlights the replication curve of S:B.1 used as reference virus in all experiments. **b**, Heatmap of viral titres relative to S:B.1 for all viruses expressed as log2-fold change at each indicated timepoint. Statistical significance between S:B.1 and the clinical isolates at each timepoint was instead determined by one-sample t-test. * p ≤ 0.05, ** p ≤ 0.01, *** p ≤ 0.001, **** p ≤ 0.0001. **c**, Replication kinetics of EG.5.1 and BA.2.86 clinical isolates relative to S:B.1 in Calu-3 cells. **d**, Replication kinetics of S:EG.5.1 and S:BA.2.86 relative to the respective wild-type clinical isolates in Calu-3 cells. **e**, Live virus-based fusion assay in AAT-GFP10/AAT-GFP11 cells.

The fusion activity was measured over time and expressed as percentage of maximum fluorescence over S:B.1. **f**, Heatmap expressed as percentage of maximum fusion (shown in e) over S:B.1 at indicated timepoints. Statistical significance between S:B.1 and the clinical isolates at each timepoint was instead determined by Two-way ANOVA. * p ≤ 0.05, ** p ≤ 0.01, *** p ≤ 0.001, **** p ≤ 0.0001. **g**, Comparisons of the fusogenic capacity between Omicron BA.1 some of its sub-lineages (data from bottom row of Panel **e**). For (**a**,**c**,**d**,**e** and **g**) data are mean ± standard error to the mean (SEM) of at least three independent experiments, each performed in triplicate. Statistical significance between S:B.1 and the VOCs across timepoints was determined by one-way ANOVA with multiple comparisons between area under the curve or (**g**) between Omicron BA.1 and other sub-lineages using the Holm-Šídák post-hoc test. Calculated p-values are indicated in the figures.

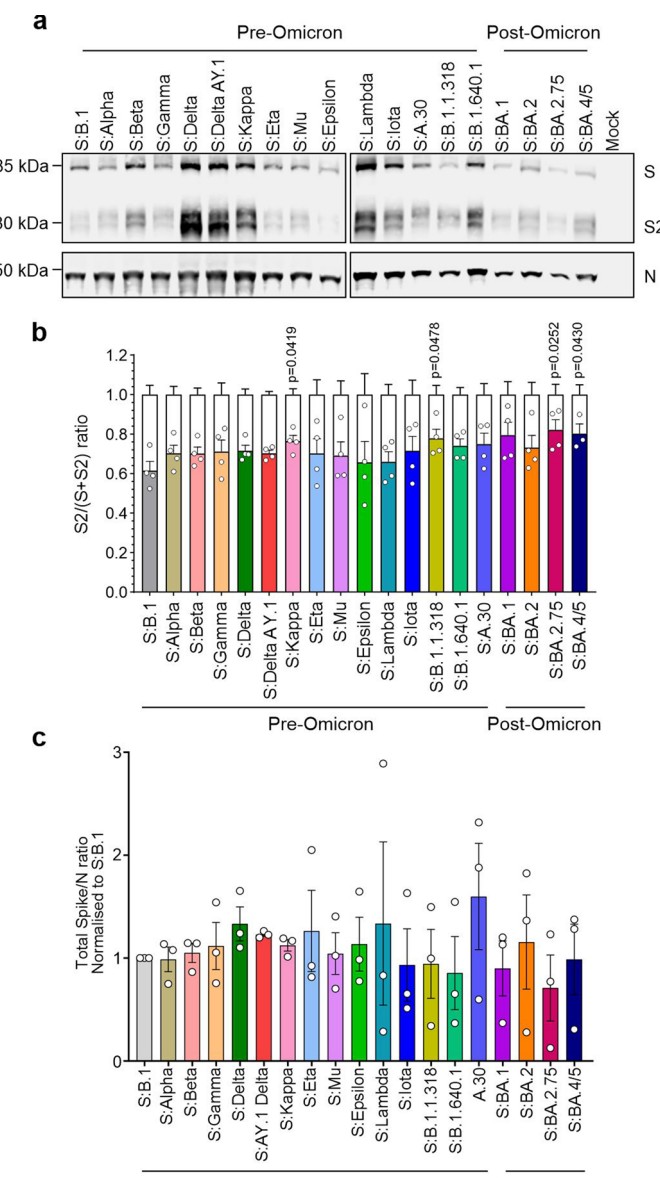

**Extended Data Fig. 3 | Proteolytic processing of spike of different SARS-CoV-2 recombinant viruses. a**, Western blotting of spike protein associated with SARS-CoV-2 virions released in supernatants of Calu-3 infected cells. An antibody against S2 was used to differentiated processed and unprocessed spike. SARS-CoV-2 N was used as an internal control for virus load. Experiments were repeated independently four times and representative blots are shown. **b**, Quantification of spike proteolytic cleavage resulting from four independent western blots as in (**a**). Results are expressed as cleaved S2 over total spike protein. **c**, Incorporation of spike on SARS-CoV-2 virions. Results are expressed as total spike per N protein and normalised to S:B.1 for comparisons. In general, we observed a trend for a marginal increase in spike processing for all viruses,

although reaching statistical significance only with S:Kappa, S:B.1.1.318, S:BA.2.75 and S:BA.4/5. We also observed marginally higher levels of spike incorporation in S:Delta, S:Delta AY.1, S:Kappa and S:Lambda (Extended Data Fig. 3a, c). Nonetheless, the quantification of spike incorporation into viral particles (that is the ratio between S and N signal) did not show any significant difference for all variants compared to S:B.1 (Extended Data Fig. 3c). **b,c**, Data are mean ± standard error to the mean (SEM) of four independent experiments. Statistical significance between S:B.1 and the spike recombinant SARS-CoV-2 was determined by one-way ANOVA with multiple comparisons between the different spike-bearing viruses and S:B.1 using the Holm-Šídák post-hoc test. Significance is indicated with * $p \leq 0.05$, ** $p \leq 0.01$, *** $p \leq 0.001$, **** $p \leq 0.0001$.

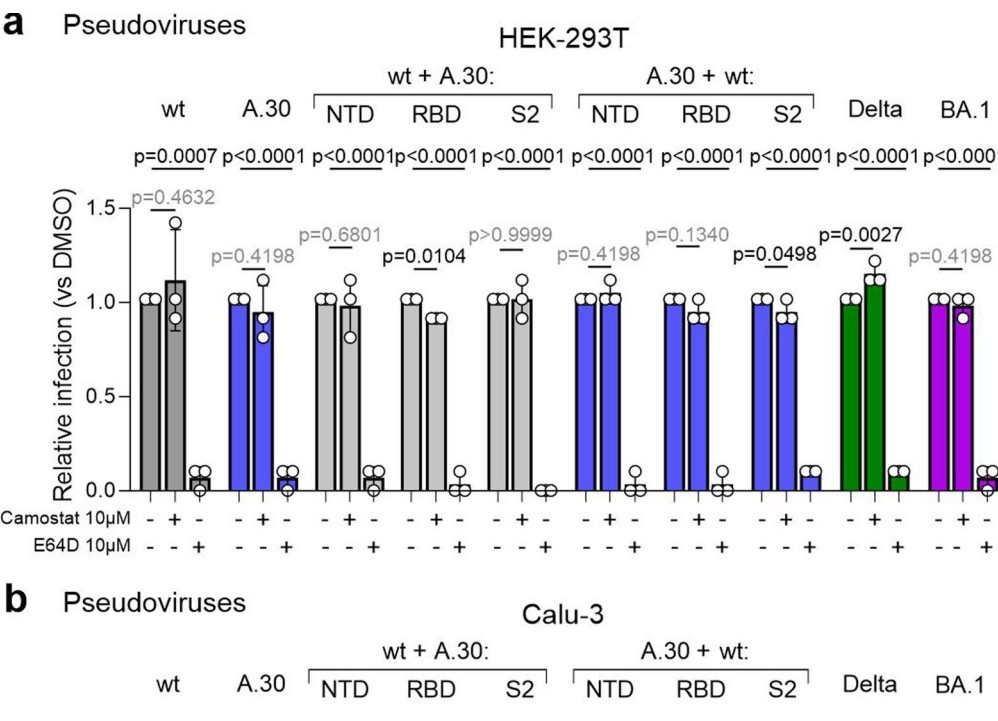

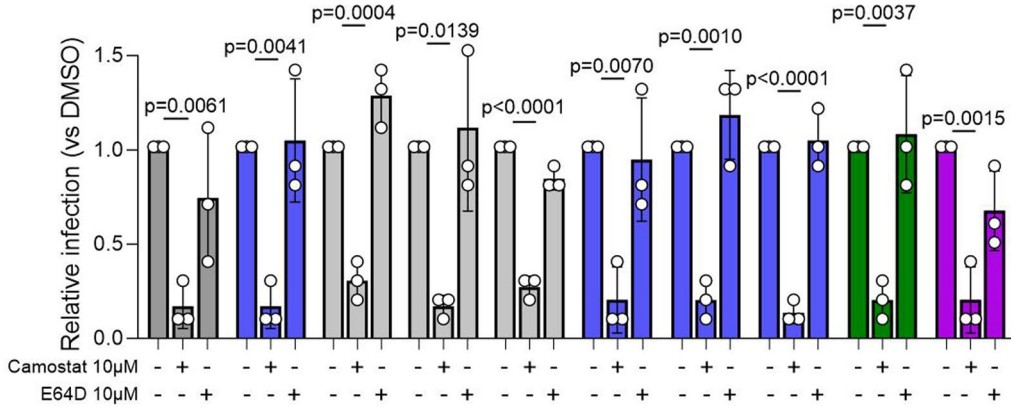

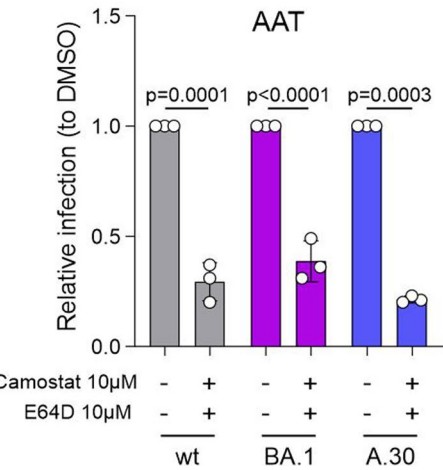

**Extended Data Fig. 4 | Cell entry pathways used by pseudoviruses (PVs).**
**a,b**, Sensitivity of domain swap PVs to Camostat and E64d in HEK-293T (**a**) and
Calu-3 (**b**) cells. **c**, Sensitivity of B.1, BA.1 and A.30 PVs to simultaneous treatment
with Camostat and E64d. Data are expressed as relative to the DMSO-treated

control for each virus. Data are mean ± standard error to the mean (SEM) of
three independent experiments each performed in triplicate (n = 3). Statistical
significance between DMSO-treated and inhibitor-treated conditions was
determined by One-sample t-test. * p ≤ 0.05, ** p ≤ 0.01, *** p ≤ 0.001, **** p ≤ 0.0001.

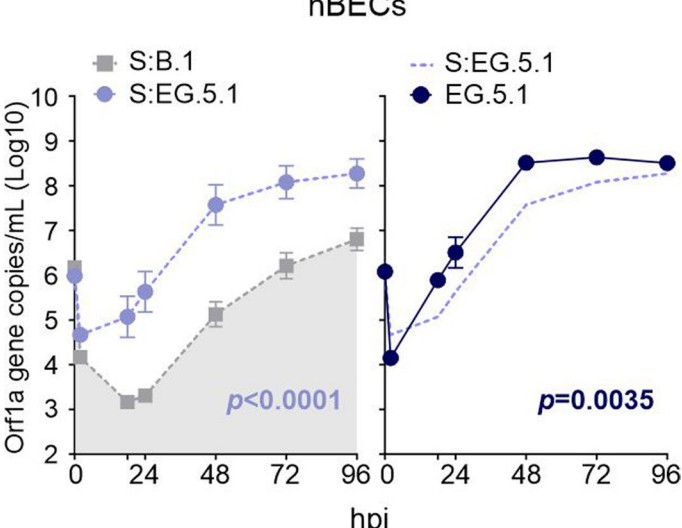

**Extended Data Fig. 5 | Comparative fitness of recombinant S:EG.5.1 and EG.5.1 clinical isolate in bronchial epithelium.** Replication kinetics of S:EG.5.1 and S:B.1 (Left) or corresponding EG.5.1 clinical isolate (Right). The copy numbers of viral RNA in the culture supernatant of infected hBECs were determined by RT-qPCR at the indicated timepoints. Data are mean ± standard error to the mean (SEM) of three biological replicates. Statistical significance between S:B.1 (Left) or EG.5.1 clinical isolate (Right) and the spike recombinant virus across time points was determined by one-way ANOVA with multiple comparisons between area under the curves using the Holm-Šídák post-hoc test. The calculated p-values are indicated in the figures.

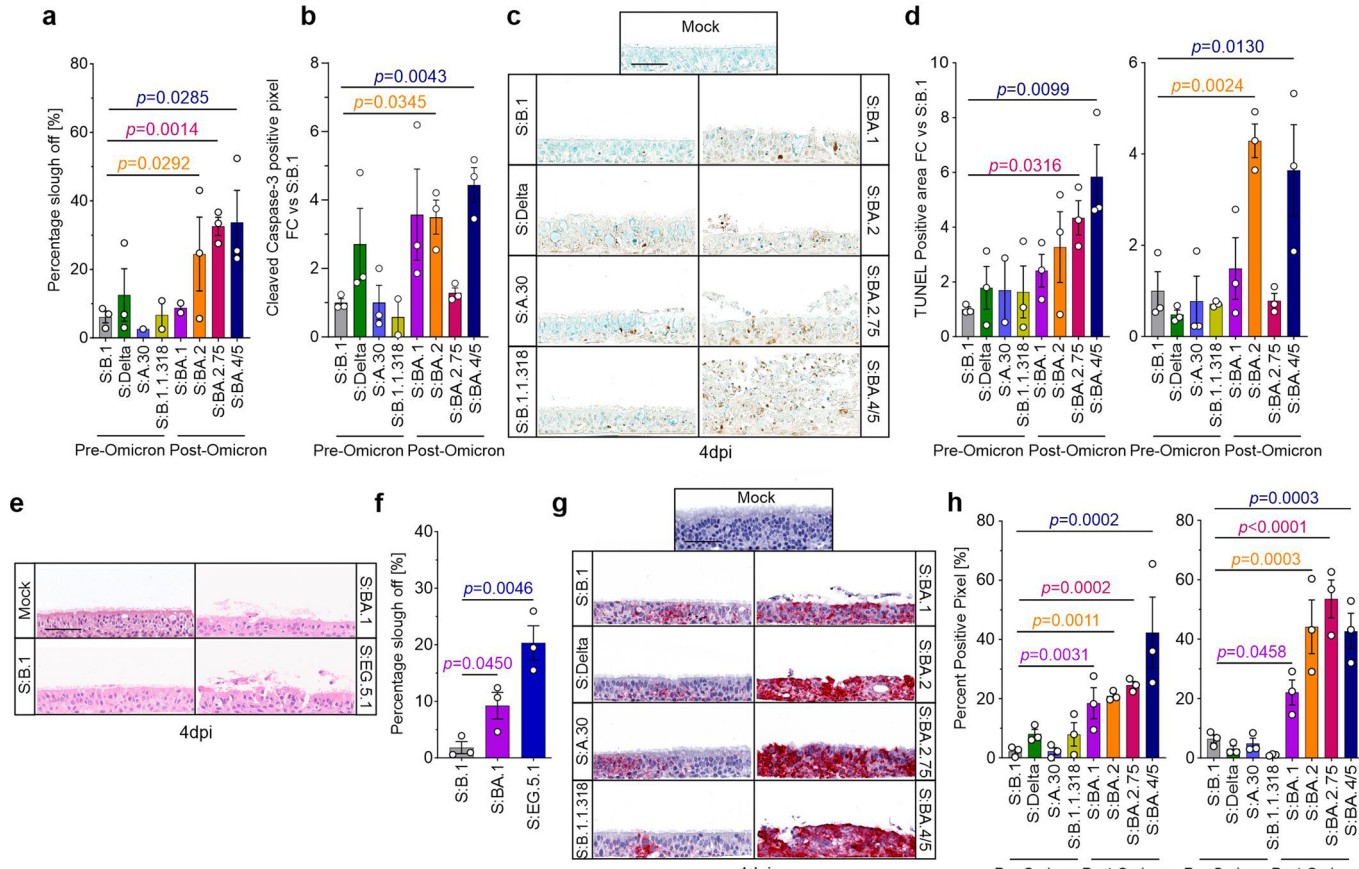

**Extended Data Fig. 6 | Cytopathogenicity of SARS-CoV-2 recombinant viruses in nasal epithelium at day 4. a**, Quantification of cell sloughing from hNECs infected with another clone of spike recombinant virus (n = 3) as in Fig. 4g, h, using whole-section software assisted imaging as described in Methods. **b**, Quantification of caspase-3 cleavage in hNECs (n = 3). Apoptotic cells were quantified using whole-section software assisted image analysis. **c**, Microphotographs showing apoptotic cells, as revealed by TUNEL assays, in hNEC infected with the indicated viruses at 4 dpi. Apoptotic nuclei are stained in brown. **d**, Quantification of apoptotic nuclei using whole-section software assisted image analysis as described in Methods. Three biological replicates are shown from two independent sets of clones for each spike recombinant virus.

**e**, Microphotographs of hNEC infected with either S:B.1, S:BA.1 or S:EG.5.1 and stained with haematoxylin and eosin to reveal sloughing. **f**, Quantification of cell sloughing in infected hNEC cultures (n = 3). **g**, Microphotographs showing RNA in situ hybridisation for the detection of IFIT1 at 4 dpi in hNEC infected with the indicated viruses. **h** Quantification of IFIT1 expression of two independent sets of clones (each n = 3) using whole-section software assisted image analysis. In (**a**,**b**,**d**,**f** and **h**) data are mean ± standard error to the mean (SEM) of three biological replicates. Statistical significance between S:B.1 and other spike recombinant SARS-CoV-2 was determined by one-way ANOVA with multiple comparisons between conditions using the Holm-Šídák post-hoc test. The calculated p-values are indicated in the figures. Scale bar: 200 μm.

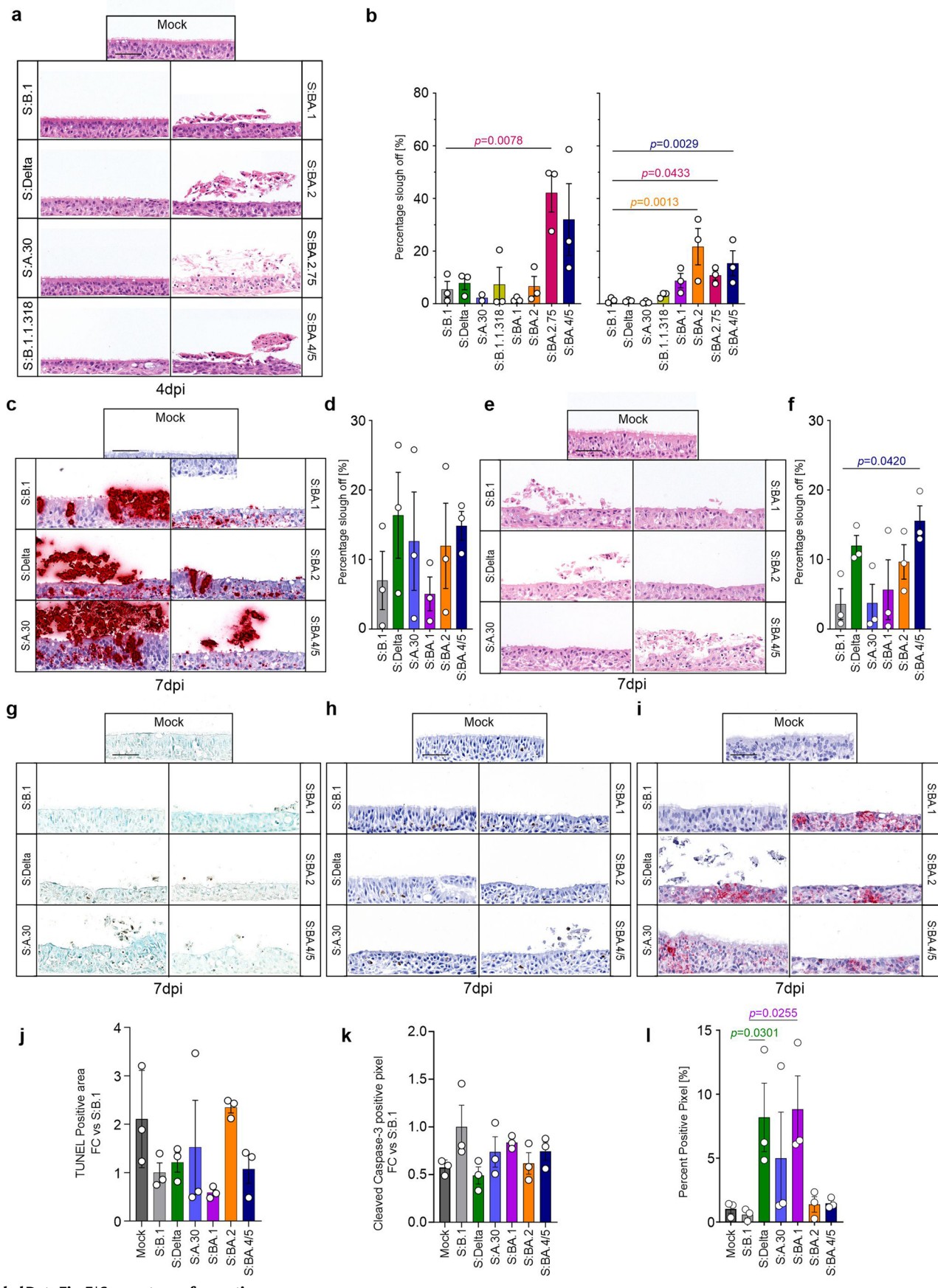

**Extended Data Fig. 7 | See next page for caption.**

**Extended Data Fig. 7 | Cytopathogenicity of recombinant viruses in nasal epithelium at day 7. a**, Microphotographs showing cell sloughing detected by haematoxylin and eosin (H&E) in infected hNEC with the indicated viruses at 4 dpi. **b**, Quantification of cell sloughing in three biological replicates using two independent set of clones for each spike recombinant virus using whole-section software assisted imaging. **c**, Microphotographs showing RNA in situ hybridisation of SARS-CoV-2 spike in hNEC at 7 dpi. **d**, Quantification of cell sloughing at 7 days post infection from three biological replicates. **e**, Microphotographs showing cell sloughing detected by H&E in infected hNEC with the indicated viruses at 7 dpi. **f**, Quantification of cell sloughing at 7dpi from three biological replicates. **g**, Microphotographs showing apoptotic cells, as revealed by TUNEL assays, in hNEC infected with the indicated viruses at 7 dpi.

Apoptotic nuclei are stained in brown. **h**, Microphotographs of cleaved caspase-3 in hNEC infected with the indicated viruses at 7 dpi. Cells with cleaved caspase-3 are stained in brown. **i**, Microphotographs showing RNA in situ-hybridisation for the detection of IFIT1 RNA at 7 dpi in hNEC infected with the indicated viruses. **j**, Quantification of apoptotic nuclei (revealed by TUNEL assays as in (**g**) using whole-section software assisted image analysis. **k**, Quantification of caspase-3 cleavage by software assisted image analysis. **l**, Quantification of IFIT1 RNA. Data are mean ± standard error to the mean (SEM) of three biological replicates. Statistical significance between S:B.1 and other spike-bearing SARS-CoV-2 was determined by one-way ANOVA with multiple comparisons between conditions using the Holm-Šídák post-hoc test. The calculated p-values are indicated in the figures. Scale bar: 200 µm.

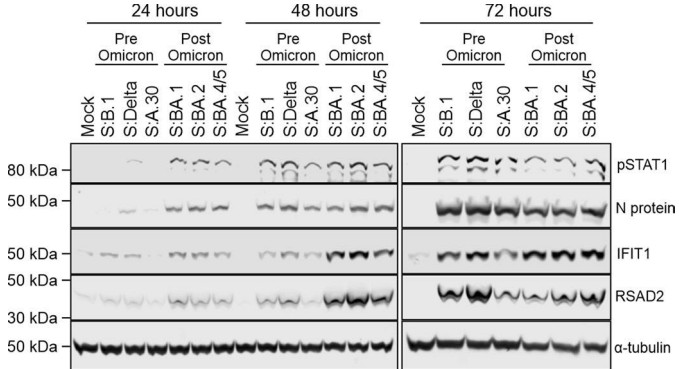

**Extended Data Fig. 8 | Induction of type-I interferon response in infected cells.** Western blotting of cell lysates obtained from hNEC infected with the indicated pre-Omicron and post-Omicron viruses. Lysates were blotted to detect markers of the type-I IFN response (pSTAT1, IFIT1 and RSAD2), alpha-tubulin and SARS-CoV-2 N, with the latter two used as loading control and to monitor levels of virus infection, respectively. One of two independent western blots is shown.

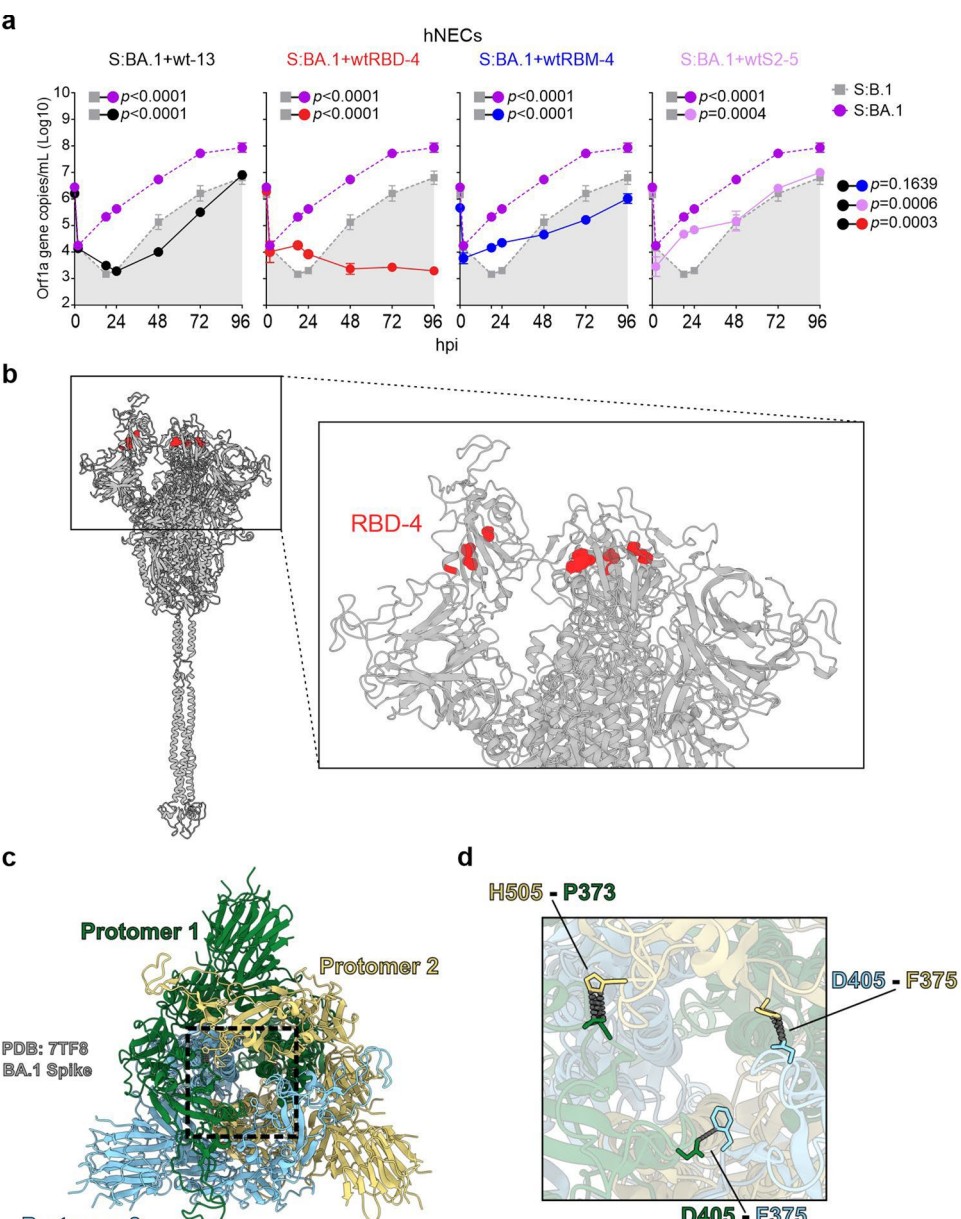

**Extended Data Fig. 9 | RBD mutations modulate hNEC infectivity.**
**a**, Replication kinetics of the SARS-CoV-2 spike mutants in human nasal epithelium (hNEC). Graphs represent data collected using an independent set of clones for each SARS-CoV-2 mutant (n = 3) as in Fig. 5c. S:B.1 (indicated by light grey area) and S:BA.1 (dashed purple line) viruses were used as reference viruses in all experiments. The copy numbers of viral RNA in the culture supernatant of infected hNECs were quantified by RT-qPCR at the indicated timepoints. Data are mean ± standard error to the mean (SEM). a, statistical significance between S:B.1 and the mutant spike virus panel across time points was determined one-way ANOVA with multiple comparisons between area under the curve of the different viruses and S:B.1 using the Holm-Šídák post-hoc test. The calculated p-values are indicated in the figure. **b**, Ribbon diagram molecular structure of spike[98] (based on PDB:6VSB, one RBD up conformation[99]) illustrating the location of the RBD-4 mutations. Left, full spike context; inset, zoom on the apex of spike. **c**, Top-down

view of BA.1 spike trimer (PDB:7TF8[117]) individual protomers are colour-coded. **d**, Inset, as marked in (**c**), displaying intra-protomer interactions mediated by BA.1 RBD mutations. Side chains are shown for interacting residues with contacts denoted by dashed grey lines. Interacting residues are labelled and color-coded by protomer, as in (**c**). Previous cryo-electron microscopy analysis indicates that the F371, P373 and F375 mutations modulate inter-protomer RBD-RBD[100], for instance by mediating interaction with the RBM of the adjacent protomer (for example H505-P373). It is likely that this will affect the conformational plasticity of the RBD (for example up/down switching), which in turn may impact ACE2 acquisition and spike stability. Reverting these residues in the context of BA.1 evidently creates incompatibilities that prevent hNEC infection; however, determining precisely how this relates to spike conformation and mechanism will require further investigation.

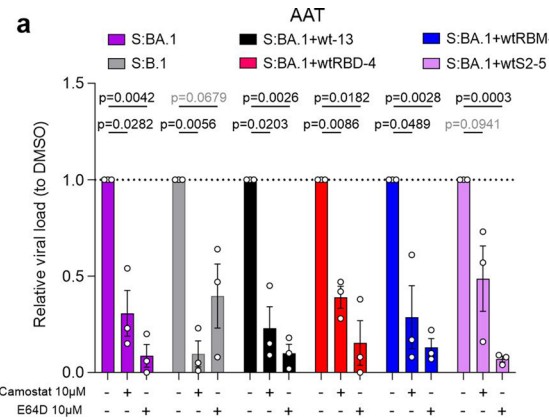

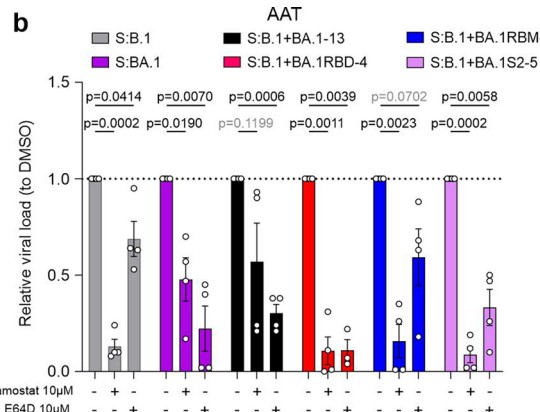

**Extended Data Fig. 10 | Cell entry pathway usage by SARS-CoV-2 mutants in nasal epithelium. a,b,** Sensitivity of the SARS-CoV-2 spike mutants to 10 µM each of Camostat and E64d in AAT cells. The dashed line represents the relative virus titre of DMSO-treated condition set as 1.0. Data are represented as mean ± standard error to the mean (SEM) of three or four independent experiments each performed in triplicate. Statistical significance between DMSO- and inhibitor-treated data was determined by one-sample t-test. * p ≤ 0.05, ** p ≤ 0.01, *** p ≤ 0.001, **** p ≤ 0.0001.

# Reporting Summary

## Statistics

For all statistical analyses, confirm that the following items are present in the figure legend, table legend, main text, or Methods section.

| n/a | Confirmed | |
|---|---|---|
| ☐ | ☒ | The exact sample size (*n*) for each experimental group/condition, given as a discrete number and unit of measurement |
| ☐ | ☒ | A statement on whether measurements were taken from distinct samples or whether the same sample was measured repeatedly |
| ☐ | ☒ | The statistical test(s) used AND whether they are one- or two-sided<br>*Only common tests should be described solely by name; describe more complex techniques in the Methods section.* |
| ☐ | ☒ | A description of all covariates tested |
| ☐ | ☒ | A description of any assumptions or corrections, such as tests of normality and adjustment for multiple comparisons |
| ☐ | ☒ | A full description of the statistical parameters including central tendency (e.g. means) or other basic estimates (e.g. regression coefficient) AND variation (e.g. standard deviation) or associated estimates of uncertainty (e.g. confidence intervals) |
| ☐ | ☒ | For null hypothesis testing, the test statistic (e.g. *F*, *t*, *r*) with confidence intervals, effect sizes, degrees of freedom and *P* value noted<br>*Give P values as exact values whenever suitable.* |
| ☒ | ☐ | For Bayesian analysis, information on the choice of priors and Markov chain Monte Carlo settings |
| ☒ | ☐ | For hierarchical and complex designs, identification of the appropriate level for tests and full reporting of outcomes |
| ☒ | ☐ | Estimates of effect sizes (e.g. Cohen's *d*, Pearson's *r*), indicating how they were calculated |

*Our web collection on statistics for biologists contains articles on many of the points above.*

## Software and code

Policy information about availability of computer code

| | |
|---|---|
| Data collection | Odissey infrared CLx Imager with Image Studio Lite software V 5.2, Applied Biosystems 7500 software, CLARIOstar Plus microplate reader (BMG Labtech), GloMax Explorer GM3500 multimode plate reader (Promega), Aperio VERSA Pathology slide scanner with software V1.0.1.125 and ImageScope V12.4.3 (Leica) |
| Data analysis | Applied Biosystems 7500 software, Image Studio Lite V5.2 (LI-COR), MARS data analysis software v5.02 R1 (BMG Labtech), GraphPad Prism Versions 9 & 10.2.2, HALO Image analysis platform V3.6 (Indica Labs), QuPath digital pathology and whole slide image analysis software V0.3.2, UCSF ChimeraX (UCSF RBVI) |

For manuscripts utilizing custom algorithms or software that are central to the research but not yet described in published literature, software must be made available to editors and reviewers. We strongly encourage code deposition in a community repository (e.g. GitHub). See the Nature Portfolio guidelines for submitting code & software for further information.

## Data

Policy information about availability of data

All manuscripts must include a data availability statement. This statement should provide the following information, where applicable:

- Accession codes, unique identifiers, or web links for publicly available datasets
- A description of any restrictions on data availability
- For clinical datasets or third party data, please ensure that the statement adheres to our policy

> All data generated and analysed during this study are included in this manuscript and supplementary information files will be made available before publication. SARS-CoV-2 sequence counts and genomic data were extracted from GISAID and COG-UK. No new algorithms were developed for this project.Raw data underpinning the figuresfigures associated with this manuscript are available in the Enlighten repository (https://doi.org/10.5525/gla.researchdata.1698)

## Research involving human participants, their data, or biological material

Policy information about studies with human participants or human data. See also policy information about sex, gender (identity/presentation), and sexual orientation and race, ethnicity and racism.

| | |
|---|---|
| Reporting on sex and gender | N/A |
| Reporting on race, ethnicity, or other socially relevant groupings | N/A |
| Population characteristics | N/A |
| Recruitment | N/A |
| Ethics oversight | N/A |

Note that full information on the approval of the study protocol must also be provided in the manuscript.

# Field-specific reporting

Please select the one below that is the best fit for your research. If you are not sure, read the appropriate sections before making your selection.

☒ Life sciences ☐ Behavioural & social sciences ☐ Ecological, evolutionary & environmental sciences

For a reference copy of the document with all sections, see nature.com/documents/nr-reporting-summary-flat.pdf

# Life sciences study design

All studies must disclose on these points even when the disclosure is negative.

| | |
|---|---|
| Sample size | No statistical methods were used to pre-determine sample sizes. Our sample sizes are similar to those reported in previous publications (Willett et al, 2022, Nature Microbiology; Reuschl et al, 2024, Nature Microbiology; Meehan et al, 2023, Plos Pathogens; Meng et al, 2022, Nature; Peacock et al, 2021, Nature Microbiology. Multiple independent experiments were repeated to allow for appropriate statistical analysis. |
| Data exclusions | No data were excluded |
| Replication | In vitro experiments were performed independently at least 3 times (unless stated otherwise) to allow for appropriate confidence in the reproducibility of the results. All attempts at replication were successful. |
| Randomization | No randomisation was performed. Experimental groups were treated identical except for the specific variables being tested. Thus, randomisation is not required. |
| Blinding | Blinding was not necessary as all measurements were quantified by automated machines and softwares. No data were excluded. |

# Reporting for specific materials, systems and methods

We require information from authors about some types of materials, experimental systems and methods used in many studies. Here, indicate whether each material, system or method listed is relevant to your study. If you are not sure if a list item applies to your research, read the appropriate section before selecting a response.

## Materials & experimental systems

| n/a | Involved in the study |
|-----|----------------------|
| ☐ | ☒ Antibodies |
| ☐ | ☒ Eukaryotic cell lines |
| ☒ | ☐ Palaeontology and archaeology |
| ☐ | ☒ Animals and other organisms |
| ☒ | ☐ Clinical data |
| ☒ | ☐ Dual use research of concern |
| ☒ | ☐ Plants |

## Methods

| n/a | Involved in the study |
|-----|----------------------|
| ☒ | ☐ ChIP-seq |
| ☒ | ☐ Flow cytometry |
| ☒ | ☐ MRI-based neuroimaging |

# Antibodies

| | |
|---|---|
| Antibodies used | Primary antibodies for western blot:<br>Rabbit anti-SARS-CoV-2 spike S2 (ThermoFisher, PA1-41165), sheep anti-SARS-CoV-2 nucleocapsid protein DA114( Rihn et al, 2021, Plos Biology https://mrcppu-covid.bio/antibodies/134473), rabbit anti-phospho STAT1 Tyr701 clone 58D6 (CellSignaling Technologies, 9167), mouse anti-alpha-Tubulin DM1A (Sigma-Aldrich, T6199), rabbit anti-RSAD2 (Proteintech, 28089-1-AP), mouse anti-IFIT1 (Origene, TA5009487). All used at 1:1000 dilution.<br><br>Secondary antibodies for western blot:<br>anti-rabbit IgG (H+L) DyLight 800 conjugate (CellSignalling Technology, 5151S, 1:20000 dilution), anti-mouse IgG (H+L) DyLight 680 conjugate (CellSignalling Technology, 5470S, 1:15000 dilution), anti-sheep IgG451 (H+L) DyLight 800 (Thermo Fisher, SA5-10060, 1:15000 dilution).<br><br>Primary antibodies for immunohistochemistry:<br>rabbit anti-IBA-1 (Alpha labs, 019-19741, 1:2500 dilution), mouse anti-TTF1 (Leica Biosystems, NCL-L-TTF-1, 1:200 dilution), rabbit anti-h/m Active Caspase 3 ( R&D systems, AF835, 1:500 dilution), CD3 (Agilent Dako, A0452, 1:200 dilution). |
| Validation | Rabbit anti-SARS-CoV-2 spike S2 (ThermoFisher, PA1-41165), validated by the manufacturer: https://www.thermofisher.com/antibody/product/SARS-Coronavirus-Spike-Protein-Antibody-Polyclonal/PA1-41165<br>Sheep anti-SARS-CoV-2 nucleocapsid protein (Rihn et al, 2021, Plos Biology): Validated by comparison of mock-infected and infected cell lysates. Migrates at the expected molecular weight (Rihn et al, 2021, Plos Biology https://doi.org/10.1371/journal.pbio.3001091)<br>Rabbit anti-phospho STAT1 Tyr701 clone 58D6 (CellSignaling Technologies, 9167): Validated by the manufacturer. Detects endogenous levels of STAT1 only when phosphorylated at tyrosine 701. It does not cross-react with the corresponding phospho-tyrosines of other STAT proteins. https://www.cellsignal.com/products/primary-antibodies/phospho-stat1-tyr701-58d6-rabbit-mab/9167<br>mouse anti-alpha-Tubulin DM1A (Sigma-Aldrich, T6199): Validated by the manufacturer: https://www.sigmaaldrich.com/GB/en/product/sigma/t6199<br>rabbit anti-RSAD2 (Proteintech, 28089-1-AP), validated by the manufacturer: https://www.ptglab.com/products/RSAD2-Antibody-28089-1-AP.htm<br>mouse anti-IFIT1 (Origene, TA5009487), validated by the manufacturer: https://www.origene.com/catalog/antibodies/primary-antibodies/ta500948/ifit1-mouse-monoclonal-antibody-clone-id-oti3g8<br><br>rabbit anti-IBA-1 (Alpha labs, 019-19741), validated by the manufacturer https://www.alphalabs.co.uk/019-19741#literature<br>mouse anti-TTF1 (Leica Biosystems, NCL-L-TTF-1), validated by the manufacturer: https://shop.leicabiosystems.com/en-gb/ihc-ish/ihc-primary-antibodies/pid-thyroid-transcription-factor-1.<br>rabbit anti-h/m Active Caspase 3 ( R&D systems, AF835), https://www.rndsystems.com/products/human-mouse-active-caspase-3-antibody_af835. CD3 (Agilent Dako, A0452), validated by the manufacturer, https://www.agilent.com/en/product/immunohistochemistry/antibodies-controls/primary-antibodies/cd3-(concentrate)-76133. |

# Eukaryotic cell lines

Policy information about cell lines and Sex and Gender in Research

| | |
|---|---|
| Cell line source(s) | Calu-3 cells were commercially obtained from ATCC (HTB-55), A549 cells (ATCC, #CCL-185) expressing hACE2 and TMPRSS2 were generated at the CVR and described previosuly (Rihn et al, 2021, Plos Biology). HEK-293T cells (ATCC, #CRL-3216) expressing hACE2 were described previously (Willett et al, 2022, Nature Microbiology). Reconstituted human nasal and bronchial epithelium cultures hNECS and hBECs were commercially obtained from EPITHELIX, Switzerland (Peacock et al, 2021, Nature Microbiology; Willett et al, 2022, Nature Microbiology). BHK-N-hACE2 cells deriving from BHK-21 () ATCC #CCL-10 were generated at CVR and described previously (Willett et al, 2022, Nature Microbiology). |
| Authentication | Cell lines were commercially procured and authenticated by the supplier. |
| Mycoplasma contamination | Mycoplasma testing was conducted throughout the duration of the study and cells tested negative. |
| Commonly misidentified lines<br>(See ICLAC register) | None |

# Animals and other research organisms

Policy information about studies involving animals; ARRIVE guidelines recommended for reporting animal research, and Sex and Gender in Research

| | |
|---|---|
| Laboratory animals | 8-12 weeks old Golden syrian hamsters (HsdHan®:AURA) |
| Wild animals | No wild animals were used in the study |
| Reporting on sex | All experiments were done with sex and age matched groups. Bot sexes were used equally split within each group. |
| Field-collected samples | No field collected samples were used in the study |
| Ethics oversight | Procedures were performed under UK Home Office Licence PP0271643 in accordance with the Animals (Scientific Procedures) Act 1986 and approved by the University of Glasgow ethics committee. All animal research adhered to ARRIVE guidelines. |

Note that full information on the approval of the study protocol must also be provided in the manuscript.

# Plants

| | |
|---|---|
| Seed stocks | *Report on the source of all seed stocks or other plant material used. If applicable, state the seed stock centre and catalogue number. If plant specimens were collected from the field, describe the collection location, date and sampling procedures.* |
| Novel plant genotypes | *Describe the methods by which all novel plant genotypes were produced. This includes those generated by transgenic approaches, gene editing, chemical/radiation-based mutagenesis and hybridization. For transgenic lines, describe the transformation method, the number of independent lines analyzed and the generation upon which experiments were performed. For gene-edited lines, describe the editor used, the endogenous sequence targeted for editing, the targeting guide RNA sequence (if applicable) and how the editor was applied.* |
| Authentication | *Describe any authentication procedures for each seed stock used or novel genotype generated. Describe any experiments used to assess the effect of a mutation and, where applicable, how potential secondary effects (e.g. second site T-DNA insertions, mosiacism, off-target gene editing) were examined.* |

