## [Peer Review File · Nature Microbiology]

Phenotypic evolution of SARS-CoV-2 spike during the COVID-19 pandemic

Corresponding Author: Professor Massimo Palmarini

Version 0:

Decision Letter:

10th April 2024

Dear Professor Palmarini,

Thank you very much for your enquiry about submitting your manuscript "Phenotypic evolution of SARS-CoV-2 spike during the COVID-19 pandemic" to Nature Microbiology. We find the topic to be interesting, however, I'm sure you'll understand that we cannot make a firm decision about whether to send the paper out to review until we have carefully read the full manuscript to consider your discussion, along with the broader context and background literature.

In order to submit your complete manuscript to Nature Microbiology, please use the link below:

Link Redacted

If you have any questions, please feel free to contact me.

Best regards,

Version 1:

Reviewer comments:

Reviewer #1

(Remarks to the Author)

This submission provides comparative analyses of 27 SARS-CoV-2 spike proteins. The spikes represent those evolving throughout the pandemic and are clustered into "pre-omicron" and "post-omicron" groups. The spikes were incorporated into infectious SARS-CoV-2s, all of which have the same non-spike B.1 backbone genes. These 27 recombinant viruses were then compared for relative virus growth in immortalized cells and also in human nasal airway epithelial cultures (hNEC). The recombinant viruses were also compared in hamster pathogenicity tests.

The results support several important claims. Relatively to viruses with pre-omicron spikes, those with post-omicron spikes grow most robustly in hNEC and are generally less pathogenic in hamsters. Pronounced post-omicron growth in hNEC caused cell sloughing, an important finding that may help explain high omicron transmissibility. This adaptation to hNEC and attenuation of virulence is not correlated with spike-mediated preference for particular host proteases and is not correlated with the potency of spike-mediated cell-cell fusion. Focusing on a particular post-omicron BA.1 spike, the adaptation to hNEC has some correlation with evolution of the RBD, although detailed mechanism of this RBD effect remains unknown.

There have been a great many papers on this topic of SARS-CoV-2 spike evolution in relation to spike functions, virus tropism and virus virulence. In these papers, the tests of spike functions parallel those used in this submission, i.e., they test for successful infection in different cell lines, for potency of spike-directed membrane fusion, and for suppression of spike functions by host protease inhibitors. But most of these papers have used separately-expressed spike genes and/or pseudovirus surrogates of SARS-CoV-2 for analyses, and many have stopped short of using hNEC or hHAE for testing spike functions. The strength of this comprehensive paper is with its construction of a very large collection of isogenic replication-competent recombinant SARS-CoV-2 viruses varying only in spike, and with the use of these viruses in physiologically relevant hNEC and in vivo experimental conditions. Therefore, the results in this submission have increased reliability. This submission also makes a few significant discoveries about specific RBD determinants of hNEC infection and hNEC infected-cell sloughing. The comprehensive data provide a platform for future investigations of spike functions necessary for nasal tropism and endemic virus

transmissibility.

Comments:

1. The input multiplicities of infection are stated in terms of viral genome equivalents per cell. Can input multiplicities be stated in terms of infectious units per cell? This is asked because readers may want to know whether the growth curves in Figs 2, 4, 5 and some supplementary figs are single-cycle or multi-cycle. It is nearly certain that they are multi-cycle but it would help if input MOIs were stated or at least estimated.
2. Figs 2CD and text lines 168-181: it may not be possible to make strong conclusions about variable spike-mediated cell-cell fusion without knowing the relative levels of spike on plasma membrane surfaces. Also, cell-cell fusion signals may be artificially high in AAT overexpression cells, and it is not known whether cell-cell fusion is a meaningful feature of hNEC or in vivo SARS-CoV-2 infection and spread. It may be worth stating these limitations.
3. Fig. 3 and Fig 5D: Can the protease inhibitor studies be performed using hNEC? The advantages of the AAT host cells in discerning TMPR vs cathepsin utilization are nicely described but the assumption that findings from AAT can be extended to hNEC conditions might be questioned.
4. Fig 5BC results are truly remarkable in showing RBD substitutions controlling hNEC infection. Minor point, but might the RBD changes be put into spike structural context; to get further toward some hypothetical mechanistic insight into the very pronounced effect of these RBD changes on virus tropism?

Reviewer #2

(Remarks to the Author)

In this carefully presented manuscript (MS), the authors employed a reverse genetics system using the SARS-CoV-2 ancestral strain as the backbone to generate recombinant viruses by replacing the S-protein with those from 27 strains. Using S:B.1 recombinant virus as the reference (SARS-CoV-2 ancestral strain with the G614D Spike). The authors systematically compared the effects of different S-proteins on viral replication, membrane fusion, viral entry pathways, cell tropism, as well as viral virulence and pathogenicity. Additionally, they categorized the strains into pre-Omicron and post-Omicron groups to analyze the phenotypic changes of the recombinant viruses. Overall, the study is comprehensive and the data obtained from the reverse genetics approach are particularly valuable. The study provides interesting new data in understanding the SARS-CoV-2 S-protein evolution.

In general, the work is important and the editor may consider the following issues being clarified before publication:

Major points:

1. In Fig.2A/B, a few of the replication curves between S:B.1 and S:variant are remarkably similar in the left panels (For example S:EG.5.1 in Fig.2B). However, many time points are scored as significantly different in the heat map on the right side panels (For example S:EG.5.1 in Fig.2B). Whether the statistical method used to score the differences robust? Why at or near time 0, there are two data points? Are the first points for virus quantification? If so, why in different curves the first data points have different values? I am not sure if it would be better to show the differences between control and variant curves if the curves are shown in Log2 scale instead of Log10 scale. The heat-map is shown in Log2 scale, would it be better to match up? In the legend, the right panel heat maps are defined as "viral loads difference". Is the viral load evaluated the same as the left panels as orf1 gene copy/ml? In addition, not sure why heat maps are not included for Fig.S1?
2. In Fig.3, there is a significant difference in the inhibition of S:BA.1 (A/D) and BA.1(F) by camostat. Several published studies (Willett BJ, et al. Nat Microbiol. 2022. Fig.5G; Emerg Microbes Infect. 2022 Dec;11(1):277-283. Fig.3C; mBio. 2023 Feb 28;14(1):e0317622. Fig.2) have reported that camostat is unable to effectively inhibit BA.1. However, in this study, S:BA.1 appears to be more sensitive to camostat. This is especially true in light of the reference (Willett BJ, et al. Nat Microbiol), in which some authors in this MS also participated, that in AAT cells (presumably the same cell used in this study?), clinically isolated BA.1 variant was not efficiently inhibited by camostat. Can the authors discuss possible reasons?

Does this result imply a difference in the entry mechanism between the recombinant virus and the clinically isolated virus, suggesting that the recombinant Spike-variant only virus may not fully reproduce the infection phenotype of the clinically isolated virus? The authors cited references implying that cell tropism determinants outside of the spike gene are implicated. Based on reverse genetic virus data presented, can the authors confirm such proposal? If not what future experiments are needed?

Also in Fig.3A/B, the data dispersion is rather high, could such experimental features/errors prevent accurate assessment of inhibition by different protease inhibitors?

3. The findings presented in Fig.5 are crucial for mechanistic understanding of key S-protein amino acid residues for the nasal epithelial cell tropism. While the result is already informative, I am surprised that Pastorio et al., 2022, Cell Host & Microbe (PMID: 35931073) is not cited when D339, F371, P373 and F375 were identified to play an important role in nasal cell infection. In Pastorio et al., when F371 and F375, and to a lesser degree P373, mutations were introduced individually in the ancestral S-protein, the spike processing and pseudovirus entry were severely affected. Based on Fig. 5 results, the author somewhat dismiss in the Discussion section earlier proposed correlations between spike processing, fusogenicity, cell entry pathways and

tissue tropism. Would it be useful to test phenotypic changes when D339, F371, P373 and F375 are introduced individually or together on B.1 S-protein as recombinant viruses rather than just simply reverting them in the BA.1 S-protein?

Also I am not sure if spike processing and fusion data for S:BA.1-wt13, S:BA.1-wt4-RBD, S:BA.1-wt4-RBM and S:BA.1-wt5-S2 are presented?

Please also cite other function of F371, P373 and F375 mutations such as McCallum et al (PMID: 35076256) and possibly others on antibody evasion and He et al (PMID: 38454157) on interference with antigen presentation.

4. If I understand correctly, the authors somewhat hint virus independent factors for tissue tropism, including cell type and expression of ADAM and MMP. Would it be more informative, if at all possible in reasonable time, but also possible in a follow up study, phenotypic characteristics in a few more selected cell lines tested, for example Caco-2 and Vero-TMPRSS2 cells used in Pastorio et al and Kimura et al to enable direct comparison between studies?

Minor points:

1. Line 60, please unify spelling of ACE2.

2. Line 205-206 may not be technically accurate. Cathepsin L sites have been reported in S1 for SARS-CoV-2 (PMID: 34548480, PMID: 35668062) and near the S1/S2 site in both SARS-CoV-1 (PMID: 18562523) and SARS-CoV-2 (PMC7255728). Is there evidence to show that S2' site is cut by Cathepsin in SARS-CoV-2?

3. In Fig.2A/B, there are certain differences in the growth curves of S:B.1 when compared with different recombinant strains. For instance, in Fig.2B S:BA.2.75, the gray dashed region is noticeably lower than the corresponding regions in other strains. As a control, S:B.1's proliferation curve should ideally remain consistent across different experimental groups. Please explain the reason for this phenomenon.

4. Some data were labeled for cell line used but not all. Please label the cell line corresponding to each part of the data consistently throughout the MS, including the ones in Fig.4 and Fig 5B/C.

5. Line 178-179 "Importantly, S:EG.5.1 and S:BA.2.86 were as fusogenic, if not more, than S:B.1 (Figure 2D)" Kei Sato found that BA.2.86 induced fusion significantly lower than EG.5.1, approaching BA.2 (Tamura et al., 2024, Cell Host & Microbe 32, 1–11); Olivier found that BA.2.86 Spike induced fusion ability was lower than XBB, much lower than D614G (Nature Communications (2024) 15:2254); these results differ from those in this article. Could the authors discuss possible reasons for different observations among different studies?

6. Line 183-190, should "Figure S1B" be "Figure S1D"?

7. Line 233-235, The use of only one lineage A strain in this experiment may not be sufficient to conclusively assert that the entire A lineage exclusively employs the endosomal pathway? Additionally, since recombinant viruses were used in this study, and cell entry may be dependent on spike independent factors as implied by the author, whether further investigation is required to determine if clinical lineage A strains still exhibit this characteristic?

8. Line 611 "infected 5 days later with the equivalent of 104 Orf1a genome copies per well in serum free RPMI-1640 medium." Why did the authors quantify the virus using Orf1a genome copies instead of using multiplicity of infection (MOI) or TCID50? Have the authors confirmed that virus genome copy number correlates well with the virus infectivity across different strains? Will there be interference from defective viral particles?

Reviewer #3

(Remarks to the Author)

Furnon et al perform an extensive study of many of the SARS-CoV-2 variants that emerged during the COVID-19 pandemic. Infectious isogenic recombinant and original strains of virus were used to infect a variety of cell lines and primary cells. The results are in general agreement with previous studies, demonstrating that the virus has continued to evolve to better infect nasal tissue, and by extension, become more transmissible. In addition, some of the newer variants, in addition to better infecting nasal tissue, also have regained some of the features of early variants associated with the earlier, more virulent variants (cell surface vs. endosomal entry, fusogenicity, etc). The data are convincing and are more comprehensive than previous studies. The manuscript is well written. However, the amount of new information is limited, other than the data in Figure 5. Almost all of the conclusions were made in previous publications, albeit in studies of a smaller set of variants.

Specific comments.

1. Line 302, Figure 4E-Bronchial epithelia cells were used. Presumably these are primary cells. Their origin should be described in the Materials and Methods. Studies using primary human bronchial cells infected with a subset of the pre-Omicron and post-Omicron isolates should also be performed to support the conclusions from the studies using Calu-3 cells. Alternatively, published reports that use primary cells could be included to support the lung cell conclusions.

2. Page 13, Figure 5-The data in this figure are interesting. Specifically, the effects of changes in the non-RBM part of the RBD are striking in their effects of virus replication in primary nasal epithelial cells. This part of the manuscript should be further developed.

-The authors show that the mutations have no effects on growth in Calu3 cells. Were any effects observed on growth in primary bronchial cells (HAEs)?

-Information about S protein processing (S1-S2 cleavage; S2' cleavage) or fusogenicity in nasal epithelial cells might provide

insight into the basis of these differences.

3. Line 387, Figure 7-Levels of infectious virus or genomic RNA in hamster lungs should be included in these analyses.

4. Figure 7, line 352 (“...potentially enhancing viral transmission...”)—Since SARS-CoV-2-infected hamsters are a very useful model for studying virus transmission, transmission should be directly analyzed in infected hamsters. Alternatively, it could be discussed since some transmission data with the different isolates have previously been published.

Minor comments.

Line 184—Should be Figure S1D, not S1B.

Line 322-(iv) S:BA.1-wt-RBD possesses the 4 B.1 mutations in the RBD. This is not accurate because the RBD also contains all of the mutations present in the RBM.

Decision Letter:

17th May 2024

Dear Professor Palmarini,

Thank you for your patience while your manuscript "Phenotypic evolution of SARS-CoV-2 spike during the COVID-19 pandemic" was under peer-review at Nature Microbiology. It has now been seen by 3 referees, whose expertise and comments you will find at the end of this email. Although they find your work of some potential interest, they have raised a number of concerns that will need to be addressed before we can consider publication of the work in Nature Microbiology.

In particular, we would ask that you repeat a subset of analyses in additional cell lines (including primary cells), as suggested by Reviewers #1-3, and include the additional in vivo data outlined by Reviewer #3. Otherwise, we would ask that you extend your analyses following the additional suggestions outlined by each reviewer.

Please include a data availability statement as a separate section after Methods but before references, under the heading "Data Availability". This section should inform readers about the availability of the data used to support the conclusions of your study. This information includes accession codes to public repositories (data banks for protein, DNA or RNA sequences, microarray, proteomics data etc...), references to source data published alongside the paper, unique identifiers such as URLs to data repository entries, or data set DOIs, and any other statement about data availability. At a minimum, you should include the following statement: "The data that support the findings of this study are available from the corresponding author upon request", mentioning any restrictions on availability. If DOIs are provided, we also strongly encourage including these in the Reference list (authors, title, publisher (repository name), identifier, year). For more guidance on how to write this section please see: <http://www.nature.com/authors/policies/data/data-availability-statements-data-citations.pdf>

* If you have not done so already we suggest that you begin to revise your manuscript so that it conforms to our Article format instructions at <http://www.nature.com/nmicrobiol/info/final-submission>. Refer also to any guidelines provided in this letter.

When submitting the revised version of your manuscript, please pay close attention to our [href="https://www.nature.com/nature-portfolio/editorial-policies/image-integrity">Digital Image Integrity Guidelines](https://www.nature.com/nature-portfolio/editorial-policies/image-integrity) and to the following points below:

Link Redacted

Note: This url links to your confidential homepage and associated information about manuscripts you may have submitted or be reviewing for us. If you wish to forward this e-mail to co-authors, please delete this link to your homepage first.

Nature Microbiology is committed to improving transparency in authorship. As part of our efforts in this direction, we are now requesting that all authors identified as 'corresponding author' on published papers create and link their Open Researcher and Contributor Identifier (ORCID) with their account on the Manuscript Tracking System (MTS), prior to acceptance. This applies to primary research papers only. ORCID helps the scientific community achieve unambiguous attribution of all scholarly contributions. You can create and link your ORCID from the home page of the MTS by clicking on 'Modify my Springer Nature account'. For more information please visit www.springernature.com/orcid.

If you wish to submit a suitably revised manuscript we would hope to receive it within 3-4 months. If you cannot send it within this time, please let us know. We will be happy to consider your revision, even if a similar study has been accepted for publication at Nature Microbiology or published elsewhere (up to a maximum of 6 months).

Best regards,

Reviewer Expertise:

Referee #1: Molecular biology of coronaviruses
Referee #2: Mechanisms of (corona)virus infection
Referee #3: Molecular biology of coronaviruses

Reviewer Comments:

Reviewer #1 (Remarks to the Author):

This submission provides comparative analyses of 27 SARS-CoV-2 spike proteins. The spikes represent those evolving throughout the pandemic and are clustered into "pre-omicron" and "post-omicron groups. The spikes were incorporated into infectious SARS-CoV-2s, all of which have the same non-spike B.1 backbone genes. These 27 recombinant viruses were then compared for relative virus growth in immortalized cells and also in human nasal airway epithelial cultures (hNEC). The recombinant viruses were also compared in hamster pathogenicity tests.

The results support several important claims. Relatively to viruses with pre-omicron spikes, those with post-omicron spikes grow most robustly in hNEC and are generally less pathogenic in hamsters. Pronounced post-omicron growth in hNEC caused cell sloughing, an important finding that may help explain high omicron transmissibility. This adaptation to hNEC and attenuation of virulence is not correlated with spike-mediated preference for particular host proteases and is not correlated with the potency of spike-mediated cell-cell fusion. Focusing on a particular post-omicron BA.1 spike, the adaptation to hNEC has some correlation with evolution of the RBD, although detailed mechanism of this RBD effect remains unknown.

There have been a great many papers on this topic of SARS-CoV-2 spike evolution in relation to spike functions, virus tropism and virus virulence. In these papers, the tests of spike functions parallel those used in this submission, i.e., they test for successful infection in different cell lines, for potency of spike-directed membrane fusion, and for suppression of spike functions by host protease inhibitors. But most of these papers have used separately-expressed spike genes and/or pseudovirus surrogates of SARS-CoV-2 for analyses, and many have stopped short of using hNEC or hHAE for testing spike functions. The strength of this comprehensive paper is with its construction of a very large collection of isogenic replication-competent recombinant SARS-CoV-2 viruses varying only in spike, and with the use of these viruses in physiologically relevant hNEC and in vivo experimental conditions. Therefore, the results in this submission have increased reliability. This submission also makes a few significant discoveries about specific RBD determinants of hNEC infection and hNEC infected-cell sloughing. The comprehensive data provide a platform for future investigations of spike functions necessary for nasal tropism and endemic virus transmissibility.

Comments:

1. The input multiplicities of infection are stated in terms of viral genome equivalents per cell. Can input multiplicities be stated in terms of infectious units per cell? This is asked because readers may want to know whether the growth curves in Figs 2, 4, 5 and some supplementary figs are single-cycle or multi-cycle. It is nearly certain that they are multi-cycle but it would help if input MOIs were stated or at least estimated.

2. Figs 2CD and text lines 168-181: it may not be possible to make strong conclusions about variable spike-mediated cell-cell fusion without knowing the relative levels of spike on plasma membrane surfaces. Also, cell-cell fusion signals may be artificially

high in AAT overexpression cells, and it is not known whether cell-cell fusion is a meaningful feature of hNEC or in vivo SARS-CoV-2 infection and spread. It may be worth stating these limitations.

3. Fig. 3 and Fig 5D: Can the protease inhibitor studies be performed using hNEC? The advantages of the AAT host cells in discerning TMPR vs cathepsin utilization are nicely described but the assumption that findings from AAT can be extended to hNEC conditions might be questioned.

4. Fig 5BC results are truly remarkable in showing RBD substitutions controlling hNEC infection. Minor point, but might the RBD changes be put into spike structural context; to get further toward some hypothetical mechanistic insight into the very pronounced effect of these RBD changes on virus tropism?

Reviewer #2 (Remarks to the Author):

In this carefully presented manuscript (MS), the authors employed a reverse genetics system using the SARS-CoV-2 ancestral strain as the backbone to generate recombinant viruses by replacing the S-protein with those from 27 strains. Using S:B.1 recombinant virus as the reference (SARS-CoV-2 ancestral strain with the G614D Spike). The authors systematically compared the effects of different S-proteins on viral replication, membrane fusion, viral entry pathways, cell tropism, as well as viral virulence and pathogenicity. Additionally, they categorized the strains into pre-Omicron and post-Omicron groups to analyze the phenotypic changes of the recombinant viruses. Overall, the study is comprehensive and the data obtained from the reverse genetics approach are particularly valuable. The study provides interesting new data in understanding the SARS-CoV-2 S-protein evolution.

In general, the work is important and the editor may consider the following issues being clarified before publication:

Major points:

1. In Fig.2A/B, a few of the replication curves between S:B.1 and S:variant are remarkably similar in the left panels (For example S:EG.5.1 in Fig.2B). However, many time points are scored as significantly different in the heat map on the right side panels (For example S:EG.5.1 in Fig.2B). Whether the statistical method used to score the differences robust? Why at or near time 0, there are two data points? Are the first points for virus quantification? If so, why in different curves the first data points have different values? I am not sure if it would be better to show the differences between control and variant curves if the curves are shown in Log₂ scale instead of Log₁₀ scale. The heat-map is shown in Log₂ scale, would it be better to match up? In the legend, the right panel heat maps are defined as "viral loads difference". Is the viral load evaluated the same as the left panels as orf1 gene copy/ml? In addition, not sure why heat maps are not included for Fig.S1?

2. In Fig.3, there is a significant difference in the inhibition of S:BA.1 (A/D) and BA.1(F) by camostat. Several published studies (Willett BJ, et al. Nat Microbiol. 2022. Fig.5G; Emerg Microbes Infect. 2022 Dec;11(1):277-283. Fig.3C; mBio. 2023 Feb 28;14(1):e0317622. Fig.2) have reported that camostat is unable to effectively inhibit BA.1. However, in this study, S:BA.1 appears to be more sensitive to camostat. This is especially true in light of the reference (Willett BJ, et al. Nat Microbiol), in which some authors in this MS also participated, that in AAT cells (presumably the same cell used in this study?), clinically isolated BA.1 variant was not efficiently inhibited by camostat. Can the authors discuss possible reasons?

Does this result imply a difference in the entry mechanism between the recombinant virus and the clinically isolated virus, suggesting that the recombinant Spike-variant only virus may not fully reproduce the infection phenotype of the clinically isolated virus? The authors cited references implying that cell tropism determinants outside of the spike gene are implicated. Based on reverse genetic virus data presented, can the authors confirm such proposal? If not what future experiments are needed?

Also in Fig.3A/B, the data dispersion is rather high, could such experimental features/errors prevent accurate assessment of inhibition by different protease inhibitors?

3. The findings presented in Fig.5 are crucial for mechanistic understanding of key S-protein amino acid residues for the nasal epithelial cell tropism. While the result is already informative, I am surprised that Pastorio et al., 2022, Cell Host & Microbe (PMID: 35931073) is not cited when D339, F371, P373 and F375 were identified to play an important role in nasal cell infection. In Pastorio et al., when F371 and F375, and to a lesser degree P373, mutations were introduced individually in the ancestral S-protein, the spike processing and pseudovirus entry were severely affected. Based on Fig. 5 results, the author somewhat dismiss in the Discussion section earlier proposed correlations between spike processing, fusogenicity, cell entry pathways and tissue tropism. Would it be useful to test phenotypic changes when D339, F371, P373 and F375 are introduced individually or together on B.1 S-protein as recombinant viruses rather than just simply reverting them in the BA.1 S-protein? Also I am not sure if spike processing and fusion data for S:BA.1-wt13, S:BA.1-wt4-RBD, S:BA.1-wt4-RBM and S:BA.1-wt5-S2 are presented?

Please also cite other function of F371, P373 and F375 mutations such as McCallum et al (PMID: 35076256) and possibly others on antibody evasion and He et al (PMID: 38454157) on interference with antigen presentation.

4. If I understand correctly, the authors somewhat hint virus independent factors for tissue tropism, including cell type and expression of ADAM and MMP. Would it be more informative, if at all possible in reasonable time, but also possible in a follow up study, phenotypic characteristics in a few more selected cell lines tested, for example Caco-2 and Vero-TMPRSS2 cells used

in Pastorio et al and Kimura et al to enable direct comparison between studies?

Minor points:

1. Line 60, please unify spelling of ACE2.
2. Line 205-206 may not be technically accurate. Cathepsin L sites have been reported in S1 for SARS-CoV-2 (PMID: 34548480, PMID: 35668062) and near the S1/S2 site in both SARS-CoV-1 (PMID: 18562523) and SARS-CoV-2 (PMC7255728). Is there evidence to show that S2' site is cut by Cathepsin in SARS-CoV-2?
3. In Fig.2A/B, there are certain differences in the growth curves of S:B.1 when compared with different recombinant strains. For instance, in Fig.2B S:BA.2.75, the gray dashed region is noticeably lower than the corresponding regions in other strains. As a control, S:B.1's proliferation curve should ideally remain consistent across different experimental groups. Please explain the reason for this phenomenon.
4. Some data were labeled for cell line used but not all. Please label the cell line corresponding to each part of the data consistently throughout the MS, including the ones in Fig.4 and Fig 5B/C.
5. Line 178-179 "Importantly, S:EG.5.1 and S:BA.2.86 were as fusogenic, if not more, than S:B.1 (Figure 2D)"
Kei Sato found that BA.2.86 induced fusion significantly lower than EG.5.1, approaching BA.2 (Tamura et al., 2024, Cell Host & Microbe 32, 1–11); Olivier found that BA.2.86 Spike induced fusion ability was lower than XBB, much lower than D614G (Nature Communications (2024) 15:2254); these results differ from those in this article. Could the authors discuss possible reasons for different observations among different studies?
6. Line 183-190, should "Figure S1B" be "Figure S1D"?
7. Line 233-235, The use of only one lineage A strain in this experiment may not be sufficient to conclusively assert that the entire A lineage exclusively employs the endosomal pathway? Additionally, since recombinant viruses were used in this study, and cell entry may be dependent on spike independent factors as implied by the author, whether further investigation is required to determine if clinical lineage A strains still exhibit this characteristic?
8. Line 611 "infected 5 days later with the equivalent of 104 Orf1a genome copies per well in serum free RPMI-1640 medium." Why did the authors quantify the virus using Orf1a genome copies instead of using multiplicity of infection (MOI) or TCID50? Have the authors confirmed that virus genome copy number correlates well with the virus infectivity across different strains? Will there be interference from defective viral particles?

Reviewer #3 (Remarks to the Author):

Furnon et al perform an extensive study of many of the SARS-CoV-2 variants that emerged during the COVID-19 pandemic. Infectious isogenic recombinant and original strains of virus were used to infect a variety of cell lines and primary cells. The results are in general agreement with previous studies, demonstrating that the virus has continued to evolve to better infect nasal tissue, and by extension, become more transmissible. In addition, some of the newer variants, in addition to better infecting nasal tissue, also have regained some of the features of early variants associated with the earlier, more virulent variants (cell surface vs. endosomal entry, fusogenicity, etc). The data are convincing and are more comprehensive than previous studies. The manuscript is well written. However, the amount of new information is limited, other than the data in Figure 5. Almost all of the conclusions were made in previous publications, albeit in studies of a smaller set of variants.

Specific comments.

1. Line 302, Figure 4E-Bronchial epithelia cells were used. Presumably these are primary cells. Their origin should be described in the Materials and Methods. Studies using primary human bronchial cells infected with a subset of the pre-Omicron and post-Omicron isolates should also be performed to support the conclusions from the studies using Calu-3 cells. Alternatively, published reports that use primary cells could be included to support the lung cell conclusions.
2. Page 13, Figure 5-The data in this figure are interesting. Specifically, the effects of changes in the non-RBM part of the RBD are striking in their effects of virus replication in primary nasal epithelial cells. This part of the manuscript should be further developed.
-The authors show that the mutations have no effects on growth in Calu3 cells. Were any effects observed on growth in primary bronchial cells (HAEs)?
-Information about S protein processing (S1-S2 cleavage; S2' cleavage) or fusogenicity in nasal epithelial cells might provide insight into the basis of these differences.
3. Line 387, Figure 7-Levels of infectious virus or genomic RNA in hamster lungs should be included in these analyses.
4. Figure 7, line 352 ("...potentially enhancing viral transmission...")-Since SARS-CoV-2-infected hamsters are a very useful model for studying virus transmission, transmission should be directly analyzed in infected hamsters. Alternatively, it could be discussed since some transmission data with the different isolates have previously been published.

Minor comments.

Line 184-Should be Figure S1D, not S1B.

Line 322-(iv) S:BA.1-wt-RBD possesses the 4 B.1 mutations in the RBD. This is not accurate because the RBD also contains all of the mutations present in the RBM.

Version 2:

Reviewer comments:

Reviewer #1

(Remarks to the Author)

This reviewer considers the authors' revisions and additions to be satisfactory. The construction and evaluation of additional recombinant viruses (Fig 5) strengthen the suggestions that specific Omicron RBD changes profoundly affect virus biology.

The works in this paper are comprehensive and informative. The manuscript shows that evaluations of authentic recombinant virus infections into physiologically relevant *ex vivo* cultures bring meaningful results that provide the platforms for future mechanistic studies of virus entry and tropism.

Reviewer #2

(Remarks to the Author)

The authors have adequately addressed most of my concerns. There are only 2 remaining issues:

1. Heatmap is still missing in Fig. S1, as far as I can see.
2. Please cite the latest publication regarding the S371-S373-S375 mutations implicated in the immune evasion of population antibodies. (<https://doi.org/10.1038/s41467-024-51770-3>).

Reviewer #3

(Remarks to the Author)

The authors have responded well to the comments of this reviewer. The authors nicely show that the SARS-CoV-2 S protein is not mutating in a directional manner when assessed phenotypically. The data showing virus replication in NEC continues to a strength of the manuscript.

Decision Letter:

Our ref: NMICROBIOL-24041047B

8th October 2024

Dear Dr. Palmarini,

Thank you for submitting your revised manuscript "Phenotypic evolution of SARS-CoV-2 spike during the COVID-19 pandemic" (NMICROBIOL-24041047B). Please accept my sincere apologies for the delay in providing you with a decision due to several absences in the editorial team. In addition, [redacted] has moved to support another journal in our family and I am handling your manuscript moving onwards. Nonetheless, it has now been seen by the original referees and their comments are below. The reviewers find that the paper has improved in revision, and therefore we'll be happy in principle to publish it in Nature Microbiology, pending minor revisions to satisfy the referees' final requests and to comply with our editorial and formatting guidelines.

We are now performing detailed checks on your paper and will send you a checklist detailing our editorial and formatting requirements in several weeks. Please do not upload the final materials and make any revisions until you receive this additional information from us.

Thank you again for your interest in Nature Microbiology Please do not hesitate to contact me if you have any questions.

Sincerely,

Reviewer #1 (Remarks to the Author):

This reviewer considers the authors' revisions and additions to be satisfactory. The construction and evaluation of additional recombinant viruses (Fig 5) strengthen the suggestions that specific Omicron RBD changes profoundly affect virus biology.

The works in this paper are comprehensive and informative. The manuscript shows that evaluations of authentic recombinant

virus infections into physiologically relevant ex vivo cultures bring meaningful results that provide the platforms for future mechanistic studies of virus entry and tropism.

Reviewer #2 (Remarks to the Author):

The authors have adequately addressed most of my concerns. There are only 2 remaining issues:

1. Heatmap is still missing in Fig. S1, as far as I can see.
2. Please cite the latest publication regarding the S371-S373-S375 mutations implicated in the immune evasion of population antibodies. (<https://doi.org/10.1038/s41467-024-51770-3>).

Reviewer #3 (Remarks to the Author):

The authors have responded well to the comments of this reviewer. The authors nicely show that the SARS-CoV-2 S protein is not mutating in a directional manner when assessed phenotypically. The data showing virus replication in NEC continues to a strength of the manuscript.

Version 3:

Decision Letter:

11th November 2024

Dear Massimo,

I am pleased to accept your Article "Phenotypic evolution of SARS-CoV-2 spike during the COVID-19 pandemic" for publication in *Nature Microbiology*. Thank you for having chosen to submit your work to us and many congratulations.

Over the next few weeks, your paper will be copyedited to ensure that it conforms to *Nature Microbiology* style. We look particularly carefully at the titles of all papers to ensure that they are relatively brief and understandable.

Once your paper is typeset, you will receive an email with a link to choose the appropriate publishing options for your paper and our Author Services team will be in touch regarding any additional information that may be required. Once your paper has been scheduled for online publication, the *Nature* press office will be in touch to confirm the details.

Please note that *Nature Microbiology* is a Transformative Journal (TJ). Authors may publish their research with us through the traditional subscription access route or make their paper immediately open access through payment of an article-processing charge (APC). Authors will not be required to make a final decision about access to their article until it has been accepted. [Find out more about Transformative Journals](https://www.springernature.com/gp/open-research/transformative-journals)

Authors may need to take specific actions to achieve [compliance](https://www.springernature.com/gp/open-research/funding/policy-compliance-faqs) with funder and institutional open access mandates. If your research is supported by a funder that requires immediate open access (e.g. according to [Plan S principles](https://www.springernature.com/gp/open-research/plan-s-compliance)) then you should select the gold OA route, and we will direct you to the compliant route where possible. For authors selecting the subscription publication

route, the journal's standard licensing terms will need to be accepted, including [self-archiving policies](https://www.nature.com/nature-portfolio/editorial-policies/self-archiving-and-license-to-publish). Those licensing terms will supersede any other terms that the author or any third party may assert apply to any version of the manuscript.

Congrats again to you and your co-authors! I am looking forward to seeing your paper published.

With kind regards,

P.S. Click on the following link if you would like to recommend Nature Microbiology to your librarian <http://www.nature.com/subscriptions/recommend.html#forms>

** Visit the Springer Nature Editorial and Publishing website at http://editorial-jobs.springernature.com?utm_source=ejP_NMicro_email&utm_medium=ejP_NMicro_email&utm_campaign=ejp_NMicro for more information about our career opportunities. If you have any questions please click [here](mailto:editorial.publishing.jobs@springernature.com).**

[redacted]

RE: Revision of NMICROBIOL-24041047A “Phenotypic evolution of SARS-CoV-2 spike throughout the COVID-19 pandemic” by Furnon, Cowton *et al*

On behalf of my co-authors I would like to thank you and the Reviewers for the supportive and insightful suggestions to our manuscript. We agree essentially with most, if not all, the suggestions made. The new data presented clarify and extend some of the messages provided in the original submission.

We have carried out a series of additional experiments, as suggested by the reviewers, that are now embedded in the revised manuscript. In particular we have:

- i. Carried out new replication assays in primary human broncho-epithelial cells;
- ii. Carried out protease sensitivity assays in primary human nasal cells;
- iii. Generated additional mutants (in order to have both mutants in the BA.1 and B.1 backbone); carried out with those mutants replication assays in primary nasal cells and Calu-3, in addition to fusion and protease sensitivity assays;
- iv. Added data related to virus load to the *in vivo* data.
- v. Added the “Data availability” section in our Methods and created a public data repository with all our raw data.
- vi. Shortened the Abstract within the word limits.

We will respond below, as requested, to each of the points made by the reviewers. To facilitate the editorial process we have pasted directly in this rebuttal letter (*in red*) each comment made by the reviewers. We have also copied in this letter the major pieces of text inserted in the revised manuscript.

Reviewer #1:

We thank Reviewer 1 for the supportive comments.

The reviewer provided the following specific comments/suggestions:

1. The input multiplicities of infection are stated in terms of viral genome equivalents per cell. Can input multiplicities be stated in terms of infectious units per cell? This is asked because readers may want to know whether the growth curves in Figs 2, 4, 5 and some supplementary figs are single-cycle or multi-cycle. It is nearly certain that they are multi-cycle but it would help if input MOIs were stated or at least estimated.

These are multi-cycle replication assays as the reviewer correctly stated. As mentioned in the Methods section, we equalised virus input in the various assays measuring Orf1a gene copies by RT-qPCR. The same method was also used to assess viral replication kinetics in the cells used in the study. To clarify, the method we used is necessary as both we and others have found that the different SARS-CoV-2 variants replicate with different kinetics in different cells lines (including cells normally used to titrate viruses such as VERO and BHK) and use different mechanisms of cell entry. Hence, our approach normalises virus titres independently of variant-specific differences in cell tropism or entry routes which have an impact on plaque assays or TCID₅₀ measurements. We and others have already used this method in published studies (*eg*, Reuschl *et al*, 2024 *Nature Micro* www.nature.com/articles/s41564-023-01588-4; Thorne *et al*, 2024 *Nature* www.nature.com/articles/s41586-021-04352-y). To provide an example to the reviewer, for some

of our own viruses used in this study the relationship between genome copies and PFU titres are the following:

S:B.1 = 1.62E+09 Orf1a gene copies/mL = 1.85E+06 PFU/mL in VeroE6 cells

Omicron BA.1 = 1.35E+09 Orf1a gene copies/mL = 1.60E+05 PFU/mL in VeroE6 cells

S:Omicron BA.1 = 1.42E+09 Orf1a gene copies/mL = 4.10E+05 PFU/mL in VeroE6 cells

We have added in the Methods of the resubmitted manuscript (**from line 690**) this sentence to clarify this point and added relevant references.

“Multi-cycle replication assays were carried out in different cell lines by equalising virus input measuring Orf1a gene copies by RT-qPCR as described below. The same method was also used to assess viral replication kinetics in the cells used in the study as indicated below. This method, as opposed to classic PFU/endpoint dilution analysis is necessary as both we and others have found that the different SARS-CoV-2 variants replicate with different kinetics in different cells lines (including those cells normally used to titrate viruses such as VERO and BHK) and use different mechanisms of cell entry.^{73,93,94} Hence, our approach normalises virus titres independently of variant-specific differences in cell tropism or entry routes which have an impact on plaque assays or TCID50 measurements.”

2. Figs 2CD and text lines 168-181: it may not be possible to make strong conclusions about variable spike-mediated cell-cell fusion without knowing the relative levels of spike on plasma membrane surfaces. Also, cell-cell fusion signals may be artificially high in AAT overexpression cells, and it is not known whether cell-cell fusion is a meaningful feature of hNEC or in vivo SARS-CoV-2 infection and spread. It may be worth stating these limitations.

We take on board the comment and mostly agree with the reviewer. We have added this sentence in the Results section (**from line 179**) in the originally submitted manuscript *“It is however important to note that fusion assays in vitro in AAT cells do not necessarily recapitulate all the features of the respiratory epithelium in vivo during SARS-CoV-2 infection.”*

The levels of spike at the surface we feel are less relevant. They are important of course for fusion, but from a biological standpoint our fusion assays are based on the full virus replication cycle, as opposed to transfection of spikes. Consequently, if the levels of spike at the surface were to vary between different variants, these would be reflective of biological differences in the viral replication cycles.

3. Fig. 3 and Fig 5D: Can the protease inhibitor studies be performed using hNEC? The advantages of the AAT host cells in discerning TMPR vs cathepsin utilization are nicely described but the assumption that findings from AAT can be extended to hNEC conditions might be questioned.

This is a good point. We have added a new Panel E in Figure 3 where we carried out protease assays with a selection of pre-Omicron and post-Omicron spikes in reconstituted nasal epithelium (hNEC). Although the degree of inhibition of various recombinant spike varies between assays, and between cell lines, as one would expect, we have identified clear patterns and take home messages.

First, we highlight the following commonalities. Both the pre-Omicron S:B.1 and S:Delta are in general sensitive to Camostat but relatively insensitive to E64d, in both AAT, Calu-3 and hNECs. The post-Omicron BA.1 is sensitive to both Camostat and E64d in both AAT and hNECs. Also the patterns of relative sensitivity to Camostat and E64d for S:BA4/5 and S:BA.2.86 are similar in both hNECs and AAT.

We found instead a clear difference between drug sensitivity to Camostat for S:A.30, which has a similar pattern to the other pre-Omicron recombinant viruses.

We have added these data in Fig. 3, panel E with the Results described **from line 248**:

“Human nasal epithelium cells (hNEC) express both TMPRSS2 and cathepsin L, and, therefore, may support both pathways³⁹. In hNEC, pre-Omicron recombinants including S:A.30 displayed consistent profiles, being strongly inhibited by Camostat (Figure 3E), but insensitive E64d. In contrast, the post-Omicron S:BA.1, S:BA.4/5 and S:BA.2.86 showed a variable level of sensitivity to both drugs, but with patterns relatively similar to AAT cells; for example, both BA.1 and BA.4/5 exhibited E64d sensitivity. Collectively, these data suggest that entry pathway preference has complex determinants that rely on both spike and cellular context.”

We also highlighted in the discussion, that in hNECs in our study the recombinant virus with spike BA:1 was sensitive to Camostat, while in another recently published study (<https://doi.org/10.1038/s41467-024-45075-8>) at least full length BA.1 was insensitive to both Camostat and E64d. We added the following sentence (**from line 558**):

“Of note, in this study S:BA.1 replication in primary nasal cells was found to be very sensitive to Camostat (with little or no variability between different biological replicates), while in a previous study full length BA.1 was found to be insensitive to both Camostat and E64d⁷³. This discrepancy may be due to differences in the experimental approach between the two studies, or due to variability between batches of human primary nasal cells.”

4. Fig 5BC results are truly remarkable in showing RBD substitutions controlling hNEC infection. Minor point, but might the RBD changes be put into spike structural context; to get further toward some hypothetical mechanistic insight into the very pronounced effect of these RBD changes on virus tropism?

We thank the reviewer for this excellent suggestion. We have added a number of structural model figure panels to complement the presentation and interpretation of this interesting data. Figure 5A now features a full spike model, displaying the location of all 13 mutated residues. Figure S6B includes a similar model, but focussed on the RBD-4 mutations. Figure S6C and D pertains to the proposed role of the RBD-4 mutations in interprotomer RBD-packing (as described by Gobeil et al. PMID: 35447081), which may offers some structure-to-function context for our findings. We have also added the following text (**from line 362**):

“Hence, these Omicron-specific RBD mutations profoundly affect spike biology. These residues are juxtaposed on short loops extending from the RBD; these are on the opposite face of RBD to the loops that constitute the RBM (Figure S6C). Previous cryoEM analysis indicates that the F371, P373 and F375 mutations modulate inter-protomer RBD-RBD packing⁸³, for instance by mediating interaction with the RBM of the adjacent protomer (e.g. H505-P373; Figure S6D and E). It is likely that this will affect the conformational plasticity of the RBD (e.g. up/down switching), which in turn may impact ACE2 acquisition and spike stability.”

Reviewer #2:

We thank Reviewer 2 for the supportive comments.

Major points:

1. In Fig.2A/B, a few of the replication curves between S:B.1 and S:variant are remarkably similar in the left panels (For example S:EG.5.1 in Fig.2B). However, many time points are scored as significantly different in the heat map on the right side panels (For example S:EG.5.1 in Fig.2B). Whether the statistical method used to score the differences robust?... I am not sure if it would be better to show the differences between control and variant curves if the curves are shown in Log2

scale instead of Log10 scale. The heat-map is shown in Log2 scale, would it be better to match up?

We have tried different solutions to find a compromise between showing the entire replication curves including also the virus input, the differences in the areas under the curves, but also highlighting the small but significant differences and the overall patterns. The stats are solid and use common methods (indeed two different methods followed for AUC and individual timepoints comparisons), but most importantly all the data in Fig. 2 were obtained using three biological replicates, each with three technical replicates. Moreover, viruses used in each series were derived from two distinct stocks/reverse genetic preps. Hence, we are confident of the differences observed not simply because of their p-values, but also for the robust experimental design followed.

In the revised manuscript, we have deleted in Fig. 2A-B the 72h timepoint in the heatmaps as it is distracting for the reader given that this timepoint has limited meaning because many viruses have already reached their plateau. In addition, we have made a supplementary figure (Supplementary Fig 1) where the same data presented in Figure 2A-B are reproduced using larger graphs, and reduced range of gene copies/ml (Log₁₀) values in the Y axis. In addition, only the replicative phase, starting at 18 hours post-infection, is shown.

In the legend, the right panel heat maps are defined as “viral loads difference”. Is the viral load evaluated the same as the left panels as orf1 gene copy/ml?

Apologies for using a confusing term. We corrected the legend (used the term “virus titres”). Values are indeed the same for both the replication curves and heatmaps.

Why at or near time 0, there are two data points? Are the first points for virus quantification? If so, why in different curves the first data points have different values?

We have now clarified this point in the legend of Fig. 2 as follows:

“Note, that values at time 0 reflect the titres of the virus input, while at 1hpi these are the titres of residual virus in the supernatant after 1 hour infection and one wash of the monolayer.”

In addition, not sure why heat maps are not included for Fig.S1?

Apologies, we have now added a heatmap in Fig S1.

2. In Fig.3, there is a significant difference in the inhibition of S:BA.1 (A/D) and BA.1(F) by camostat. Several published studies (Willett BJ, et al. Nat Microbiol. 2022. Fig.5G; Emerg Microbes Infect. 2022 Dec;11(1):277-283. Fig.3C; mBio. 2023 Feb 28;14(1):e0317622. Fig.2) have reported that camostat is unable to effectively inhibit BA.1. However, in this study, S:BA.1 appears to be more sensitive to camostat. This is especially true in light of the reference (Willett BJ, et al. Nat Microbiol), in which some authors in this MS also participated, that in AAT cells (presumably the same cell used in this study?), clinically isolated BA.1 variant was not efficiently inhibited by camostat. Can the authors discuss possible reasons?

We thank the reviewer – this is an important point to clarify. Principally this is a live virus versus pseudotype-based assay difference. When considering live virus data alone BA1 is consistent. We showed consistently relative sensitivity of BA.1 (and S:BA.1) to Camostat in all the virus replication assays presented in this study. In the resubmitted manuscript we show data in three cell types (AAT, Calu-3 and primary nasal cells/hNEC; new Fig. 3 panels A, D and E). This is also

consistent with the data in Willett et al where we used live virus replication assays (Extended Data Fig. 8, copied below for simplicity).

However the reviewer is correct in identifying the apparent difference in the sensitivity of the BA.1 spike to Camostat between the original Fig. 3A,D and Fig. 3F (now G in the resubmitted manuscript). The key difference between these data is that the assays in Fig. 3A,D were carried out with replication-competent live recombinant viruses, while those shown in Fig 3F were obtained using a replication-defective lentivirus-based BA.1 spike pseudovirus (PV). This discrepancy again was also shown in Willett et al.

The reviewer also refers to an apparent BA.1 spike-related disparity between our data and those reported in Zhao et al *Emerg Microbes Infect.* 2022 Dec;11(1):277-283. Fig.3C; Qu et al *mBio.* 2023 Feb 28;14(1):e0317622. Fig.2. Both studies indeed show reduced Camostat sensitivity of BA.1 spike, however, the assay conditions used are different from ours which may explain the apparent discrepancy. Firstly, both studies use Vero-derived cell line ectopically expressing either Tmprss2 or Ace 2 and Tmprss2 to test the inhibitor against live BA.1 virus. Furthermore, the Qu et al paper also used the PV model.

While PVs have been used widely and successfully to study virus entry events, they are ultimately lentiviral core particles decorated with spikes and therefore do not necessarily fully emulate the real SARS-CoV-2 virion. First, spike is in isolation without other components of the viral envelope. Secondly, the spike in PVs is C-terminally truncated and it sits on a surrogate lentiviral core, and as such its structural presentation and density/copy number is almost certainly different than in the real virions. PVs are also replication-incompetent and therefore unable to spread between cells. Finally, use of different cell types may also impact on results. Particularly noteworthy in this respect is cell type-related differences (in this case Vero cells versus all the other cell types AAT, Calu-3, hNEC) in sensing of host innate responses which could affect assay outcomes. To better present these data to the reader, we have added the label “Pseudoviruses” in Panel 3G (F of the original submission). In addition, we have added in the Results the following sentence (from line 279):

“Of note, both us and others have reported discrepancies in Camostat sensitivity for BA.1 spike depending on the assay. BA.1 is sensitive to Camostat in live virus replication assays (such as in Figure 3A-E) compared to pseudotype assays using lentiviruses bearing the BA.1 spike (Figure 3G)^{39,61}. This may reflect inherent differences in the assay-dependent context of spike (i.e. authentic SARS-2 vs lentiviral particles), or be a product of multi-cycle (live virus) vs single round (PV) infections. It is likely that the authentic virus system is a more faithful representation of the underlying biology.”

Does this result imply a difference in the entry mechanism between the recombinant virus and the clinically isolated virus, suggesting that the recombinant Spike-variant only virus may not fully reproduce the infection phenotype of the clinically isolated virus? The authors cited references implying that cell tropism determinants outside of the spike gene are implicated. Based on reverse

genetic virus data presented, can the authors confirm such proposal? If not what future experiments are needed?

As explained above, we think that differences are more related to the type of assays used.

Also in Fig.3A/B, the data dispersion is rather high, could such experimental features/errors prevent accurate assessment of inhibition by different protease inhibitors?

We appreciate the reviewer's concern. We attribute the variability observed in this assay to the way in which even small experimental perturbations (e.g. virus batch) can be propagated across the multicycle replication of authentic virus. Indeed, the most variable datapoints in Figure 3A-E are the samples with moderate or no inhibition i.e. those in which successful replication is most able to amplify natural variation. This is exacerbated by linear scaling, which is best-suited for viewing the effects of these inhibitors. Nonetheless, we maintain that where strong inhibition occurs we see high reproducibility and there are clear and consistent patterns in the data (e.g. the relative E64d insensitivity of the pre-omicron viruses). We also note that we have been careful to carry out the assays in at least three biological replicates (or four if we saw more variability) and make sure that the graphs in the figures are transparent to the reader in showing the values of individual biological replicates.

2. The findings presented in Fig.5 are crucial for mechanistic understanding of key S-protein amino acid residues for the nasal epithelial cell tropism. While the result is already informative, I am surprised that Pastorio et al., 2022, Cell Host & Microbe (PMID: 35931073) is not cited when D339, F371, P373 and F375 were identified to play an important role in nasal cell infection. In Pastorio et al., when F371 and F375, and to a lesser degree P373, mutations were introduced individually in the ancestral S-protein, the spike processing and pseudovirus entry were severely affected. Based on Fig. 5 results, the author somewhat dismiss in the Discussion section earlier proposed correlations between spike processing, fusogenicity, cell entry pathways and tissue tropism.

We apologise for missing this reference. We have now referenced the Pastorio et al paper and inserted the following text **from line 361** in the Results:

“Overall, these data suggest a critical role for residues D339, F371, P373 and F375 in the spike RBD, for the enhanced viral replication of the post-Omicron variants in hNEC. Interestingly, another study showed that mutations in the serine residues 371, 373 and 375 in B.1, both alone or in combination, affect infectivity of VSV (vesicular stomatitis virus)-based pseudoviruses in the CaCo-2 cell line⁷⁹.”

From line 382: *“A previous study using VSV pseudotypes also found that mutations in residues 371, 373 and 375 affected B.1 cell-cell fusion⁷⁹”*

We would, respectfully, note that the Pastorio et al. study does not address nasal cell infection and, therefore, does not precedent our work in this respect.

Would it be useful to test phenotypic changes when D339, F371, P373 and F375 are introduced individually or together on B.1 S-protein as recombinant viruses rather than just simply reverting them in the BA.1 S-protein? Also I am not sure if spike processing and fusion data for S:BA.1-wt13, S:BA.1-wt4-RBD, S:BA.1-wt4-RBM and S:BA.1-wt5-S2 are presented?

We thank the reviewer for the suggestion. We have now generated and analysed both mutants in the B.1 and BA.1 spike context. We compared their replication kinetics in Calu-3 and primary nasal cells, fusion and protease sensitivity. We obtained very interesting results reverting the fusion phenotype of the BA.1 spike with only 5 amino acid residues substitutions in the S2 region. The

revised manuscript contains an extended Fig. 5 with the new data. Consequently these new data are described in the Results, section “Spike amino acid residues determining nasal epithelial cell tropism” which has been extensively rewritten (from line 329 to line 397).

Please also cite other function of F371, P373 and F375 mutations such as McCallum et al (PMID: 35076256) and possibly others on antibody evasion and He et al (PMID: 38454157) on interference with antigen presentation.

We have added these references and linked the immunoevasive properties of these residues to the previous sentence on the data by Pastorio et al (from line 361) “*Importantly, the same mutations also have been reported to contribute to immune evasion*^{80,81}”

4. *If I understand correctly, the authors somewhat hint virus independent factors for tissue tropism, including cell type and expression of ADAM and MMP. Would it be more informative, if at all possible in reasonable time, but also possible in a follow up study, phenotypic characteristics in a few more selected cell lines tested, for example Caco-2 and Vero-TMPRSS2 cells used in Pastorio et al and Kimura et al to enable direct comparison between studies?*

We have added experiments in human primary cells to complement what we had presented in the original submission. The goal and strength of the manuscript is to phenotypically characterise a variety of spike variants in a single study. It would be very difficult to carry out some of these assays in all the cells used in different studies. For example, there are no strong arguments to support the biological relevance of either Caco-2 or Vero-TMPRSS2 in this specific time of the SARS-CoV-2 field.

Minor points:

1. Line 60, please unify spelling of ACE2.

Text corrected, thank you.

2. Line 205-206 may not be technically accurate. Cathepsin L sites have been reported in S1 for SARS-CoV-2 (PMID: 34548480, PMID: 35668062) and near the S1/S2 site in both SARS-CoV-1 (PMID: 18562523) and SARS-CoV-2 (PMC7255728). Is there evidence to show that S2' site is cut by Cathepsin in SARS-CoV-2?

We appreciate the comment of the reviewer, and agree that there is no direct evidence of S2' processing by cathepsin. There is, however, reasonable evidence for the role of endosomal protease, for example Yu et al. (PMID: 34930824) demonstrate ACE2-primed cleavage at S2' occurring in HEK 293T cells, which do not express TMPRSS2; it is likely that an endosomal protease is performing this function, but it has not been formally demonstrated to be a cathepsin. To reflect this nuance, and considering that the manuscript is not focused at all on the localisation of spike cleavage, we have de-specified two statements by removing direct reference to S2' (**From line 204**):

“SARS-CoV-2 can utilize two entry routes, both involving membrane fusion but occurring either at the cell surface (route 1) or within endosomes (route 2), following proteolytic processing of the spike by proteases such as TMPRSS2 or cathepsins, respectively^{2,66-68}. Omicron BA.1 and its sub-lineages, unlike the pre-Omicron variants, have a reduced dependency on TMPRSS2, and a preference for alternative activating proteases such as endosomal cathepsins^{39,61,62}.”

3. In Fig.2A/B, there are certain differences in the growth curves of S:B.1 when compared with different recombinant strains. For instance, in Fig.2B S:BA.2.75, the gray dashed region is

noticeably lower than the corresponding regions in other strains. As a control, S:B.1's proliferation curve should ideally remain consistent across different experimental groups. Please explain the reason for this phenomenon.

The replication curves shown in Fig. 2A-B were carried out over a number of weeks/ months as they represent three true biological replicates with 26 different viruses. This means that any given experiment only some of the recombinant viruses were assessed. However, S:B.1 was used as control and comparator in every single experiment. Hence, the grey dashed region of for example the S:BA.2.75 graph will not be identical to the one of S:BA.2.86 as these were obtained in different experiments.

We have clarified this point in the resubmission in the legend of Fig. 2.

4. Some data were labeled for cell line used but not all. Please label the cell line corresponding to each part of the data consistently throughout the MS, including the ones in Fig.4 and Fig 5B/C.

We have added the labels as suggested.

5. Line 178-179 “Importantly, S:EG.5.1 and S:BA.2.86 were as fusogenic, if not more, than S:B.1 (Figure 2D)”

Kei Sato found that BA.2.86 induced fusion significantly lower than EG.5.1, approaching BA.2 (Tamura et al., 2024, Cell Host & Microbe 32, 1–11); Olivier found that BA.2.86 Spike induced fusion ability was lower than XBB, much lower than D614G (Nature Communications (2024) 15:2254); these results differ from those in this article. Could the authors discuss possible reasons for different observations among different studies?

The reviewer is correct in identifying the discrepancies between the fusogenicity profiles of BA.2.86 and EG.5.1 in our manuscript and the Tamura et al. 2024 and Planas et al. 2024 studies. In these studies, the assay conditions used to measure cell-cell fusion are different from ours. Firstly, in Tamura et al. 2024 and Planas et al. 2024, the authors measure fusogenicity following the transfection of VOC spikes alone, whereas we measured cell-cell fusion following viral infection, which we believe to be more biologically relevant. Although all authors used dual-split protein reporter systems to measure fusogenicity, Planas et al. 2024 quantified the total area of fused cells, Tamura et al. 2024 measured total luciferase activity per well as normalised to the level of surface Spike protein, and we quantified total GFP signal per well as normalised to the fusion of S:B.1. Finally, the studies used different cell-lines. In our assays, we used A549 cells overexpressing ACE-2 and TMPRSS2, while Planas et al. 2024 used IGROV-1 cells, VERO E6 and VERO TMP-1 cells, and Tamura et al., 2024 used Calu-3 cells.

We amended the text in the Results **from line 173** as follows:

“Importantly, S:EG.5.1 and S:BA.2.86 were as fusogenic, if not more, than S:B.1 (Figure 2D) in contrast to previous reports which observed the opposite phenotype, albeit using transient spike over-expression assays (and different cells)^{33,65}, as opposed to authentic viral infection.”

6. Line 183-190, should “Figure S1B” be “Figure S1D”?

We have corrected the references also in the context of the heatmaps added in the revised manuscript.

7. Line 233-235, The use of only one lineage A strain in this experiment may not be sufficient to conclusively assert that the entire A lineage exclusively employs the endosomal pathway? Additionally, since recombinant viruses were used in this study, and cell entry may be dependent on spike independent factors as implied by the author, whether further investigation is required to determine if clinical lineage A strains still exhibit this characteristic?

This is a good point. We have toned down this statement as follows (**from line 233**).

“These data suggest that the potential to use the endosomal pathway may have evolved independently before the emergence of Omicron.”

8. Line 611 “infected 5 days later with the equivalent of 104 Orf1a genome copies per well in serum free RPMI-1640 medium.”

Why did the authors quantify the virus using Orf1a genome copies instead of using multiplicity of infection (MOI) or TCID50? Have the authors confirmed that virus genome copy number correlates well with the virus infectivity across different strains? Will there be interference from defective viral particles?

We refer to the response given to Reviewer # 1, point 1 and the references provided.

Reviewer #3:

We thank the reviewer for the supportive review.

Specific comments.

1. Line 302, Figure 4E-Bronchial epithelia cells were used. Presumably these are primary cells. Their origin should be described in the Materials and Methods. Studies using primary human bronchial cells infected with a subset of the pre-Omicron and post-Omicron isolates should also be performed to support the conclusions from the studies using Calu-3 cells. Alternatively, published reports that use primary cells could be included to support the lung cell conclusions.

We agree and have carried out the experiments using bronchial epithelial cells as suggested with a selection of pre-Omicron and post-Omicron spikes (New Fig 4E).

We found in these cells a similar phenotype to what we found in nasal cells. Please note however that hBECs derive from bronchial epithelial cells and not alveolar type 2 pneumocytes, which are the cells originating Calu-3.

These data are in accord with Hui et al (2022) (<https://www.nature.com/articles/s41586-022-04479-6>) but somewhat different from those presented by Barut et al. In the latter however, the authors have derived these cells from lung slices and therefore results are not directly comparable.

As mentioned above, data is presented in Fig. 4E-F. The following text was added in the Results section (**from line 317**).

“We also tested a subset of pre-Omicron and post-Omicron recombinant viruses for their replication in human primary reconstituted bronchial epithelium (hBEC) (Figure 4E and 4F). Similarly to what was observed in the nasal epithelium, the post-Omicron recombinant viruses S:BA.1, S:BA.4/5 and S:BA.2.86 displayed higher replication kinetics than the pre-Omicron S:B.1, S:Alpha and S:Delta also in hBEC. These data are in accord with a previous report showing the Omicron variant to replicate better in the bronchi compared to the Alpha, Beta and Delta variants⁷⁵. Of note, another study showed the opposite phenotype for Omicron, although in the latter bronchial cells were generated from precision cut lung slices unlike in this study⁷².”

2. Page 13, Figure 5-The data in this figure are interesting. Specifically, the effects of changes in the non-RBM part of the RBD are striking in their effects of virus replication in primary nasal epithelial cells. This part of the manuscript should be further developed. The authors show that the mutations have no effects on growth in Calu3 cells. Were any effects observed on growth in primary bronchial cells (HAEs)? Information about S protein processing (S1-S2 cleavage; S2' cleavage) or fusogenicity in nasal epithelial cells might provide insight into the basis of these differences.

We agree with the reviewer and as highlighted to Reviewer # 2 (point 2) we have made reciprocal mutants in the BA.1 spike backbone and carried out all the mutants in replication assays in Calu-3, fusion assays, and drug sensitivity assays. Please see response to Reviewer # 2, point 2

3. Line 387, Figure 7-Levels of infectious virus or genomic RNA in hamster lungs should be included in these analyses.

We have added PCR data for oral swabs collected at day 2 (now in Figure 7F). This study focused on the development of lung pathology with the different recombinant viruses. As such hamsters were killed at day 6 post-infection. We don't routinely quantify virus at this timepoint as we find this value not representative of active viral replication in the lungs. We have published previously that at day 6 there is very little virus that can be detected by immunohistochemistry or in situ hybridisation (with either Delta, Omicron/BA.1 or BA.5 (e.g see Fig. in 1 <https://doi.org/10.1371/journal.ppat.1011589>)).

4. Figure 7, line 352 ("...potentially enhancing viral transmission...")-Since SARS-CoV-2-infected hamsters are a very useful model for studying virus transmission, transmission should be directly analyzed in infected hamsters. Alternatively, it could be discussed since some transmission data with the different isolates have previously been published.

We are interested in looking at transmission but this would be truly a major study by itself. These studies would also be complicated by the fact that BA.1 does not seem to transmit efficiently in hamsters (doi: 10.1371/journal.ppat.1010970).

Minor comments.

Line 184-Should be Figure S1D, not S1B.

Thank you. This has now been corrected.

Line 322-(iv) S:BA.1-wt-RBD possesses the 4 B.1 mutations in the RBD. This is not accurate because the RBD also contains all of the mutations present in the RBM.

This is a good point. We have clarified this point.

We thank again the Editor and the Reviewers for their insight. We trust we have addressed all the key points raised and hope that the manuscript is now ready for publication in Nature Microbiology.

Sincerely

Massimo Palmarini

Arvind Patel

[redacted]

RE: Final revision of NMICROBIOL-24041047B “Phenotypic evolution of SARS-CoV-2 spike during the COVID-19 pandemic” by Furnon, Cowton *et al*

I will respond below to the minor remaining issues raised by the reviewers and the technical queries raised on the 8th of November by [redacted]. We'll provide our answer below **in red**

Reviewer #2:

Remarks to the Author:

The authors have adequately addressed most of my concerns. There are only 2 remaining issues:

1. Heatmap is still missing in Fig. S1, as far as I can see.

The heatmap was there but we may have confused the reviewer as it had become in the revised Manuscript Extended Data Figure 2, rather than one. Now the heatmaps are in Extended Data Fig. 2b and 2f.

2. Please cite the latest publication regarding the S371-S373-S375 mutations implicated in the immune evasion of population antibodies. (<https://doi.org/10.1038/s41467-024-51770-3>).

Added – it is now reference # 69

Reviewer #3:

Remarks to the Author:

The authors have responded well to the comments of this reviewer. The authors nicely show that the SARS-CoV-2 S protein is not mutating in a directional manner when assessed phenotypically. The data showing virus replication in NEC continues to a strength of the manuscript.

We thank both reviewers for their insightful comments throughout the review process.

Technical Queries raised by [redacted] on the 8th of November:

- Please organize your source data according to figure numbers. **SourceData_ExtendedDataFig3, 4 and 5 had typo in the tabs labelling. Now corrected accordingly and files were replaced into the Author Approval Folder.**
- Please note that the source data labelling for Ext. Data Figure 3a does not fit the labelling of the main figure. **Indeed, the labelling was misplaced. This is now correct and the file was replaced into the Author Approval Folder.**
- Looking at the source data for Extended Data Figure 8. Please reach out to your handling editor and clarify whether detection for all proteins (N, IFIT1, tubulin, pStat1 and RSAD2) was on the same blot. **No, the panels derive from two different blots - each panel has its own controls.**

- As a guideline, up to 50 references are typical for an article of this type and our maximum are 100 references which is a hard limit. Please consider reducing the number of references where possible while still maintaining good citation practice. For that please consider referencing reviews that summarize several findings, or prioritize to reference only one key paper, if several report the same finding. **We did our best to reduce the number of references. We now have 99 references and could not reduce any further without compromising the citation practice.**
- Please make sure whether the Enlighten repository conforms to our Nature guidelines (See: <https://www.nature.com/sdata/policies/repositories#general>). In particular, institutional generalist data repositories must offer creation of DataCite DOIs, and to allow data to be shared under open terms of use (for example the CC0 waiver). Please reach out to your handling editor and let her know if this is the case. In addition, please let her know if you have been given a timeline for when <https://doi.org/10.5525/gla.researchdata.1698> will be functional. **Yes Enlighten conforms to the nature Guidelines and data have been shared with CC0 waiver. The link should become functional on Monday or Tuesday (11-12 November).**

Sincerely

Massimo Palmarini